# Mixed success for carbon payments and subsidies in support of forest restoration in the neotropics

Katherine Sinacore [1] ✉, Edwin H. García[2], Alex Finkral[3],
Michiel van Breugel [4,5], Omar R. Lopez[6], Carlos Espinosa[7], Andrea Miller[8],
Theodore Howard[9] & Jefferson S. Hall [10]

Restoration of forests in low- and middle-income countries (LMICs) has the potential to contribute to international carbon mitigation targets. However, high upfront costs and variable cashflows are obstacles for many landholders. Carbon payments have been promoted as a mechanism to incentivize restoration and economists have suggested cost-sharing by third parties to reduce financial burdens of restoration. Yet empirical evidence to support this theory, based on robust, dynamic field sampling is lacking. Here we use large, long-term datasets from Panama to evaluate the financial prospects of three forest restoration methods under different cost-sharing and carbon payment designs where income is generated through timber harvests. We show some, but not all options are economically viable. Further work combining growth and survival data from field trials with more sophisticated financial analyses is essential to understanding barriers and realizing the potential of forest restoration in LMICs to help meet global carbon mitigation commitments.

To combat climate change, governments, nongovernmental organizations, and industry have promoted large-scale forest restoration initiatives. Though these initiatives are admirable, the many challenges of restoring deforested areas efficiently, economically, and justly have often proven to be a barrier to achieving the goals these initiatives seek to accomplish[1]. Information on biophysical aspects of forest restoration has outpaced information on socioeconomic challenges, particularly in low- and middle-income countries (LMICs), where social science research has primarily focused on deforestation reduction[2], but not necessarily on the influence of financial incentive structures of restoration. One of the main barriers to restoration is the prohibitively expensive establishment and maintenance costs. Yet upfront costs represent just one financial burden of forest restoration. A second is the disruption of annual cashflows a landholder may be accustomed to

when shifting from a land use that sustains annual cashflows (e.g. cattle ranching) to one that provides cashflows in 15+ years (e.g. timber management). Annual carbon payments could buffer the resistance to land use transitions by replacing cashflows from cattle and other forms of agriculture with cashflows from carbon payments[3]. Fenichel et al.[4] modeled economic solutions to these financial barriers, proposing three cost-sharing scenarios, whereby the establishment costs are completely assumed by a third party, partly assumed by a third party, or assumed completely by the landholder.

In LMICs, where competition for land between forest and agriculture is high, paying for forest carbon has been promoted as an efficient way to support forest restoration[2], despite the dearth of empirical studies that test these theories. There is an especially urgent need for such studies on low soil nutrient sites, where restoration

[1]Rohr Fellow, Agua Salud Project, Smithsonian Tropical Research Institute, Balboa, Ancón, Panamá, Panamá. [2]Facultad de Ciencias Naturales, Exactas y Tecnología, Universidad de Panamá, Ciencias Biológicas, Panamá, Panamá. [3]Eastwood Forests, Chapel Hill, NC, USA. [4]Department of Geography, National University of Singapore, AS2-03-01, 1 Arts Link Singapore, 117568 Singapore, Singapore. [5]Yale-NUS College, College Avenue West, Singapore, Singapore. [6]Instituto de Investigaciones Científicas y Servicios de Alta Tecnología (INDICASAT), Edifício, 209 Clayton, Panamá. [7]Universidad Tecnológica de Panamá, Ancón, Panamá. [8]Geoversity and Sostante, Nature-Based Learning, Clayton, Panamá, Panamá. [9]University of New Hampshire, Durham, NH, USA. [10]ForestGEO, Agua Salud Project, Smithsonian Tropical Research Institute, Balboa, Ancón, Panamá, Panamá. ✉e-mail: ksinacore@gmail.com

efforts are more likely to occur[5] but where financial outcomes might be less certain than on those sites where soil nutrients are higher[6]. While carbon payments offer a potential to offset some of the associated establishment and transition costs, as well as provide cash-in-hand annually, it is an open question as to whether carbon payments and external support of upfront cost-sharing structures will make certain forest restoration strategies financially feasible.

Natural regeneration of secondary forests is one of the lowest-cost forest restoration options. Yet secondary forests are vulnerable in many tropical countries in the Americas because of their low short-term profitability (compared to agricultural uses or development) and tenuous land use rights[2,7–9]. For example, in some Central and South American countries, landholders can lose rights to their land if they do not actively manage the land[3,10]. This has incentivized cutting or burning of land to maintain ownership, even if the landholder does not then use the land for production, such as cattle ranching[11,12]. For secondary forests to persist, landholders must maintain land tenure security and derive value and/or cash flows competitive with alternative uses.

In contrast to naturally regenerating secondary forests, tree plantations can have establishment costs ranging from US\$1,200-US\$1,500 per hectare for non-native and native species monoculture plantations in Panama[13–15], a cost that is comparable to other countries in the region[16,17]. A possible downside to tree plantations is that in many plantations in the neotropics, as elsewhere, only a handful of species are planted, typically in monoculture designs, simplifying forest structure and tree diversity when compared to secondary forests[18]. Yet research over the last few decades on tree plantations has produced robust information on the biophysical aspects affecting the survival and growth of a greater number of native species (see refs. [19–21]). Even so, tree plantations, whether with native or non-native species, may not always sequester the same amount of carbon as secondary forests. While this can be true, there are many tree plantations that do sequester carbon at similar or greater rates to adjacent secondary forest, at least early on in development (e.g. refs. [22–24]).

Another restoration method being promoted, one that endeavors to blend the benefits of secondary forests and tree plantations, is enrichment planting, whereby commercially valuable timber trees are interplanted in either young secondary forests or in underperforming tree plantations (e.g. *Tectona grandis* plantations grown on inadequate soils[25] that may never reach commercial value[15]). Enrichment plantings are promising for three main reasons[26]: First, establishment costs are much lower than tree plantations. Second, newly planted trees can be added in high enough densities such that there is value added to the areas in which they are planted. Third, the previously established trees or vegetation benefits the young seedlings because they can grow in a more favorable, shaded microclimates that can lower incidences of transplant shock[27] and surrounding vegetation can promote straight growth of boles, which improves the timber value.

We take advantage of robust landscape-scale datasets (1.1 million, 250,000, and 18,000 individual tree measurements in naturally regenerated secondary forests, native species plantations, and enrichment plantings, respectively) from long-term research on low fertility soils in a neotropical landscape (that were previously pastures) to assess and address socioeconomic potential (and barriers) of different restoration strategies. Our 15-year tree growth data record includes the wettest year on record (2010, including the flood of record) and the extreme drought of 2015–2016[28] thereby including disturbance extremes which are often excluded in economic modeling[29].

We focus on low fertility soils as these are dominant across the tropics and because it is reasonable to assume that high fertility soils will be required to feed growing populations in these regions or they will be used for other high value/return products based on location[30].

Modeling both variability in tree growth and timber prices, we first compare the net present value of forest restoration methods, specifically naturally regenerating secondary forests, native tree plantations, and enrichment plantings to determine if they can be financially viable and to determine the interest rates at which they are no longer profitable. We then compare three cost-sharing models emerging from the loan program analysis of Fenichel et al.[4] that link natural capital to capital markets - *full* payments where all upfront and management costs are assumed by a third party (e.g., as might happen with development aid), *half payments* where only 50% of the costs are covered by a third party, and a *no payments* option where no costs are covered by a third party. Given that Fenichel et al.[4] found ecosystem service payments to be potentially financially attractive to smallholders in LMICs with missing financial service markets, we modeled annual carbon payments across land uses and financial support models. We followed Fenichel et al.[4] and used a flat payment which assumes similar carbon accrual across treatments, something we know to not always be true[22], but that is a practical solution for quickly upscaling carbon payment programs globally, reducing transaction costs[4], and creating a more equitable program. We therefore also discuss these results in the context of actual carbon accrual as determined by locally derived allometric equations scaled to treatments[15,23,31,32]. Our study demonstrates that fiscal feasibility varies among restoration options and depends not only on species-specific growth and pricing, but also the inclusion of cost-sharing models and/or carbon payments. Our study elucidates the socio-economic conditions under which timber production linked with carbon payments and cost-sharing (e.g. financial support) may and may not be necessary to incentivize different restoration methods, on a strictly financial basis.

## Results
### Net present value without financial structures or carbon payments
Our analysis incorporates species-specific growth, mortality, and price variability. We focus our net present value (NPV) results on interest rates (IRs) of 7%[6], which is the mean of the real interest rates used in forestry in LMICs. The IRs shown are also based on private (financial) discount rates instead of social (economic) rates because the focus of our study is on private land-management decisions. For simplicity, in the two tree planting restoration options (native species plantations and enrichment plantings) we emphasize the two species that are planted in both restoration options, but include the full suite of species in Figs. S4 and S5. All NPVs will be given in US dollars. Any NPVs below zero signify an investment that would not be financially viable.

Without carbon payments or financial cost-sharing, secondary forests had a mean NPV of negative US\$1,781 ha[-1] at 7% IR (which signifies it costs more to harvest the trees than the revenue generated from the harvested trees). The native species plantations and enrichment plantings had NPVs at 7% from negative (e.g. nonprofitable) to positive (e.g. profitable). Specifically, the native species plantations of *Dalbergia retusa* had a mean NPV of US\$7,473 ha[-1] and the *Terminalia amazonia* plantations had a mean NPV of US\$37 ha[-1] (Fig. 1a). In the best-case scenarios, *D. retusa* and *T. amazonia* plantations had a maximum potential NPV of US\$37,744 ha[-1] and US\$13,790 ha[-1] at 7% IR, respectively.

The enrichment planting had lower NPVs at 7% IR than the native species plantings (Fig. 1a). A common species between the two active forest restoration options, *T. amazonia*, had a mean NPV of negative US\$1,041 ha[-1] at 7% interest (Figs. 1a and 3). Like *T. amazonia*, the projected NPV of *D. retusa* in the enrichment planting was less than in the native species plantations, with a mean NPV of negative US\$231 ha[-1] at 7% interest. Cattle ranching – both traditional and silvopasture fell just above and near the zero line, with mean NPVs of US\$1,281 and US\$40 ha[-1], respectively (Fig. 1a, Table 4).

## The effect of cost-sharing financial support and carbon payments on net present value

We applied three cost-sharing models (full, half, and none) to each treatment. We then used a flat or area-based payment (US$130 ha⁻¹ yr⁻¹) as used by Fenichel et al.[4] and used in the well-known PES program in Costa Rica[33]. The effect of only cost-sharing on the NPV of secondary forests was mixed. When half of the costs were covered, the mean NPV was still negative (-US$846 ha⁻¹), but when the costs were fully covered, the mean NPV was US$88 ha⁻¹. However, the carbon payments significantly improved the NPV of the secondary forest, regardless of cost-sharing support. With the carbon payment, secondary forest NPV ranged from US$10,030 ha⁻¹ (no cost-sharing) to US$11,645 ha⁻¹ (full cost-sharing) (Figs. 1a and 2).

For both native species plantations of *D. retusa* and *T. amazonia*, the credible intervals overlapped between the scenarios with carbon payment and without carbon payments (Fig. 4). For *D. retusa* plantations, when carbon payments and full cost-sharing was included, the plantations reached a mean NPV of US$12,477 ha⁻¹. For *T. amazonia*, the cost-sharing and carbon payments maintained the mean NPV above US$0 ha⁻¹ in the half (US$3,117 ha⁻¹) and full (US$5,042 ha⁻¹) cost-sharing scenarios (Fig. 4).

The carbon payment for the enrichment planting increased NPV for both focal species (Fig. 3), with positive NPVs at 7% IR under each cost-sharing scenario. *Dalbergia retusa* enrichment reached a mean NPV of US$2,210 ha⁻¹ with carbon payments and full cost-sharing, while *T. amazonia* enrichment reached a mean NPV of US$1,400 ha⁻¹ under the same conditions (Fig. 3). Additional enrichment species, *Dipteryx oleifera* and *Hieronyma alchorneides*, were more valuable than *T. amazonia* and showed positive NPV when carbon payments were included (Fig. S4), but slow projected growth rates limited the estimated NPV at harvest age (Fig. S1), underpinning the importance of both value of timber and growth differences.

## Net present value of cattle production systems

Cattle management systems represent the dominant non-forested land use in our study region and across much of the neotropics.

Managing existing pastures for cattle in a traditional manner had a positive NPV for all interest rates considered (up to 15%) while the silvopasture system retained a positive cash flow up to 11% interest (Table 3). At an interest rate of 7% NPVs were found to be US$1,281 and US$40 per hectare for traditional cattle and silvopasture, respectively.

## Discussion

Information on the biophysical aspects of forest restoration in low- and middle-income countries (LMICs) is more abundant than ever and socioeconomic research is becoming more nuanced and accessible. Combined, these data sources can shed light on the pathways to designing forest restoration programs that meet discrete objectives. Government pledges and international agreements that target mitigation, will, in part, require large upscaling of forest restoration and inclusion of low- and middle-income landholders to meet national targets. Providing competitive economic incentives to landholders is key to achieving these goals[2], with the rural low-wealth areas requiring the largest incentives (Fenichel et al.[4]). Additionally, developing incentives around forest restoration will be important to reduce financial risk for the landholder and encourage restoration on low fertility soils found across the tropics. Our work suggests that without cost-sharing or carbon payments, some restoration methods would not be financially viable for the landholder through timber-based revenue alone. Even with the most profitable restoration method, native species plantations, the lower limit of the models show NPVs below zero (Fig. 1a). Across the different forest restoration methods, the range of economic value is broad and depends not only on the restoration method (e.g. naturally regenerating secondary forest, enrichment plantings, or native species plantations), but also depends on the species selected for planting, the interest rates, and the timber product prices. The data highlight the context-, species-, and economic-dependent nature underpinning the financial viability of forest restoration. There is evidence to suggest that financial support to cash strapped rural landholders may be necessary for them to assume the financial risk and ensure positive economic returns[14,34], which, at a minimum would require a return on their investment

### (a) Net present value of restoration methods

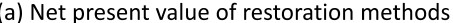
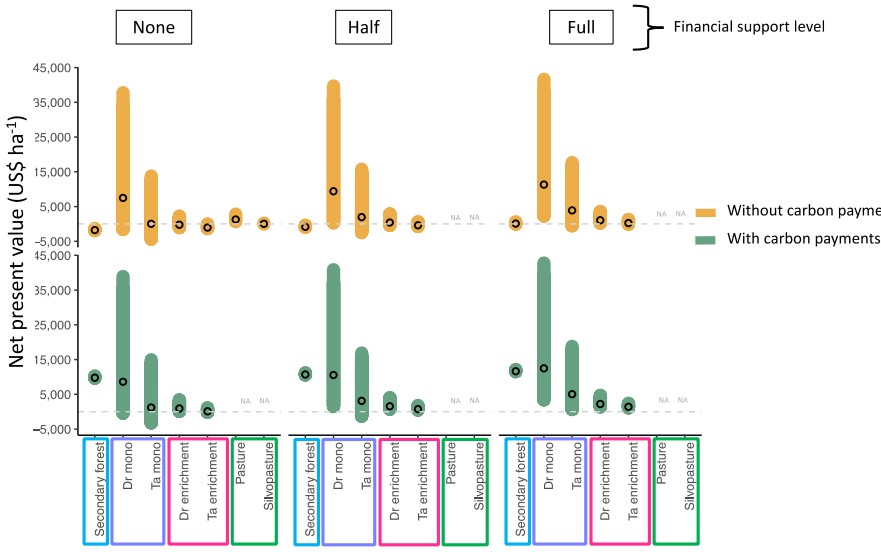

### (b) Restoration and/or land use type

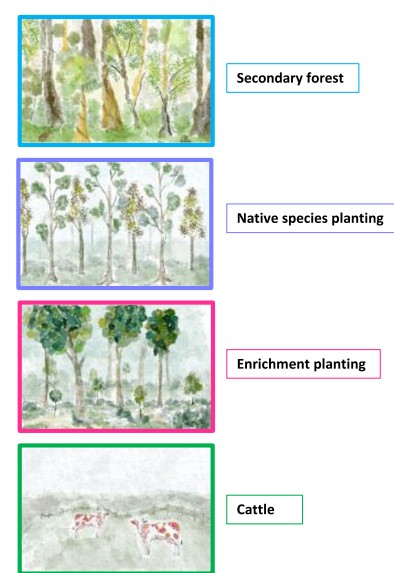

**Fig. 1 | Net present value by land use, support structures, and carbon payments.** Net present value (NPV; US$ ha⁻¹) of three restoration methods and cattle at an interest rate of 7%, including growth and price variability. **a** NPV of land uses with carbon payments (gold) and without carbon payments (green) with no (zero), half, and full financial support. Open circles represent the mean NPV at 7%. **b** Land use types. Colored boxes represent the land use type, with specific species included in the plantation and enrichment planting restoration methods. NA represents not applicable as support structure not applied to these land uses.

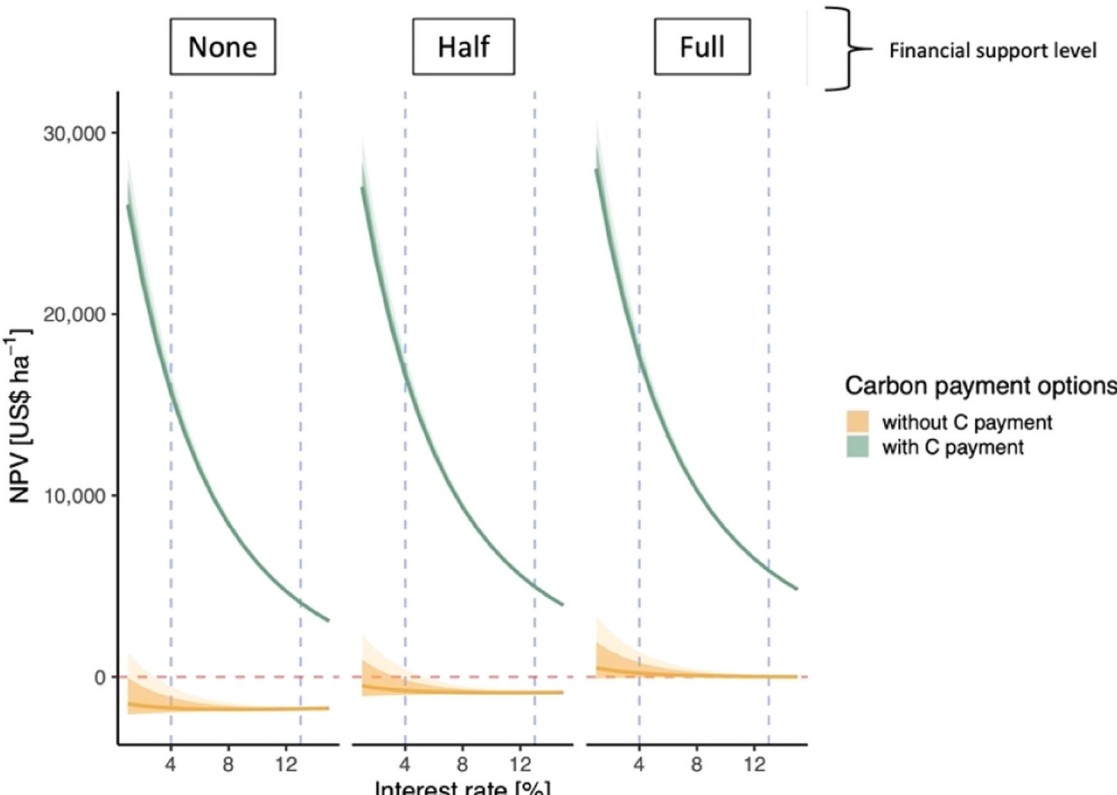

**Fig. 2 | Naturally regenerated secondary forest net present value (US$ per hectare) by interest rate (%).** NPV of ten timber species (see Methods) combined by financial support level and carbon payment options (green: with carbon payment; gold: without carbon payment). The horizontal red dashed line represents where NPV is equal to zero. The vertical blue dashed lines represent an interest rate of 4% and 13%. Lighter shading represents 90% credible intervals and darker shading represents 95% credible intervals around the mean.

superior to what they could obtain via the dominant competing land use (e.g. cattle, be it traditional pasture or silvopasture). In the absence of economic incentives, only the wealthiest landholders will be able to assume the upfront costs associated with tree planting.

Recent advances in economic modeling combining forestry and Payment for Ecosystem Services (PES)[35] have elucidated promising solutions in countries with missing or difficult-to-access financial services like those experienced by many rural residents across the tropics[4,36]. In the absence of those support systems, regenerating secondary forest appears to be the most cost-effective strategy, especially if transitioning from cattle to secondary forest, where the area is already fenced (thus no establishment costs). While some researchers have suggested that timber production in young regenerating secondary forests is economically viable and could justify their protection, management, and thus sustained carbon capture for years and decades into the future[37–39], our work here suggests that secondary forests do not generate sufficient timber income (Fig. 1a). The primary driver was likely the low number of timber trees within a given hectare of forest, combined with harvesting costs of US$65 m$^{-3}$, the upfront cost of fencing (US$200 ha$^{-1}$) and the relatively low wood value of timber species (Table S1). Further, we only used species currently recognized by timber markets. This starkly contrasts with other work in managed forests where NPV per hectare was near US$10,000[40], driven by higher timber volumes (>388 m$^3$ ha$^{-1}$), that might be realized through more targeted silviculture and logging practices[41,42]. In our work, secondary forests could generate an NPV of over US$10,000 ha$^{-1}$ only when combined with receiving an annual carbon payment (Fig. 2). Yet recurring deforestation (or re-deforestation) trends when carbon payments have ceased and land tenure rights challenges in places like Costa Rica suggest not only a complicated financial and policy

landscape[35], but that there are alternative land uses that generate higher revenue than what can be generated through secondary forests (e.g. cattle, development).

Where secondary forest cannot naturally regenerate (e.g., areas where succession is arrested[43,44] or far from mature forest and limited by dispersal[45]), or where landholders prefer to manage a native tree plantation, our results suggest that, even on infertile soils, some native species produce positive NPVs. In fact, we found that the native species plantation of *D. retusa* showed the highest financial profitability (Fig. 1a) under most economic scenarios even without carbon payment or financial support structures (Fig. 4, Fig. S5). Differences between species were largely driven by a combination of growth rates (Figs. S1–3) and timber value (Table 1). Notably, when cost-sharing and carbon payments were included, an additional species from our study (*Anacardium excelsum*) became profitable at 7% IR (Fig S5). We found a similar trend with the enrichment plantations. Although neither *D. retusa* nor *T. amazonia* in enrichment plantations proved to be profitable in the baseline scenario (no cost-sharing or carbon payment), they did become profitable with the addition of the carbon payment. The same was true for the five other enrichment planting species tested that were otherwise not profitable without carbon payments (Fig. S4).

In contrast to the native species plantation and enrichment plantations, *Tectona grandis* plantations (a commonly planted species across the neotropics and under which our enrichment planting is growing) on high fertility soils can reach NPVs of US$ 40,000 ha$^{-1}$ and US$12,700 under medium soil fertility conditions[34] while in low fertility conditions, *T. grandis* plantations in Panama have NPVs that are only positive when interest rates are 2%[15]. It is promising that some, but not all, native species can compete financially with the extensively planted

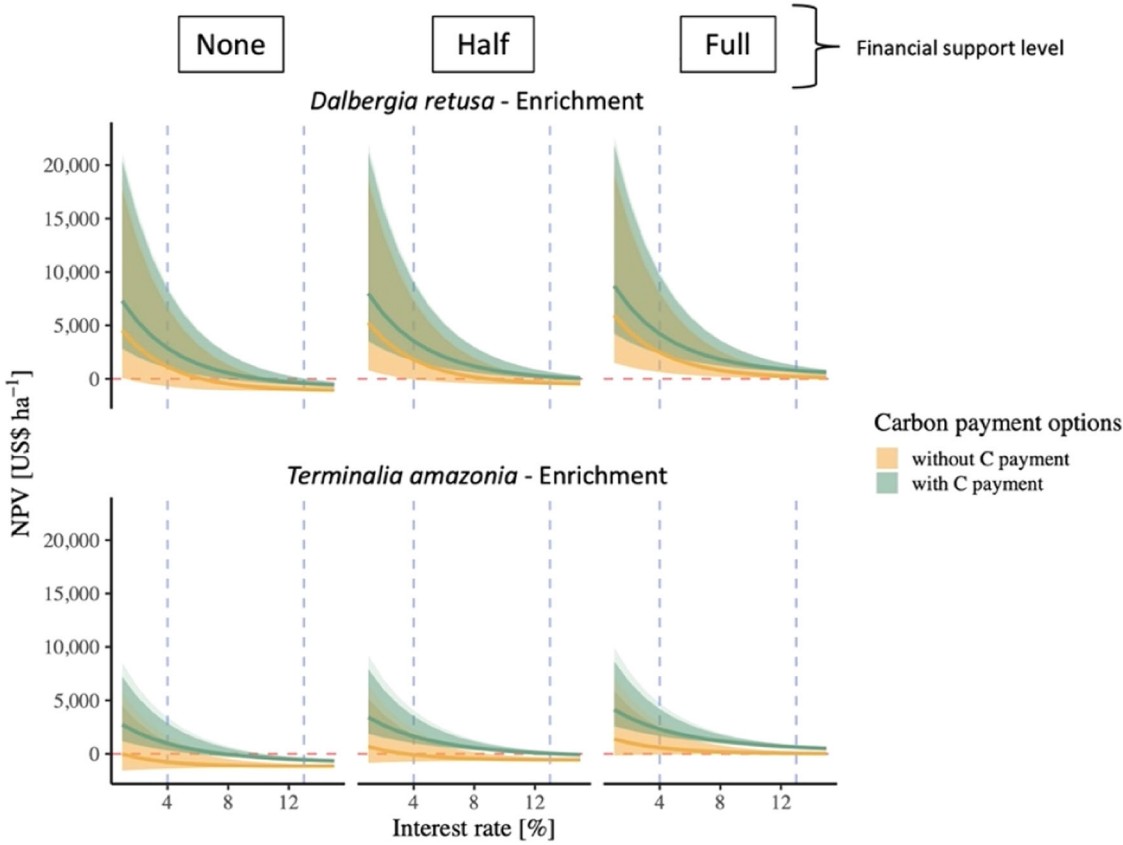

**Fig. 3 | Enrichment plantings net present value (NPV, US$ per hectare) by interest rate (%).** NPV by financial support level and carbon payment options (green: with carbon payment; gold: without carbon payment). The horizontal red dashed line represents where NPV is equal to zero. The vertical blue dashed lines represent an interest rate of 4% and 13%. Lighter shading represents 90% credible intervals and darker shading represents 95% credible intervals around the mean. For full data set of species planted see SI.

*T. grandis.* Further, the enrichment planting at our site is still relatively young and could show increased growth as the trees gain access to full sunlight through the *T. grandis* canopy and with further thinning to release trees from belowground competition[46]. In fact, in sites with poor nutrient soils near our study area, mixtures of *Dipteryx oleifera* (with *Voychysia guatemalensis*) reached an average NPV of US$ 2268 ha$^{-1}$ at 12.9% interest rates, at stocking levels double of that in our enrichment planting[17].

Based on our results, the financially rational decision (based solely on the NPV), would be to plant monocultures of *D. retusa*. While this may be practical in many scenarios, the benefits of diversifying a plantation for improved growth and reduced risk or the additional benefits secondary forests can provide (e.g. biodiversity), cannot be understated. Perhaps promising is the fact that the addition of the flat or area-based carbon payment, proved to be sufficient to push secondary forest into positive net present values (Fig. 1b). While the carbon payment made little difference in the financial viability for species in the native species plantations, primarily due to the high upfront costs that cannot be offset by a carbon payment alone and for any species where the price per cubic meter is high, the contribution of an annual US$130 ha$^{-1}$ payment can be useful when combined with cost-sharing. The combination allows more species to be financially viable options for planting and allows for secondary forests to compete financially with plantations and cattle options. Indeed, these results have been found in other tropical areas, where subsidies for some reforestation projects may be required to make plantings economically viable[47] but may not always be sufficient.

Though it is evident that financial support can impact the economic value of different restoration methods, biophysical factors also affect the financial outcomes. Across the tropics, in general, species-specific growth rates on nutrient-poor soils are varied[23,34,48], highlighting the importance of not only selecting the right forest restoration method for the area and goals, but also being highly selective about matching species to site[49]. Despite the knowledge base on matching species to site becoming well-established in many parts of the world, many forest restoration methods via tree planting have relied on only a few, typically non-native species[22], that may not be adapted to the site, could affect carbon storage capabilities, and could negatively impact biodiversity[34,50–52], but that have enough socio-economic data and market knowledge that reduce barriers to entry, compared to lesser known native species.

Another major barrier to forest restoration across the neotropics is the economic and social value of the competing land use[30], particularly pasture[53]. Cattle ranching provides an important revenue stream for many small landholders and is particularly alluring given that a cow can be sold for revenue when required (unlike trees). Yet low fertility soils common in the lowland tropics pose a challenge to cattle production leading to relatively low cattle density (~1 head per hectare as used herein), which is different from upland areas where high densities have been achieved (e.g. refs. 54–56). This capacity is similar to the average cattle density found in the Brazilian Amazon[36], and could be one contributing factor to the low NPV. However, cattle ranching can involve more than rearing cows on a single pasture and may involve moving cattle between sites or even fattening operations that enhance profitability. Many areas of intensive cattle ranching are focused on sites with different site characteristics from our own and even have different types of cattle. For example, in one study in the Amazonas region of Peru, researchers found that typical pasture and

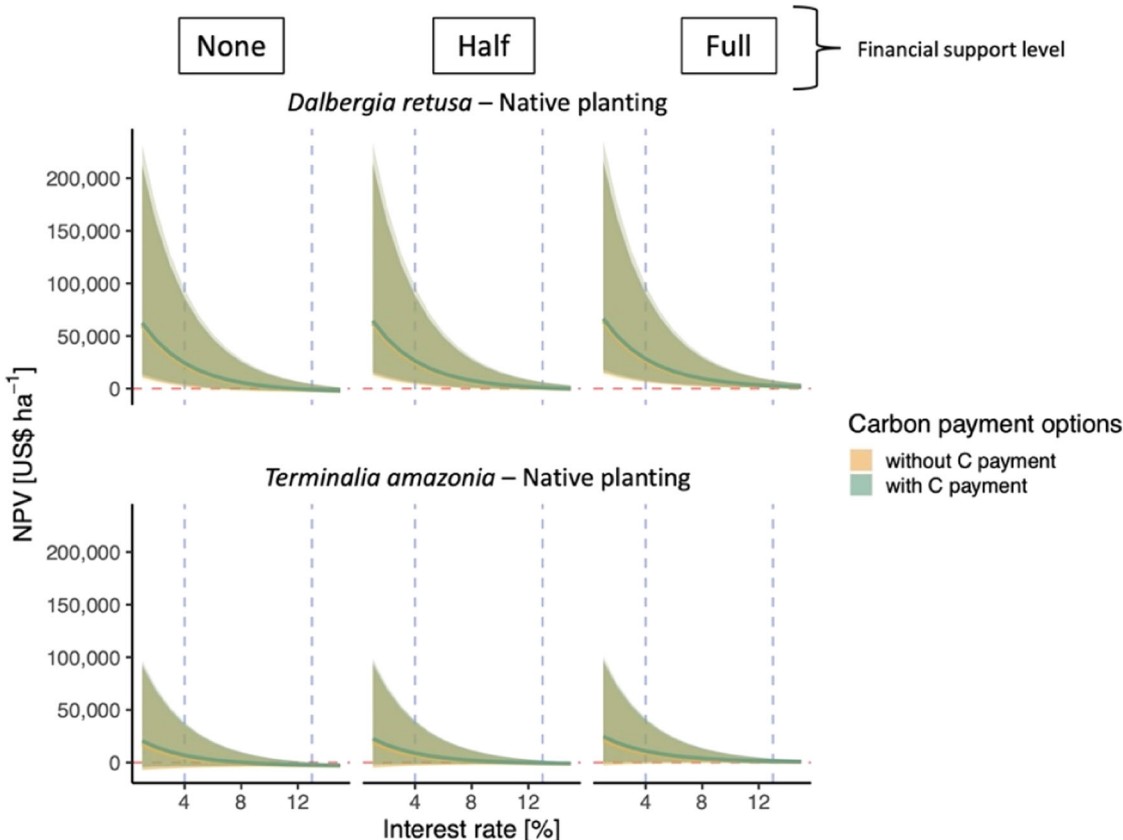

**Fig. 4 | Native species plantation net present value (US$ per hectare) by interest rate (%).** NPV of two native species plantings by financial support level and carbon payment options (green: with carbon payment; yellow: without carbon payment). The red dashed lines represents where NPV is equal to zero. The vertical blue dashed lines represent an interest rate of 4% and 13%. Lighter shading represents 90% credible intervals and darker shading represents 95% credible intervals around the mean. Note: the NPV with and without carbon payments nearly overlap entirely across the range of interest rates and NPVs for all three species. For full data set of species planted see SI.

silvopasture systems generated NPVs of US$318 and US$321 per hectare (8% interest rate), respectively[57]. Our sensitivity analysis (varying cattle density) shows an effective NPV of US$502 on traditional pasture and US$24 per hectare (8% interest rate) on silvopasture at 0.85 head per hectare, with an increase in the silvopasture to US$58 per hectare when cattle densities are 2 head per hectare (Table S3). While we erred on the conservative side (using 1 head per hectare cattle densities, see Methods), the results suggest that cattle density does have a strong effect on the NPV of the two systems. Improved financial outcomes can be achieved with the support of an externally supported extension program[36].

Without a carbon payment system, neither native species plantings, enrichment plantings, nor secondary forests provide an economic benefit until 30+ years, unless there are cashflows that occur in the interim, a timeframe that is unreasonable for most landolders. However, with carbon payment benefits paid annually, the payment can offset the value accrued through selling cattle annually (if carbon payments can be guaranteed). Nevertheless, it is unclear, and a limitation in this study, whether the NPV of secondary forests, native species plantations, or enrichment plantations, combined with the carbon payment and financial support, is sufficient to motivate the transition away from cattle and toward forest restoration via succession or planting, despite the NPV exceeding that of the two cattle land uses in certain scenarios (see e.g., refs. 58). Additional work is needed to understand the psychology and socioeconomic drivers of decision making at the landholder scale, considering both the social and cultural aspects of cattle ranching. Further, any forest restoration policy agreement can be strengthened with strong collaboration and input

from farmers and landholders, particularly to reduce uncertainty in joining a restoration program[59].

Wealthy individuals and large corporations control large areas of land used for agriculture in the tropics[35], areas that can be assumed underlain by relatively fertile soils. Similarly, industrial plantation forestry in many areas of the tropics are also on relatively fertile soils and driven by a few tried and true species[13,34]. However, the areas historically available for restoration in tropical areas are on infertile soils[60]. While wealthy individuals, large corporations, and industrial forestry companies can better weather the volatility of markets and uncertainty, a small landholder has fewer resources to do so. And while advances in silviculture and forest economics show promise for forestry on these infertile soils[15], the landholder still assumes the risks associated with forest restoration and economic uncertainty.

There are many pathways forward to restoring forest cover in the neotropics, which is critical to combatting climate change. Supported by international agreements, forest restoration is being promoted as a key step to not only mitigate climate change, but also restore biodiversity and other ecosystem services[61]. While international agreements are becoming more nuanced, forest restoration projects often overlook the human dimension – that most areas available for forest restoration are being managed by people who make choices and decisions based on their own goals, preferences, and experienced economic pressures. Our work puts estimates of carbon accumulation and forest growth into the socioeconomic context in a lowland, seasonal, moist region of the neotropics. It shows that even on low fertility soils, when species are matched to site and subsidies applied, there are multiple options available to enhance carbon accumulation, improve

**Table 1 | Value of each species in US$ per cubic meter**

| Species | Restoration land type | US$ per cubic meter Low- | Mid- | High- | Source(s) |
|---|---|---|---|---|---|
| *Anacardium excelsum* | SFD, NSP | 40 | 108 | 450 | Finkral |
| *Byrsonima crassifolia* | EP | 50 | 75 | 100 | Finkral |
| *Carapa guianensis* | EP | 40 | 108 | 175 | Finkral, FAO |
| *Chrysophyllum argenteum* | SFD | 40 | 50 | 60 | Costa Rican Prices |
| *Cordia allidora* | SFD | 50 | 150 | 474 | Costa Rican Prices |
| *Dalbergia retusa* | NSP, EP | 500 | 1500 | 4000 | Finkral, Costa Rican Prices |
| *Dipteryx oleifera* | EP | 50 | 174 | 1300 | Finkral, FAO, Costa Rican Prices |
| *Ficus insipida* | SFD | 40 | 50 | 60 | Costa Rican Prices |
| *Hieronyma alchorneoides* | SFD, EP | 40 | 260 | 825 | Costa Rican Prices, FAO |
| *Ochroma pyramidale* | SFD | 120 | 200 | 380 | Costa Rican Prices |
| *Ormosio coccinea* | SFD | 40 | 50 | 60 | Costa Rican Prices |
| *Pachira quinata* | NSP | 40 | 50 | 60 | Costa Rican Prices |
| *Platymiscium pinnatum* | EP | 40 | 342 | 480 | Finkral, FAO, Costa Rican Prices |
| *Tabebuia rosea* | NSP | 40 | 50 | 60 | Costa Rican Prices |
| *Terminalia amazonia* | SFD, NSP, EP | 30 | 216 | 600 | Finkral, FAO, Costa Rican Prices |
| *Vochysia ferruginea* | SFD | 40 | 90 | 175 | Finkral |
| *Zanthoxylum procerum* | SFD | 40 | 50 | 60 | Costa Rican Prices |

Restoration land type shows which species are found in secondary forest (SFD), native species planting (NSP), and enrichment planting (EP). The low-, mid-, and high-range prices are prices in US$ per cubic meter of volume. Sources show where the information was found: Finkral is a co-author and works in the forest industry in Panama, Costa Rica, and other countries across the Americas, FAO is the Food and Agricultural Organization, and Costa Rican Prices are from the publicly available government data.

livelihoods, and meet other landholder objectives. However, we do not focus on other benefits that forest restoration may provide (e.g. biodiversity or other ecosystem services). For example, a land use that has high carbon may not be the same land use that maximizes biodiversity or water regulation. Whether ecosystem benefits are positively correlated or trade-off is site-specific and critically important to assess.

**Challenges and future directions**

A major caveat to our work is that we use a flat carbon payment for all land uses. From our previous work, we know that secondary forest in central Panama accrues carbon at an annual smoothed rate of 6.9 Mg $CO_2e$ in aboveground tree biomass[31] (AGB) or approximately US$130 ha$^{-1}$ at US$18 per Mg $CO_2e$. Our per hectare payment is the annual payment that Panama Canal Authority pays to protect mature forest under private ownership[53]; our carbon accrual rate is equivalent to the average growth in AGB during the first 30 years of regrowth across the neotropics[62]. Plantation carbon accrual for monocultures and mixtures, including *Terminalia amazonia* are similar to those of our secondary forest but lower for other timber species and for the enrichment plantation (Table 3, Table S3). If we were to base our carbon payment on actual carbon rates, this might incentivize planting monospecific plantations of a single species, reduce the diversity of species planted, and/or lead to highly variable payments within and between treatments over time (SI and Fig S6). Given the need to scale rapidly for reforestation to make a difference in the global carbon cycle[63], a flat per area carbon payment with or without planting subsidies for each land use may be the most reasonable alternative to reach the ambitious climate agreement goals and incentivize planting with a greater mixture of species, rather than limiting species selection to only a few highly productive species.

Further, moving away from a flat payment and toward a 'market-efficiency' payment based on the actual carbon on the landscape poses potential social equity issues and ignores the multidimensional socioecological systems in which restoration happens on the ground. For example, Costa Rica's payment for ecosystem service program has been improved overtime to redress inequities that originated when corporations and larger landholders were included more than small

landholders for market-efficiency reasons[64]. The literature on social equity in restoration is quite robust and there are many facets to be considered, too many to include in this manuscript alone (see refs. 64–70). Yet an important theme throughout the case studies and choice experiments is that many landholders in LMICs prefer flexible PES contracts and guaranteed payments over upfront lump sums, varied payments, or shorter contracts[33,68]. Finally, we do not cover an important point about how certain environmental, social, or cultural values may be affected by the extraction of timber every 30 years, but the final decision to harvest is left to the landholder and constrained by society.

A second caveat is that our work does not include transaction costs (TAC), which may include insurance, monitoring, or regulatory approval of a carbon sequestration project[33]. In most projects globally, total costs are underestimated by up to 30% because transaction costs are not included[71]. Comparing transaction costs across projects and continents, Pearson et al.[71] found that insurance costs can be as much or greater than 80% of the transactions costs and found them to be approximately 40% of these costs in the reforestation project evaluated in South America. Insurance costs are handled by creating a buffer pool for carbon. In a review of protocols dominating forest carbon certification, Haya et al.[72] report buffer pools ranging from 7% to 27% of the total carbon. Sinacore et al.[73] found coarse roots of trees excavated within plantations near our study sites to represent approximately 27% of total tree dry biomass (carbon), while Hall et al.[31] report that taken together, non-tree ecosystem components add an additional 60 percent of carbon. Given these data, sufficient carbon buffers exist in our study to account for insurance in our analysis. Flat payments should reduce monitoring costs, the second largest component of TAC.

Though our work is strictly economic and does not intend to make social or cultural value judgements on different land restoration methods, a crucial aspect moving forward will be to expand efforts to understand the social and environmental impacts of forest restoration on people and communities and to understand if there are certain conditions that would enhance interest in participating in forest restoration activities.

## Methods

### Site description

The different research sites are located within the Agua Salud Project of the Smithsonian Tropical Research Institute in Panama (9° 13'N, 79° 47'W 330 m amsl). Agua Salud experiences an annual average rainfall of 2700 mm. The site has a 4-month dry season from end of December until May and an 8-month long wet season from May until end of December, with the heaviest rains occurring in May, October, and November. In December of 2010, the region experienced an estimated 300- to 500-year return rain event during what was already one of the wettest years on record[74]. The region experienced an El Niño event in 2015–2016, where rainfall fell more than 50% below the average for the site[28]. Mean daily minimum and maximum temperatures are 23 °C and 32 °C. Agua Salud soils are strongly weathered, infertile and well-drained Oxisols (Inceptic Hapludox) and Inceptisols (Oxicand Typic Dystrudepts[75]), with total and plant available (Mehlich III) phosphorus averaging 225 ppm and 1.2 ppm, respectively[23,76]. The data for this paper come from four of the land uses in the Agua Salud Project – the secondary forest network, native species plantations, enrichment planting, and silvopasture/pasture systems.

### Secondary forest network

Our secondary forest data come from plots established in the Agua Salud Project secondary forest research network, which consists of 108 plots established in 2008 and 2009 with 20 × 50 m (0.1 ha) dimensions[77]. The plots cover 15 km$^2$ and in 2008, most plots ranged from 0 to 25 years in age, with 14 estimated to be > 40 years of age; the data set includes measurements of over 120,000 independent stems and 1.1 million measurements[31]. All stems with diameter at breast height (DBH) ≥ 5 cm have been measured each year (apart from 2017), with stems ≥ 1 cm measured in half of each plot. We identified 23 timber species currently being traded on local, regional, and/or international timber markets from the secondary forest network dataset. A list of those species and the number of individuals is included in Supplementary Table S1. A total of 13 species were excluded from future analysis due to too few individuals, both inhibiting our ability to model growth, mortality, and volume and having a negligible impact on stand value.

### Native species plantations

In 2008, a native species plantation was established on 75 ha of previous pastureland. The area is divided into 2 blocks, 3 km distant of 37.5 ha each. The total area was divided into 267 plots. Monocultures of five target species (*Terminalia amazonia*, *Dalbergia retusa*, *Pachira quinata*, *Tabebuia rosea* and *Anacardium excelsum*) were planted in 27 m by 23.4 m (core, measured plots), with 3 m spacing between trees (1283 trees planted per hectare). The same size and design were implemented for every two-species mixture of the target species and the target species mixed with five companion species. Since 2008, annual inventories of all focal plots (*n* = 267) measured the diameter at breast height (DBH, cm), basal diameter (BD, cm) and height (H, m) of all trees (*n* = 25,000; > annually until 2016. After 2016, DBH and BD were measured, but H was excluded because many of the trees were taller than the 15-m pole used to measure heights. Monocultures were included in the analyses here as references of growth and net present value potential. In addition to monocultures plots, the five target species were also planted in two-species mixtures (where each of the target species is combined with the other) and in five-species mixtures (where all five target species are planted together). Neither of these mixture treatments is included in this paper, but inventory results can be found in a previously published papers (see[49,15]).

### Enrichment planting

At the same time the native species plantations were established in 2008, the teak plantations were established following the same design as the native species plantations (see above) but later thinned from 1100 to 740 trees per hectare[25]. Starting in 2016, an enrichment planting of native tree species was interplanted where thinned teak trees were removed in 2015. Between 2016 and 2018, six species (*Byrosonima crassifolia*, *Carapa guianensis*, *Dalbergia retusa*, *Dipteryx oleifera*, *Hyeronima alchorneides*, *Platymiscium pinnatum*, *Terminalia amazonia*) were planted on 6 by 6 m spacing both at our core site and two other sites within an 8 km radius. A total of 245 plots were inventoried annually since 2018 (most recent inventory in 2022). The DBH (cm), BD (cm), and H (m) of all 3,420 enrichment trees were included (approximately 18,000 measurements; for more information on design and early results see ref. 46). Growth and financial trajectories of *Tectona grandis* trees can be found in more detail in Sinacore et al.[15].

### Silvopasture and pasture systems

The pasture and silvopasture systems were integrated into the project in 2008 and 2013, respectively, where both had been traditional pastures for over 3 decades but the silvopasture underwent conversion with support from and following the silvopasture requirements of the Panama Canal Authority (ACP). Both systems retain some tree cover. Following ACP guidelines, dispersed trees were planted into the silvopasture at a density of 200 seedlings per hectare; however, most seedlings died. Improved pasture was also planted in the silvopasture system. Both systems have living fences, old remnant trees throughout the grassy areas and streamside trees, but whereas in the silvopasture buffers exceeded 10 m on either side along all stream banks and were fenced, the pasture trees buffered less than 50% of stream banks, were often one tree deep and were not fenced. The silvopasture has additional paddocks and fencing to rotate cattle.

### Estimation of volumes and carbon

Merchantable stem volume of the trees (excludes branches, leaves, and roots) was calculated by species- and site-specific equations when available (for more detail, see Table S2). For *T. amazonia*, *D. retusa*, and *P. quinata*, data were used to calculate stem volumes from García et al.[78] and follow Sinacore et al.[15]. For species in the native species plantations other than those three species, we used the *T. amazonia* equation as the tree species exhibited similar form. For species in the enrichment planting for which we did not have species-specific equations, we used a multi-species aboveground biomass (AGB) equation specific to the site[78] and multiplied it by a form factor of 0.5 to estimate stem volumes (as per[79]). Species- and site-specific volume equations are not available for the other species on our list. For species in the secondary forest, we used the *T. amazonia* stem volume equation to estimate the volume of the other species with sufficient individuals to model stand volume over time. For more information on equations and volume projections, see Supplementary Materials (Table S2, Figs. S1-S3). Aboveground biomass (AGB) was calculated based on methods and equations available to our sites[31,78,80]. The modeled AGB per hectare for each restoration method were converted to carbon dioxide equivalent (CO$_2$e) per hectare[81].

Future stem and stand volumes (based on stem, not total tree volumes), were estimated using the *stan* and *brms* packages[82,83] in R[84]. We selected a Bayesian approach over a frequentists approach for three reasons: (1) the approach's flexibility in specifying models that are appropriate for the data[85] (2) the robustness of parameter values and their credible intervals (CI) (i.e., CI do not depend on large-N approximations), and (3) the ability to combine uncertainty of volume estimates over time with a range of financial data, to more realistically show the potential tree growth and financial value possibilities across a range of land uses and tree species. Estimates were developed by species and land use, projected to 30 years, a rotation age estimate provided by our contacts in the forestry industry in Panama. The framework for the modeling is outlined in Sinacore et al.[15]. Briefly, the

**Table 2 | Costs of establishment and maintenance costs by land use**

| Restoration land type | Expense | Cost (US$ per ha) - no support |
|---|---|---|
| SFD | Fencing | 200 |
| NSP | Establishment | 1500 |
| | Maintenance year 1 | 800 |
| | Maintenance year 2 | 725 |
| | Maintenance year 3 | 625 |
| | Maintenance year 4 | 600 |
| EP | Establishment | 500 |
| | Maintenance year 1 | 270 |
| | Maintenance year 2 | 242 |
| | Maintenance year 3 | 208 |
| | Maintenance year 4 | 200 |

Table shows costs when there is no financial support included. Half support scenarios take half the amount of costs therein. Land use type codes are as follows: *SFD* secondary forest natural regeneration, *NSP* native species plantation, *EP* enrichment planting.

**Table 3 | Carbon dioxide equivalent (CO$_2$e; megagrams per hectare) for each land use and treatment**

| Land use | Treatment | Estimate | Lower | Upper |
|---|---|---|---|---|
| Secondary forest | Natural Regeneration | 185.81 | 91.78 | 323.60 |
| Native species planting | Ta mono | 221.79 | 189.35 | 254.01 |
| | Dr mono | 33.72 | 28.42 | 39.12 |
| | 5 species mixtures | 132.09 | 108.78 | 155.49 |
| | Ae & Ta | 181.09 | 149.68 | 210.94 |
| | Dr & Ta | 182.41 | 154.31 | 210.05 |
| | Pq & Ta | 188.51 | 159.50 | 218.27 |
| | Ta & Tr | 138.38 | 115.78 | 159.37 |
| Enrichment planting | Dr enrichment | 55.74 | 38.79 | 73.03 |
| | Ta enrichment | 52.97 | 35.92 | 70.79 |

Estimate is the mean CO$_2$e and the lower and upper values are the credible intervals around the estimate. Treatment codes are as follows: *Ta Terminalia amazonia, Dr Dalbergia retusa, Ae Anacardium excelsum, Pq Pachira quinata, Tr Tabebuia rosea*, 5 spp mixtures (all five previous species together); *Bc Byrosonima crassifolia, Cg Carapa guianensis, Dp Dipteryx oleifera, Ha Hyeronima alchorneides, Pp Platymiscium pinnatum.*

model uses age to predict mean stand volume or carbon dioxide equivalent by species and land use, calculating credible intervals of 95% and 90% around the predicted mean, following Salles et al.[86]. Computation outputs were checked to ensure convergence and chain resolution (ESS < 1.01 for all outputs).

## Net present values and carbon payments

The financial viability of the land uses – secondary forest extraction, native species plantations, enrichment planting – was determined by calculating the net present value (NPV, a time series of future cash flows discounted by an interest rate to the present time). The outputs from the stand volume estimates (mean, minimum and maximum volume) were extracted at age 30. Those stand volumes were then multiplied by market timber price (in terms of timber-extractive land uses) to estimate the stand revenue. We calculated NPV using the range of timber pricing, volumes, and costs (also see Tables 1 and 2). We varied the real rate of interest from 1 to 15% using the following equation:

$$NPV = -C_\theta + [(R_t - C_t) \div (1 + r)^t] \qquad (1)$$

where $C_\theta$ is the initial costs (US$), $R_t$ is the revenue (US$ at time $t$), $C_t$ is the cost (US$ at time $t$), and $r$ is the interest rate. All monetary values are given in US dollars. We modeled all NPV values based on 30-year cycles. Year 0 is considered the year the restoration was initiated (2008 for the secondary forest and native species plantations). Since the enrichment planting was started in 2016, we set that as year 0 so the NPVs would be comparable.

We have a robust model of carbon growth over time for our secondary forest where an average of 6.9 Mg CO$_2$e per ha are accrued annually in aboveground tree growth for the first 30 years of growth[31]. The Panama Canal Authority currently has a payment for ecosystem services program (PES) to protect mature forest for US$130 ha$^{-1}$. Using the rate paid for protecting forests here we have recently started paying some landholders US$130 per hectare per year for a secondary forest enrichment planting program such that our average carbon payment is US$18.84 per Mg CO$_2$e. While we present estimates of biomass accrual in our treatments over time in Figs. S1-S3, we do not use a carbon payment linked to growth in this analysis (but see Table 3 and Table S3 for carbon accrual estimates by restoration methods and Table S6 for a comparison of pricing based on variable growth rates). Significant uncertainty and differences in carbon accrual occur across the tropics, particularly in relation to the tree planting treatments such that in most areas it would be difficult to estimate with precision and

accuracy future carbon accrual for all but a few commercial and often non-native species. However, our work finds similar CO$_2$e values between some of the native species modeled in this paper and the secondary forest (Table 3). For this reason and reasons of social equity (see discussion), we apply a carbon payment of US$130 ha$^{-1}$ per year whereby the landholder cedes the rights to the sequestered carbon to the person or group making the payment. While carbon stocks in secondary forests in Agua Salud are high for moist tropical forests, at approximately 185 Mg CO$_2$e at 30 years[31] (Table 3), they are about average for all secondary forests combined (dry, moist, and wet; see[62]). Thus, to avoid the uncertainty of actual carbon accrual at sites and draw inferences about a more generalizable carbon payment program, we use a fixed carbon payment. For secondary forests, native species plantations, and enrichment planting, we estimated a second NPV based on the landholder receiving US$ 130 ha$^{-1}$ for years 5−30. This carbon payment is categorized as a carbon payment but assumes some uncertainty and range of carbon sequestration across different land uses (see more in discussion) and even within the same land use (e.g. monoculture plantations as shown in Bukoski et al.[22]).

We recognize that carbon pricing is often discounted for risk and uncertainty. Uncertainty or risk in achieving the actual carbon accrual value depends in part on the potential for perturbations[29]. We note that our study period included two of the most extreme years in relation to abundant rainfall, including a 300-to-500-year flood and a drought spanning over a year and a half representing one of the most extreme droughts during the 100-year rainfall record in central Panama (see above). We also note that through locally collected and landscape scale sampling, coarse roots, soil carbon, lianas, and coarse woody debris contributed an additional 60% carbon accrual beyond carbon sequestered in AGB[31,73] such that the actual carbon accrued per year in the secondary forest is 11.0 Mg CO$_2$e ha$^{-1}$. The additional 4.1 Mg CO$_2$e ha$^{-1}$ provide a significant buffer in the face of known and unknown risks.

## Net present value of the cattle options

We calculated the net present value of pasture and silvopasture from data collected by one of the authors (AM) as well as through personal conversations with landholders around the area of Agua Salud. The maximum capacity for cattle in the watershed in which we work is equal to 1 cow per hectare, a higher stocking for both treatments than in our focal farms[87] but similar to reported densities in the Brazilian Amazon[36]. Generally, a cow can give birth to a calf every two years.

**Table 4 | Net present value (NPV; US\$ ha⁻¹) of pasture and silvopasture by interest rate (%)**

|  | Pasture | Silvopasture |
|---|---|---|
| Interest rate (%) | NPV (\$US per ha) | |
| 1 | 2803.97 | 223.20 |
| 2 | 2407.05 | 167.41 |
| 3 | 2086.45 | 126.40 |
| 4 | 1825.05 | 95.85 |
| 5 | 1609.93 | 72.69 |
| 6 | 1431.28 | 54.80 |
| 7 | 1281.59 | 40.68 |
| 8 | 1155.06 | 29.28 |
| 9 | 1047.23 | 19.86 |
| 10 | 954.58 | 11.89 |
| 11 | 874.39 | 5.01 |
| 12 | 804.47 | -1.05 |
| 13 | 743.09 | -6.48 |
| 14 | 688.87 | -11.40 |
| 15 | 640.69 | -15.91 |

Most of the male calves are sold at age 2 for fattening or finishing as is common in the Brazilian Amazon[36,88], equivalent to US\$300 per calf. Occasionally, when a calf is particularly robust, the owners will keep them and raise them for 5 years. The owners estimate they raise one of these more robust calves every 5 years and they are sold for US\$600. The cost of raising the cows includes vaccines, feed, salt, building and maintaining fencing, manually cleaning pastures of woody shrubs and other unwanted vegetation. Based on the project's pasture and silvo-pasture costs, we estimate that the cost per year to maintain silvo-pasture is US\$200 ha⁻¹ and the cost per year to maintain pasture is US\$100 ha⁻¹. We estimated the NPV of silvopasture and pasture based on this information and using Eq. 1 from the main manuscript (Table 4). Our costs and revenues fall within the range found in Stefanski et al.[6] (conducted in similar areas)[89]. We note that in real-world practice, the conversion of traditional cattle pasture to silvopasture as well as initial maintenance costs of the latter, are borne by the Panama Canal Authority for significant areas within the Panama Canal Watershed[53]. In contrast to the tree planting methods, repeated visits to pastures to check on and tend cows are not monetized in this analysis. We do not include a carbon payment in the NPV for avoided cattle emissions or for trees on the silvopasture treatment. We did not have good data on either for our system, so have not included that in the cattle analysis.

### Reporting summary

Further information on research design is available in the Nature Portfolio Reporting Summary linked to this article.

## Data availability

All the relevant tree inventory data are available through Jefferson Scott Hall (https://stri.si.edu/scientist/jefferson-hall). All data related to pricing are in the Supplementary Materials. All volume data will be available on the first author's OSF page, provided here: https://osf.io/3gqz2/?view_only=e992606bc46f4499ad49b3288f60455b.

## Code availability

All codes will be available on the first author's OSF page, provided here: https://osf.io/3gqz2/?view_only=e992606bc46f4499ad49b3288f60455b.

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

## Acknowledgements

The work was supported by the Agua Salud Project and the Smart Reforestation® program at the Smithsonian Tropical Research Institute (STRI). The funding for this study was provided by the Working Land and Seascapes program at the Smithsonian Institute's Conservation Commons and well as the Mark and Rachel Rohr Foundation. Core support to support the long-term inventories of all the sites comes from Frank and Kristin Levinson, Stanley Motta, and the Hoch Family. We would like to thank Mario Bailón, Johana Balbuena, Guillermo Fernandez, Julia Gonzalez, Miguel Nuñez, Anabel Rivas, Adriana Tapia, Daniela Weber, Estrella Yanguas, Adrian Brox, and Connor Breton for the work measuring the sites. We would also like to thank landholders who shared their cattle data with us.

## Author contributions

J.S.H. and K.S. conceived the study. K.S., J.S.H. and Mv.B. designed the study. K.S., E.H.G. and A.M. collected the tree volume and cattle economic data. K.S. carried out the data analysis with input from J.S.H., Mv.B., A.J.F., O.L. and T.H. K.S. and J.S.H. wrote the manuscript, and it was edited by all coauthors.

## Competing interests

The author declares no competing interests.
