## [Peer Review File · Nature Communications]

Reviewers' Comments:

Reviewer #1:

Remarks to the Author:

I would like to express my gratitude for the opportunity to review this interesting manuscript. The authors contribute to answering an interesting question: under which circumstances could forest restoration become economically interesting for landowners? They address this question by studying how upfront finance and carbon payments could mitigate some of the barriers for landowners to transition to forest restoration, exploring three options of forest restoration considering three different cost-sharing scenarios and the potential income of carbon payments. The calculation of a Net Present Value is an appropriate methodology for rendering explicit the economic viability of a transition towards forest restoration. Finally, the manuscript is generally very well written despite some minor language issues. Below I will expound a small number of general suggestions for improvement of the manuscript, while in the attached file I have provided some minor comments in the text.

Firstly, this study addresses the question of what makes a transition towards forest restoration economically feasible in different scenarios of upfront finance and carbon payments. The authors compare this with NPV of cattle production systems. Yet the NPV in this sector seems rather low compared to the restoration systems. Could you explain why this is the case? How does this compare with other countries (e.g. Brazil) where cattle production is a prominent contributor to GDP? Even with carbon payments alone, restoration becomes more competitive than cattle production, which is the "dominant non-forest land use" in Panama. Does this then entail that the "prohibitive" initial costs are the principal barrier to such transitions? All these questions arise from the relatively low NPV compared to restoration methods, so it would be nice to elaborate on this.

The discussion/conclusion could be improved in two ways. Firstly, what are the implications of the results of this study for other countries in South America (and other continents). Perhaps the authors could spend a few sentences on which parameters will likely vary, including some examples. Particularly, in what way with the NVP of pastures change in export-driven sectors in countries like Brazil, and what does this mean for the economic competitiveness of financed restoration projects?

Another improvement in the discussion/conclusion involves the inclusion of a statement of caution in relation to what the text now says about carbon payments and payments for ecosystem services. While the results underscore that carbon payments can render restoration projects competitive with pastures, it is important to nuance the extension to which restoration prompted by carbon payments will not unequivocally "restore biodiversity and other ecosystem services". The monoculture plantations of native species, for instance, may have lower levels of biodiversity and other ecosystem services compared to naturally regenerated secondary forests or even enrichment planting despite higher NPVs reported here. Furthermore, the value losses as a result of extracting timber after the 30-year period used here has not been included in this study.

**Mixed success for carbon payments and subsidies in support of forest**
**restoration in the neotropics**

-

**Abstract**

Restoration of forests in low- and middle-income countries (LMICs) has the
potential to contribute to international carbon mitigation targets. However, high
upfront costs and variable cashflows pose barriers for many landowners. Carbon
payments have been promoted as a mechanism to pay for restoration and economists
have suggested cost-sharing by third parties should reduce financial barriers to
restoration. Yet empirical evidence to support this theory, based on robust, dynamic
field sampling is lacking. Here we use large, long-term datasets from Panama to
evaluate the financial prospects of three forest restoration methods under different
cost-sharing and carbon payment schemes where income is generated through
timber harvests. We show some, but not all options are economically viable. Further
work combining growth and survival data from field trials with more sophisticated
financial analyses is essential to understanding barriers and realizing the potential of
forest restoration in LMICs to help meet global carbon mitigation commitments.

**Introduction**

To combat climate change, governments, nongovernmental organizations, and industry
have promoted large scale forest restoration initiatives. Though these initiatives are
admirable, the many challenges of restoring deforested areas efficiently, economically and
justly have often proven to be a barrier to achieving the goals these initiatives seek to
accomplish¹. Information on biophysical aspects of forest restoration has outpaced
information on socioeconomic challenges, particularly in low-and-middle income countries
(LMICs), where social science research has primarily focused on deforestation reduction²,
but not necessarily on the influence of financial incentive structures of restoration. One of
the main barriers to restoration is the prohibitively expensive establishment and
maintenance costs. Fenichel et al³ modeled economic solutions to these financial barriers,
proposing three cost-sharing scenarios, whereby the establishment costs are completely
assumed by a third party, partly assumed by a third party, or assumed completely by the
landowner. Yet upfront costs represent just one financial barrier to forest restoration. A
second is the disruption of annual cashflows a landowner is accustomed to when shifting
from a land use that sustains annual cashflows (e.g. cattle ranching) to one that provides
cashflows in 15+ years (e.g. timber management). Annual carbon payments could buffer
the resistance to land use transitions by replacing income from cattle and other forms of
agriculture with income from carbon payments.

In LMICs, where competition for land between forest and agriculture is high,
paying for forest carbon has been promoted as an efficient way to support forest
restoration², despite the dearth of empirical studies that test these theories. There is an
especially urgent need for such studies on low soil nutrient sites, where restoration efforts
are more likely to occur⁴ but where financial outcomes might be less than those sites where
soil nutrients are higher⁵. An additional obstacle to transitioning away from cattle-ranching
and toward carbon rich land use with timber-focused revenue streams is that the wait time
for accruing financial assets is longer (15-50 years) than it is for cattle (can be sold annually
or when cash is needed^{3,6}). While carbon payments offer a potential to offset some of those
costs, as well as provide cash-in-hand annually, it is an open question under which
scenarios of carbon and timber prices, carbon payment schemes, and external support of
upfront cost-sharing associated with tree planting are feasible.

Secondary forests are vulnerable in many tropical countries in the Americas because
of their low short-term profitability (compared to agricultural uses or development) and
tenuous land use rights². For example, in some Central American countries, landowners can
lose rights to their land if they do not actively manage the land^{6,7}. This has incentivized
cutting or burning of land to maintain ownership, even if the landowner does not then use
the land for production, such as cattle ranching^{8,9}. For secondary forests to persist,
landowners must maintain land tenure security and derive value and/or cash flows
competitive with alternative uses.

In contrast to naturally regenerating secondary forests, tree planting requires
significant upfront costs which can create a financial burden for many landowners. For
example, establishment costs can range from \$1,200-\$1,500 per hectare for non-native and
native species monoculture plantations in Panama¹⁰⁻¹². While tree planting in the neotropics
has generally resulted in only a handful of species selected, and typically in monoculture
designs, research on active forest restoration via planting has produced robust information
on the biophysical aspects affecting survival and growth of a diversity of native species (see
¹³⁻¹⁵). While not all these restoration land uses have the same climate change mitigation
capabilities as secondary forests, there are many planted forests that do, sequestering
carbon at similar or greater rates to adjacent secondary forest, at least early on in
development (e.g. ¹⁶⁻¹⁸).

Another restoration method being promoted is enrichment planting, whereby
commercially valuable timber trees are interplanted in either young secondary forests or in
underperforming tree plantations (e.g. *Tectona grandis* plantations grown on inadequate
soils¹⁹ that may never reach commercial value¹²). Enrichment plantings are promising for
three main reasons²⁰: First, establishment costs are much lower than tree plantations.
Second, newly planted trees can be added in high enough densities such that there is value-
added to the areas in which they are planted. Third, the previously established trees or
vegetation benefits the young seedlings because they can grow in a more favorable, shaded
microclimates that can lower incidences of transplant shock²¹ and surrounding vegetation
can promote straight growth of boles, which improves the trees' value.

We take advantage of robust landscape scale datasets (1.1 million, 250,000, and
18,000 individual tree measurements in secondary forest, native species plantations, and

enrichment planting, respectively) from long-term research²³ in a low soil fertility
neotropical landscape to assess and address socioeconomic potential (and barriers) of
different restoration strategies. Our 15-year tree growth data record includes the wettest
128 year on record (2010, including the flood of record) and the extreme drought of 2015-16²²
thereby including disturbance extremes which are often excluded in economic modeling²³.

We focus on low fertility soils as these are dominant across the tropics and because
it is reasonable to assume that high fertility soils will be required to feed growing
populations in these regions or they will be used for other high value/return products based
on location²⁴. Modeling both variability in tree growth and timber prices¹⁶, we first
compare the net present value of forest restoration methods, specifically secondary forest
succession, tree planting, and enrichment planting to determine if they can be financially
viable and to determine the interest rates at which they are no longer profitable. We then
apply payment for ecosystem service options, found by Fenichel et al³ to be potentially
financially attractive to small holders in LMICs with missing financial service markets. We
compare two financial support models emerging from Fenichel et al³ from their analysis of
loan programs - *full* payments where all upfront and management costs are assumed by a
third party (e.g., as might happen with development aid), *half payments* where only 50% of
the costs are covered by a third party, with a *no payments* option where no costs are
covered by a third party. We further overlay these programs with carbon payments assumed
annually. Flat carbon payments assume similar carbon accrual across treatments, something
we know not to be true¹⁶, but is a practical solution for quickly upscaling carbon payment
programs globally. We therefore also discuss these results in the context of actual carbon
accrual as determined by locally derived allometric equations scaled to treatments^{12,17,25,26}.
Our study demonstrates that fiscally feasibility varies between options and depends not
only on species-specific growth and pricing, but also the inclusion of financial support
models. Our study elucidates the socio-economic conditions under which carbon payments
and financial support may and may not be necessary to incentivize the different restoration
methods, on a strictly financial basis.

**Results**

**Net present value without financial structures and carbon payments**

Our analysis incorporates both species-specific growth, mortality, and price variability; for
simplicity we emphasize the two species planted in both the plantations and enrichment
plantings for these restoration options. Without carbon payments or financial cost-sharing,
secondary forests had negative net present values (NPV; US\$ ha⁻¹) at 7% interest, the target
rate typical for forestry investments in the tropics⁵. The native species plantations and
enrichment plantings had NPVs at 7% from negative (e.g. non-profitable) to positive (e.g.
profitable), with the native species planting of *Dalbergia retusa* showing the greatest range
(from just under \$0 ha⁻¹ to just under \$40,000 ha⁻¹) (Fig 1a). In contrast, the native species
plantation of *Terminalia amazonia* showed a smaller range of positive NPVs at 7% interest
rate, ranging from just under \$0 ha⁻¹ to just under \$15,000 ha⁻¹) (Fig 1a).

The enrichment planting had lower NPVs at 7% interest rate than the native species
plantings (Fig. 1a). A common species between the two active forest restoration options, *T.*
*amazonia*, had a mean NPV below \$0 ha⁻¹ at 7% interest (Fig 1a, Fig 3). Like *T. amazonia*,
the projected NPV of *D. retusa* in the enrichment planting was less than in the native
species plantations, with a mean NPV below zero at 7% interest. Cattle ranching – both
traditional and silvopasture fell just above and right on the zero line, respectively (Fig. 1a).

**The effect of cost-sharing support and carbon payments on net present value**

The effect of cost-sharing and carbon payments significantly improved the NPV of the
secondary forest. In all the cost-sharing scenarios with the carbon payment, secondary
forest NPV was around \$10,000 ha⁻¹ (Fig 1a).

Carbon payments for the enrichment planting increased NPV for both focal species
(Fig. 3), with positive NPV at 7% interest rates under each cost-sharing scenario.
*Terminalia amazonia* enrichment scenarios had mean NPV's less than \$2,500 ha⁻¹ at 7%. In
contrast, NPV of *D. retusa* in enrichment planting was just under \$5,000 ha⁻¹ at 7% with
the inclusion of the carbon payment and half- or full- cost-sharing (Fig. 3). Additional
enrichment species, *Dipteryx oleifera* and *Hieronyma alchorneides*, were more valuable
than *T. amazonia* and showed positive NPV when carbon payments were included (Fig.

S4), but slow projected growth rates limited the estimated NPV at harvest age (Fig. S1),
 underpinning the importance of both value of timber and growth differences.

For the native species plantings, the NPV at 7% was unchanged with the addition of
 the carbon payments (Fig. 4) in the absence payments. For *D. retusa* in the plantations, the
 cost-sharing scenarios did not significantly change the NPV, with means of above zero in
 the half and full support options, respectively at 7%. For *T. amazonia*, the cost-sharing
 increased the mean NPV above \$0 ha⁻¹ in the half- and full- scenarios (Fig. 4)

Net present value of cattle production systems

Cattle management systems represent the dominant non-forested land use in our study
 region and across much of the neotropics. Managing existing pastures for cattle in a
 traditional manner had a positive NPV for all interest rates considered (up to 15%) while
 the silvopasture system retained a positive cash flow up to 11% interest (Table 3). At an
 interest rate of 7% NPVs were found to be \$1,281 and \$40 per hectare for traditional cattle
 and silvopasture, respectively.

**Fig. 1 Net present value by land use, support structures, and carbon payments.** Net present value (NPV;
 \$US ha⁻¹) of three restoration methods and cattle at an interest rate of 7% including growth and price
 variability. (a) NPV of land uses with carbon payments (gold) and without carbon payments (green) with no
 (zero), half, and full financial support (b) Land use types. Colored boxes represent the land use type, with
 specific species included in the plantation and enrichment planting restoration methods.

Fig. 2 Secondary forest net present value (US\$ per hectare) by interest rate (%). NPV of ten timber species (see Methods) combined by financial support level and carbon payment options (green: with carbon payment; yellow: without carbon payment). The horizontal red dashed line represents where NPV is equal to zero. The vertical blue dashed lines represent an interest rate of 4% and 13%. Lighter shading represents 90% credible intervals and darker shading represents 95% credible intervals around the mean.

Fig. 3 Enrichment planting net present value (NPV, US\$ per hectare) by interest rate (%). NPV by financial support level and carbon payment options (green: with carbon payment; yellow: without carbon payment). The horizontal red dashed line represents where NPV is equal to zero. The vertical blue dashed lines represent an interest rate of 4% and 13%. Lighter shading represents 90% credible intervals and darker shading represents 95% credible intervals around the mean. For full data set of species planted see SI.

 **Fig. 4. Net present value (US\$ per hectare) by interest rate (%).** NPV of two native species plantings by
 financial support level and carbon payment options (green: with carbon payment; yellow: without carbon
 payment). The red dashed lines represents where NPV is equal to zero. The vertical blue dashed lines
 represent an interest rate of 4% and 13%. Lighter shading represents 90% credible intervals and darker
 shading represents 95% credible intervals around the mean. Note: the NPV with and without carbon payments
 nearly overlap entirely across the range of interest rates and NPVs for all three species. For full data set of
 species planted see SI.

Discussion

Information on the biophysical aspects of forest restoration in low- and middle-income
 countries (LMICs) is more abundant than ever, but socioeconomic research has lagged.
 Government pledges and international agreements will require large upscaling of forest
 restoration and inclusion of low- and middle-income landowners to meet national targets.
 Providing competitive economic incentives to landowners is key to achieving these goals².
 Our work shows that developing incentives around forest restoration will be important to
 reduce financial risk and encourage restoration on low fertility soils found across the
 tropics. We found evidence that the range of economic value is broad and depends not only
 on the forest restoration method (e.g. secondary forest succession, enrichment planting, or
 plantation forestry), but also depends on the species selected for the planting methods, the
 interest rates, and the product prices. The data highlight the context-, species-, and
 economic-dependent nature underpinning the financial viability of forest restoration. Our
 results demonstrate that in all cases financial support to cash strapped rural landowners will

be necessary for them to assume the financial risk and ensure positive economic returns^{11,28},
which, at a minimum would require a return on their investment superior to what they could
obtain via cattle, be it traditional pasture or silvopasture. In the absence of economic
incentives, only the wealthiest landowners will be able to assume the upfront costs
associated with the two active planting methods.

Wealthy individuals and large corporations control large areas of land used for
agriculture in the tropics²⁷, areas that can be assumed underlain by relatively fertile soils.
Similarly, industrial plantation forestry in many areas of the tropics are also on relatively
fertile soils and driven by a few tried and true species^{10,28}. However, the areas historically
available for restoration in tropical areas are on infertile soils²⁹. It is an open question as to
the extent to which combinations of wood pricing and carbon credits can encourage
reforestation¹¹, particularly on infertile soils. Yet tropical forests contain dozens to
hundreds of species recognized by local, regional, and international markets^{13,30}, and
advances in silviculture and forest economics show promise for forestry on these infertile
soils¹².

Recent advances in economic modeling combining forestry and Payment for
Ecosystem Services (PES)²⁸ have elucidated three potentially promising solutions in
countries with missing or difficult to access financial services like those experienced by
many rural residents across the tropics. Some authors have suggested that timber
production in young regenerating secondary forests is economically viable and could justify
their protection, management, and thus sustained carbon capture for years and decades into
the future³¹⁻³³. Yet re-deforestation trends when carbon payments have ceased and land
tenure rights challenges in places like Costa Rica suggest a complicated financial and
policy landscape²⁷. We found that secondary forest timber values had a negative NPV at all
interest rates, and where 90% credible intervals on the high end were only slightly above
zero (Fig. 2). The primary driver was likely the low number of timber trees within a given
hectare of forest, combined with harvesting costs of \$65 m⁻³, the upfront cost of fencing
(\$200 ha⁻¹) and the relatively low wood value of timber species (Table S1). Further, we
only used species currently recognized by timber markets. This starkly contrasts with other
work in managed forests where NPV per hectare was near \$10,000³⁴, driven by higher

timber volumes ($>388 \text{ m}^3 \text{ ha}^{-1}$), that might be realized through more targeted silviculture
and logging practices^{30,35}.

We found that the native species plantings showed the highest financial profitability
(Fig. 1a) under most economic scenarios even without carbon payment or financial support
structures (Fig. 4, Fig. S5). Species achieving a mean positive NPV ranged breakeven rate
of 6% interest rates (\$US 111), at or below the minimum rate desired to make forestry
investments in the region^{14,29} to \$US 12,900 ha^{-1} (1% interest rates) for *Anacardium*
*excelsum*, to a mean breakeven interest rate of 11 % (\$103 to \$59,000 ha^{-1} at 1% interest for
*D. retusa*). *Terminalia amazonia* fell ~~between~~ but at the lower end of this scale (\$37 ha^{-1} at
7% interest rate to \$17,700 ha^{-1} at 1% interest rate). Differences between species were
largely driven by a combination of growth rates (Figs S1-3) and timber value (Table 1,
Methods).

In contrast to the establishment and maintenance costs of native species plantings,
the enrichment planting had lower initial costs. Even so, the projected value of the
enrichment species, while sufficient for four of the species (*D. retusa*, *D. oleifera*, *H.*
*alchorneides*, and *T. amazonia*), still generally had NPVs lower than zero, except at interest
rates less than 4% (Fig. SI4). In contrast, *T. grandis* plantations (a commonly planted
species across the neotropics and under which our enrichment planting is growing) on high
fertility soils can reach NPVs of US\$ 40,000 ha^{-1} and \$12,700 under medium soil fertility
conditions²⁸ while in low fertility conditions, *T. grandis* plantations in Panama have NPVs
that are only positive when interest rates are 2%¹². The enrichment planting at our site is
still relatively young and could show increased growth as the trees gain access to full
sunlight through the *T. grandis* canopy and with further thinning to release trees from
belowground competition⁴⁷. While the native species in the enrichment planting may not all
achieve NPVs like those found for *T. grandis* on high quality sites, in sites with poor
nutrient soils near our study area, mixtures of *Dipteryx oleifera* (with *Voychysia*
*guatemalensis*) reached an average NPV of US\$ 2,268 ha^{-1} at 12.9% interest rates, at
stocking levels nearly double of that in our enrichment planting³⁶.

The addition of the carbon payment (US\$ 130 ha^{-1}), proved to be sufficient to push
secondary forest into positive net present values (Fig. 1b). The carbon payment made little
difference in the financial viability for species in the native species plantations, primarily

due to the high upfront costs that cannot be offset by a carbon payment alone and for any
species where the NPV is exceptionally high ($> \$10,000 \text{ ha}^{-1}$), the contribution of $\$130 \text{ ha}^{-1}$
is minimal. Only with the inclusion of full upfront costs paid, do the NPV of native species
or two native species plantations stay above zero (Fig. 1b), with the other three near or
below zero (Fig. S5). With half of the upfront costs paid, NPV also remains above zero
except for in the poorest growing and least favorable economic conditions.

In contrast, the secondary forest shows the largest benefit of the carbon payment,
with mean NPV exceeding $\$10,000 \text{ ha}^{-1}$, largely driven by a single species, *T. amazonia*,
which is the most abundant timber species in the secondary forest. As expected, the full and
half financial support provided the larger increase in NPV than that of the carbon payment
alone. The carbon payment alone also did not prove to be sufficient for all species in the
enrichment planting (Fig. 3). Mainly, *B. crassifolia* and *C. guinanensis* remained
financially unsuitable even with the carbon payment (Fig. S4). For enrichment planting
species that fell within a range of positive NPVs, the range of NPV across interest rates still
fell below zero in the full and half subsidy scenarios. These results have been found in
other tropical areas, where subsidies for some reforestation projects may be required to
make plantings economically viable³⁷ but may not always be sufficient.

The biophysical factors affect the financial outcomes, with the most productive trees
having the highest probability of positive NPVs. Across the tropics, in general, species-
specific growth rates on nutrient poor soils are varied^{17,28,38}, highlighting the importance of
not only selecting the right forest restoration method for the area and goals, but also being
highly selective about matching species to site. Despite the knowledge base on matching
species to site becoming well-established in many parts of the world, many forest
restoration methods via tree planting have relied on only a few, typically non-native
species¹⁶, that may not be adapted to the site, could affect carbon storage capabilities, and
could negatively impact biodiversity^{28,39-41}, but that have enough socioeconomic data and
market knowledge that reduce barriers to entry, compared to lesser known native species.

A barrier to forest restoration across the neotropics is the value of the competing
land use²⁴, ~~a main one being~~ pasture⁴². Cattle ranching provides an important revenue
stream for many small landowners and is particularly alluring given that a cow can be sold
for revenue when required (unlike trees). Without a carbon payment system, neither native

species plantings, enrichment plantings, nor secondary forests provide an economic benefit
until 30+ years, a timeframe that is unreasonable for most landowners. However, with
carbon payment benefits paid annually, the payment can replace the value accrued through
selling cattle annually (if carbon payments can be guaranteed). Nevertheless, it is unclear,
and a limitation in this study, whether the NPV of secondary forests and enrichment
planting, combined with the carbon payment and financial support, is sufficient to motivate
transition away from cattle and toward forest restoration via succession or planting, despite
the NPV exceeding that of the two cattle land uses. Additional work is needed to
understand the psychology and socioeconomic drivers of decision making at the landowner
scale, considering both the social and cultural aspects of cattle ranching.

[revised manuscript text omitted]

mixture nor the two-species mixture are included in this paper (see ⁵⁰) and for financial
differences between monocultures and mixtures see ¹².

**Enrichment planting**

At the same time the native species plantations were established in 2008, the teak
plantations were established following the same design as the native species plantations
(see below) but later thinned from 1100 to 740 trees per hectare²². Starting in 2016, an
enrichment planting of native tree species was interplanted where thinned teak trees were
removed in 2015. Between 2016 and 2018, six species (*Byrsonima crassifolia*, *Carapa*
*guianensis*, *Dalbergia retusa*, *Dipteryx oleifera*, *Hyeronima alchorneides*, *Platymiscium*
*pinnatum*, *Terminalia amazonia*) were planted on 6 by 6 m spacing both at our core site and
two other sites within an 8 km radius. A total of 245 plots were inventoried annually since
2018 (most recent inventory in 2022). The DBH (cm), BD (cm), and H (m) of all 3,420
enrichment trees were included (approximately 18,000 measurements; for more information
on design and early results see ⁵¹). Growth and financial trajectories of *Tectona grandis*
trees can be found in more detail in Sinacore et al¹².

**Estimation of stem volumes**

Merchantable stem volume of the trees (excludes branches, leaves, and roots) was
calculated by species- and site-specific equations when available (for more detail, see Table
S2). For *T. amazonia*, *D. retusa*, and *P. quinata*, data were used to calculate stem volumes
from García et al. (in prep) and follow Sinacore et al¹². For species in the native species
plantations other than those three species, we used the *T. amazonia* equation as the tree
species exhibited similar form. For species in the enrichment planting for which we did not
have species-specific equations, we used a multi-species aboveground biomass (AGB)
equation specific to the site (García et al. in prep) and multiplied it by a form factor of 0.5
to estimate stem volumes (as per ⁵²). Species- and site-specific volume equations are not
available for the other species on our list. For species in the secondary forest, we used the
*T. amazonia* stem volume equation to estimate the volume of the other species with
sufficient individuals to model stand volume over time. For more information on equations
and volume projections, see Supplementary Materials (Table S2, Figs. S1-S3).

Future stem and stand volumes (based on stem, not total tree volumes), were
estimated using the *stan* package⁵³ in R⁵⁴. We selected a Bayesian approach over a
frequentists approach for three reasons: (1) the approach’s flexibility in specifying models
that are appropriate for the data⁵⁵ (2) the robustness of parameter values and their credible
intervals (CI) (i.e., CI do not depend on large-N approximations), and (3) the ability to
combine uncertainty of volume estimates over time with a range of financial data, to more
realistically show the potential tree growth and financial value possibilities across a range
of land uses and tree species. Estimates were developed by species and land use, projected
to 30 years, a rotation age estimate provided by our contacts in the forestry industry in
Panama. The framework for the modeling is outlined in Sinacore et al ¹². Briefly, the model
uses age to predict mean stand stem volume by species and land use, calculating credible
intervals of 95% and 90% around the predicted mean, following Salles et al. 2019.
Computation outputs were checked to ensure convergence and chain resolution (ESS < 1.01
for all outputs).

**Net present values and carbon payments**

Financial viability of the land uses – secondary forest extraction, native species plantations,
enrichment planting – was determined by calculating the net present value (NPV, a time
series of future cash flows discounted by an interest rate to the present time). The outputs
from the stand volume estimates (mean, minimum and maximum volume) were extracted at
age 30. Those stand volumes were then multiplied by market timber price (in terms of
timber-extractive land uses) to estimate the stand revenue. We calculated NPV using the
range of timber pricing, volumes, and costs (also see Table 1 and 2). We varied the nominal
rate of interest from 1 to 15% using the following equation:

$$486 \quad NPV = -C_0 + [(R_t - C_t) \div (1 + r)^t]$$

where C_0 is the initial costs (US\$), R_t is the revenue (US\$ at time t), C_t is the cost (US\$ at
time t), and r is the interest rate. We modeled all NPV values based on 30-year cycles.

We have a robust model of carbon growth over time for our secondary forest where
an average of 6.9 Mg CO_{2e} per ha are accrued annually in aboveground tree growth for the

first 30 years of growth²⁵. The Panama Canal Authority currently has a payment for
ecosystem services program (PES) to protect mature forest for \$130 ha⁻¹. Using the rate
paid for protecting forests here we have recently started paying some landowners \$130 per
ha per year for a secondary forest enrichment planting program such that our average
carbon payment is \$18.84 per Mg CO_{2e}. While we present estimates of biomass accrual in
our treatments over time in Figs. S1-S3, we do not use a carbon payment linked to growth
in this analysis. Significant uncertainty and differences in carbon accrual occur across the
tropics, particularly in relation to the tree planting treatments such that in most areas it
would be difficult to estimate with precision and accuracy future carbon accrual for all but
a few commercial and often non-native species. For this reason, we apply a carbon payment
of \$130 ha⁻¹ per year whereby the landowner cedes the rights to the sequestered carbon to
the person or group making the payment. While carbon accrual rates in Agua Salud are
high, at approximately 110 Mg of aboveground biomass (AGB) at 30 years²⁵ they are about

[revised manuscript text omitted]

211–214 (2016).
- 45. Busch, J. *et al.* Potential for low-cost carbon dioxide removal through tropical
reforestation. *Nat. Clim. Chang.* **9**, 463–466 (2019).
- 46. Pörtner, H.-O. *et al.* Climate change 2022: Impacts, adaptation and vulnerability. *IPCC*
*Sixth Assessment Report* (2022).
- 47. Ogden, F., Crouch, T., Stallard, R. & Hall, J. Effect of land cover and use on dry
season river runoff, runoff efficiency, and peak storm runoff in the seasonal tropics of
Central Panama. *Water Resources Research* 1–82 (2013)
doi:10.1002/2013WR013956.
- 48. van Breugel, M. *et al.* Soil nutrients and dispersal limitation shape compositional
variation in secondary tropical forests across multiple scales. *Journal of Ecology* **107**,
566–581 (2019).
- 49. Püspök, J. Microbial phosphorus immobilization slows soil phosphorus cycling in
tropical secondary succession. (Universität Wien, 2019).
- 50. Mayoral, C., van Breugel, M., Cerezo, A. & Hall, J. S. Survival and growth of five
Neotropical timber species in monocultures and mixtures. *Forest Ecology and*
*Management* **403**, 1–11 (2017).
- 51. Marshall, A., Nelson, C. R. & Hall, J. S. Species selection and plantation management

in enrichment planting with native timber species in the Panama Canal watershed.
*Front. For. Glob. Change* **5**, 925877 (2022).
52. Montagnini, F. Accumulation in above-ground biomass and soil storage of mineral
nutrients in pure and mixed plantations in a humid tropical lowland. *Forest Ecology*
*and Management* **134**, 257–270 (2000).
53. Stan Development Team. RStan: the R interface to Stan. (2022).
54. R Core Team. A language and environment for statistical computing. (2017).
55. Kruschke, J. K. Bayesian Analysis Reporting Guidelines. *Nat Hum Behav* **5**, 1282–
1291 (2021).
56. Grado, S. C., Hovermale, C. H. & St. Louis, D. G. A financial analysis of a
silvopasture system in southern Mississippi. *Agroforestry Systems* **53**, 313–322 (2001).

**Acknowledgements**

-

**Author contributions**

-

Reviewer #2:

Remarks to the Author:

Your paper addresses an important topic, forest restoration in low- and middle-income countries (LMICs) as a climate change mitigation measure. As you point out in the introduction, fundamental questions about the economics of forest restoration in LMICs remain unanswered because economists have focused so heavily on the opposite issue, deforestation. You have a valuable dataset for investigating and comparing forest restoration methods that are relevant in landscapes typical of much of the tropics, not just your study site in Panama. Your financial benefit-cost analysis is not fancy, but that's okay. The strength of your study lies in the importance of its topic, in terms of both research and application, and your data.

I see a big problem with the study in its current form, however: the assumption of an identical carbon payment across the forest restoration methods it investigates. As a result of this assumption, the study ignores differences in the amounts of carbon that are sequestered and their timing, both of which affect carbon payments in most actual forest carbon offset programs and are critical for a science-based understanding of forest carbon sequestration. The study equates the carbon payment to the fixed annual payment the Panama Canal Authority offers to private landowners to protect mature forests. As I understand it, that payment was motivated by a concern about hydrological services, not carbon sequestration, and it refers to forest protection, not forest restoration.

The study would be much more valuable if it investigated the financial incentives for forest restoration generated by payments based on actual amounts of carbon sequestered and carbon prices that are pertinent to current and prospective forest management. By the latter, I mean such prices as typical ones for actual afforestation/reforestation projects (\$5-10/ton CO₂e; see various issues of the World Bank's State and Trends of Carbon Pricing reports and Ecosystem Marketplace's State of the Voluntary Carbon Markets reports); prices needed to hit the Paris Accord targets (many studies estimate these); and the social cost of carbon (see Rennert et al., Comprehensive evidence implies a higher social cost of CO₂, Nature, 2022).

You argue that a uniform, fixed payment is a practical means of scaling up carbon financing for forest restoration, but I am not convinced. There is great policy concern about the integrity of forest carbon offset projects. See, for example, news stories over the past 1-2 years about Lyme Timber CEO Jim Hourdequin; fraud allegations in Australia; and The Guardian's recent expose on REDD+ projects (<https://www.theguardian.com/environment/2023/jan/18/revealed-forest-carbon-offsets-biggest-provider-worthless-verra-aoe>). Governments, conservation organizations, and some companies are demanding better, not cruder, forest carbon payment programs. More accurate measurement of sequestered carbon is increasingly possible thanks to advances in remote sensing, including sub-meter satellite data and remarkable imagery from drone- and aircraft-mounted LiDAR. Companies on the cutting edge of the forest carbon sequestration business, such as Mombak in Brazil, are demonstrating the feasibility of bringing rigorously designed forest carbon projects to market and the willingness of carbon-emitting companies to pay a premium for them.

You appear to have sufficient data to estimate annual amounts of carbon sequestered by the different forest restoration methods you investigate. You refer to annual tree inventories, locally derived allometric equations, etc. Why not take full advantage of these data, and link the resulting annual estimates of sequestered carbon to pertinent carbon prices? The paper currently highlights the effects of varying assumptions about the discount rate on carbon-based financial incentives for forest restoration. Although discount rates have a major impact on financial analyses, the effects of varying assumptions about carbon prices—especially their impacts on the relative attractiveness of the different restoration methods—would be more interesting and more valuable from a policy perspective.

Other comments (please pay special attention to the one about p. 5)

p. 3, line 86: "were" should be "where"; "than those" should be "than on those." These are rare typos in a very clean manuscript; thank you for preparing it so carefully.

p. 4, line 94: You refer to the first restoration method as "secondary forests." Secondary forests are an outcome, not a method. The restoration method that generates them is natural regeneration. I recommend replacing "secondary forests" with "natural regeneration" whenever possible in the paper, including in Fig. 1.

p. 4, line 94: On the vulnerability of secondary forests in the Neotropics, you might also cite Schwartz et al., "Reversals of reforestation across Latin America limit climate mitigation potential of tropical forests" (Frontiers in Forests and Global Change, 2020) and Sloan, "Reforestation reversals and forest transitions"(Land Use Policy, 2022). Holl et al., "Redefining 'abandoned' agricultural land in the context of reforestation" (Frontiers in Forests and Global Change, 2022) might also be relevant.

p. 4, line 102: You refer to the second restoration method as "tree planting." This is confusing, because your third method, enrichment planting, is also a form of tree planting. In a few places, you use "plantation" instead of "tree planting." "Plantation" is more clear than "tree planting." I recommend replacing "tree planting" with "plantation," including in Fig. 1.

p. 5: Somewhere around this point in the paper, you need to explain the initial site conditions for the three restoration methods. Based on information in the Supplementary Materials, I infer that the initial conditions were pastures in all cases, with grazing ceasing and restoration starting in 2008-9. Is that correct? Relatedly, it would be helpful to explain that Year 0 for calculation of the NPVs for all the methods corresponds to 2008-9 and that the NPVs for the three restoration methods do not account for the opportunity cost of the land (i.e., the NPV of pasturing)—right?

p. 6, line 162: I suspect that some readers might quibble with your choice of a 7% discount rate. I agree that it's a suitable rate; I've used it in my own work, with the justification that it equals the mean of the real interest rate in LMICs. You might add a sentence explaining that you use a private (financial) discount rate instead of a social (economic) because you are focusing on private land-management decisions.

p. 7, Fig. 1: I like the layout of the figure, but the figure would be more informative if it displayed box plots instead of bars. Readers will be interested in the distribution of the NPVs, not just the range. Box plots would provide standard distributional information: medians, 25th and 75th percentiles, minima, maxima. The boxes would be small, but I believe they would be readable.

p. 7, Fig. 1: Why does the figure include Silvopasture, given that there are no NPVs for it? On the other hand, why not add Silvopasture as a fourth restoration method and include carbon payments for it?

pp. 8-9: Fig. 2-4 are not essential. Move them to the Supplementary Materials, and replace them with figures showing the implications of varying the carbon price.

p. 9, line 237: You might cite Shyamsundar et al., "Scaling smallholder tree cover restoration across the tropics" (Global Environmental Change, 2022), for additional support on the importance of including smallholders in restoration initiatives.

p. 11, line 280: If restoration systems are financially viable without carbon payments, then the carbon sequestration they provide is not additional and thus would not qualify for carbon payments, would it? Adding some discussion on the implications of your findings for the circumstances under which the different restoration methods deliver incremental sequestration would enrich the paper.

p. 16, line 430: Including the results for mixed-species plantations would strengthen the paper. Why haven't you included them?

p. 16, lines 436-438: This sentence leaves me unclear as to how Year 0 is defined for enrichment planting. Is it defined as 2008 or 2016? For an apple-to-apples comparison to the other restoration methods, it should be defined as 2008, and the NPV for enrichment planting should be defined as the difference between the NPV for a teak plantation with enrichment planting and the NPV for one

without it.

p. 17, lines 483-4: I assume you meant to type "real rate of interest," not "nominal rate of interest," but if you used a nominal rate of interest, did you project the timber prices in Table 1 (and all other project revenues and expenses) forward in time with an inflation rate? If so, what rate did you use?

Reviewer #3:

Remarks to the Author:

This is a very interesting and informative study of the effect of carbon payments and cost share-arrangements on the financial performance of different forest restoration interventions in a particular location in the tropics (Panama), under various discount rates, and of how the financial performance of these restoration options compares to that of conventional and silvopasture alternatives.

The paper is timely and important given the urgency to deploy nature-based solutions in general and forest conservation and restoration in particular, and the methods seem sound. It also is well-written and concise, and contributes to a relatively sparse literature on an important issue: how to make forest restoration financially attractive for tropical and subtropical land operators.

The authors find that some interventions can have substantial NPV, even without carbon (C) payments and cost sharing of initial and maintenance costs, but this finding is sensitive to species choice and discount rate. In line with intuition, individually and especially in combination, C payments and cost sharing improve the financial attractiveness of restoration options, but relatively more so for some than for others.

I strongly support the publication of this manuscript. However, there are a few issues that need to be addressed prior to publication.

1) 245-247: The authors state that "Our results demonstrate that in all cases financial support to cash strapped rural landowners will be necessary for them to assume the financial risk and ensure positive economic returns."

I am not sure that your results alone support this very strong statement. The native species plantings have fairly substantial NPVs, so potentially, even cash-strapped farmers might be able to finance these activities in the absence of subsidies if credit access is available at sufficiently low interest rates. This cost-of-credit would need to be included in the NPV calculations to answer my question. It may well be that sufficiently low rates that would result in positive NPVs are not available; that credit access for the type of landowner relevant to your analysis is limited or that credit; or that landowners would be unwilling to assume the credits for the lower NPV that would result. But none of these factors are discussed in your analysis. I would suggest making your statement less categorical, or introducing the additional information that would support such a conclusion.

To my point above: line 44 states that "We show some, but not all options are economically viable"; and in 279-281, that "We found that the native species plantings showed the highest financial profitability (Fig. 1a) under most economic scenarios even without carbon payment or financial support structures (Fig. 4, Fig. S5)." Both of these are inconsistent with the statement in 245-247 that "in all cases financial support to cash strapped rural landowners will be necessary for them to assume the financial risk and ensure positive economic returns."

2) Discussion: I am glad to see the authors mention the non-financial drivers of land use decisions that affect land manager behavior.

3) 371-372: "There is evidence that carbon prices must be higher than \$100 ton CO₂e to incentivize large-scale forest restoration in the tropics", citing Busch et al (2019). I do not think

that this characterization of Busch et al.'s findings is correct. Busch et al. find that

"A carbon price of US\$20 tCO₂ -1 would incentivize tropical land users to increase reforestation by 31.8 Mha (8.2%) to 419.6 Mha ... from 2020 to 2050 relative to a BAU scenario (Supplementary Fig. 6). ... A carbon price of US\$50 tCO₂-1 would increase reforestation by 84.1 Mha (21.7%) to 471.9 Mha..."

As their Fig. 1 shows, predicted sequestration from reforestation increases nearly linearly with the C payment level, so total additional reforestation extent due to C pricing at US\$ 100 tCO₂-1 is nearly twice that at US\$ 50 tCO₂-1, or around 160 M ha or 1.6 M km²—approximately the size of Mongolia or Iran. I doubt reforestation at this scale can be considered not large-scale. It is true that this level of reforestation is still far below the identified biophysical maximum potential, but it certainly nevertheless is large-scale.

Please reword your statement accordingly.

4) Lines 501-503 state that "we apply a carbon payment of \$130 ha⁻¹ per year whereby the landowner cedes the rights to the sequestered carbon to the person or group making the payment." Then, in lines 507-509, it states that landowners receive "an additional US\$ 130 ha⁻¹ for years 5-30" in the estimation of a second NPV. This is a bit confusing. It suggests to me that landowners holding secondary forests, or implementing native species plantations or enrichment plantings are paid US\$ 130 ha⁻¹ in years 1-4, and then US\$ 260 ha⁻¹ in years 5-30. If so, what is the rationale for this higher payment in years 5-30? On the other hand, in the remainder of the manuscript, I only see mention of US\$ 130 ha⁻¹ yr⁻¹. As a result, I am unsure what the actual payment levels are. Please clarify the description of payment levels.

5) I did not see any mention of C payments for silvopasture. If this is correct, could you explain why tree cover in silvopasture is not receiving any C payments? There certainly are PES programs in tropical countries that support silvopasture establishment, so it would be helpful to understand why no C payments for this land use were considered.

6) Fig 1: Very informative figure. However, I do not understand how traditional pasture can have a positive NPV range in the case with no financial support or C payments, but a zero NPV in the C payments scenario (as indicated by no visible marker in the No C payments panel -- the lower left panel in the figure)? Presumably, traditional pastures do not receive any C payments, so how can it be that C payments affect the NPV of traditional pasture?

7) The authors approach of making C payments independent of actual biomass accumulation is an interesting one, and one that would reduce uncertainty for landowners. Presumably, this also might reduce the transaction costs (TAC) associated with C payments, as the monitoring of C accumulation may be less involved than in performance-based payments on the basis of "actual" C accumulation.

This issue of high TAC of tropical forest C offsets (e.g., Pearson et al. 2014; Rendón Thompson et al. 2013; van Kooten et al. 2002) is not discussed. These costs are a function of market access and market rules (search and contract costs, certification, mandatory inventory reports, mandatory C buffer pools etc.). If the C credits generated by the program studied in this paper do not get transacted on the global voluntary C markets but rather accrue to some local entity (say, the Panama Canal authority), then landowners presumably would face much lower TAC associated from this C scheme than they would on global markets. Still, these costs would not be zero. It would be good to see the topic of TAC at least briefly mentioned in the paper.

On a side note: If the local entity to which the C credits accrue were to seek to transact these credits on a C market (to recoup some of the program's costs), it would then of course need to comply with the requirements of that market, and so this entity would be the actor incurring these high TAC. I realize this would not affect the financial viability of the analyzed reforestation options for participating landowners – the topic of this paper-- but it would affect the overall cost of the program.

References

Pearson TRH, Brown S, Sohngen B, Henman J, Ohrel S. 2014. Transaction costs for carbon sequestration projects in the tropical forest sector. *Mitigation and Adaptation Strategies for Global Change* 19:1209–1222. <https://doi.org/10.1007/s11027-013-9469-8>

Rendón Thompson OR, Paavola J, Healey JR, Jones JPG, Baker TR, Torres J. 2013. Reducing emissions from deforestation and forest degradation (REDD+): transaction costs of six Peruvian projects. *Ecology and Society* 18(1):17. <http://dx.doi.org/10.5751/ES-05239-180117>

van Kooten G, Shaikh S, Suchánek P. 2002. Mitigating climate change by planting trees: The transaction costs trap. *Land Economics* 78(4):559-572. 10.2307/3146853.

Response to Reviewers Outline

Overview	1
Reviewer #1	2
General Comments	2
General Response	2
Line comments and responses.....	5
Reviewer #2	9
General Comments	9
General Response	9
Table RR1. CO _{2e} estimates by land use and treatment (included in Supplementary materials)	10
Fig. RR1 (S6) Annual carbon dioxide equivalent (CO _{2e}) payments (\$US) for focal land uses.....	12
Line comments and responses.....	14
Reviewer #3	18
General comments.....	18
General Response	18
Line comments and responses.....	18
Extended carbon payment discussion	22
Extended cattle discussion	26
Table RR2. Sensitivity analysis of Net Present Value (NPV) of traditional and silvopasture systems in central Panama	29
Extended transaction costs discussion	31
Literature cited	33

Overview

We have divided our response to reviewers into a several different sections to ensure we answer all comments and suggestions by the reviewers and are able to expand upon a few specific concerns raised by more than one reviewer. The first three sections are divided by the general and line comments, followed by our responses. At the end there are **three extended discussions** that address concerns around our use of a **flat carbon payment (pages 22-26)**, how we handled the **cattle analyses (pages 26-31)**, and the **transaction costs (pages 31-33)** associated with reforestation projects. The Table of Contents has hyperlinks that can be clicked to find each section more easily. All line numbers refer to the track changes version of the manuscript.

Reviewer #1

General Comments

I would like to express my gratitude for the opportunity to review this interesting manuscript. The authors contribute to answering an interesting question: under which circumstances could forest restoration become economically interesting for landowners? They address this question by studying how upfront finance and carbon payments could mitigate some of the barriers for landowners to transition to forest restoration, exploring three options of forest restoration considering three different cost-sharing scenarios and the potential income of carbon payments. The calculation of a Net Present Value is an appropriate methodology for rendering explicit the economic viability of a transition towards forest restoration. Finally, the manuscript is generally very well written despite some minor language issues. Below I will expound a small number of general suggestions for improvement of the manuscript, while in the attached file I have provided some minor comments in the text.

General Response

Thank you for reviewing our manuscript, your endorsement, and the helpful comments. We have incorporated many of them and feel they have improved the manuscript. We expound on them below.

Firstly, this study addresses the question of what makes a transition towards forest restoration economically feasible in different scenarios of upfront finance and carbon payments. The authors compare this with NPV of cattle production systems. Yet the NPV in this sector seems rather low compared to the restoration systems. Could you explain why this is the case? How does this compare with other countries (e.g. Brazil) where cattle production is a prominent contributor to GDP? Even with carbon payments alone, restoration becomes more competitive than cattle production, which is the “dominant non-forest land use” in Panama. Does this then entail that the “prohibitive” initial costs are the principal barrier to such transitions? All these questions arise from the relatively low NPV compared to restoration methods, so it would be nice to elaborate on this.

We have wondered for a long time about the profitability of cattle and marveled at the incredible investments, by the World Bank, Inter-American Development Bank, and others, to help promote this sector. We appreciated the excuse (and necessity) to take a deep dive into the literature.

Small landowner cattle ranching in the area where our analyses are done is part of a commodity chain for raising and selling cattle. Generally, in our research area, male calves are only raised until 2-years of age, before they are sold to a larger landowner (often toward Western Panama), for the cattle to be reared until the age (or size) for beef production. For a small

landowner, the carrying capacity is low (often less than 1 cattle head per hectare), herd sizes are generally low, and the main source of food is grass (with minor supplementation given). Based on literature from Brazil, there seem to be more developed export routes than what exists in Panama.

As Reviewer 1 mentioned, it is interesting to consider, why, given the NPV of land restoration ends up being higher than cattle, the transition to land restoration is not more widespread. We hypothesize that one reason may be the high upfront costs of establishment (\$1,200-1500 for plantations, half that for enrichment). While secondary forests do not require substantial upfront costs, we hypothesize that cultural and tenure right pressures also influence cattle over forest restoration. Further, many small landowners do not have access to loans to support the establishment costs or transitions to a new land use.

Based on the literature from we reviewed from several Latin American countries, cattle production is a marginally profitable business, even on relatively fertile soils. The business is complicated, particularly as there are illegal activities that make it opaque in Brazil. Nevertheless, several authors have been able to take a deep look and it appears that cattle production in Brazil benefits from being very large scale and will also be increasingly dependent upon external conservation (e.g., certification for legal production to avoid deforestation) and payments for environmental services, principally carbon. In Panama we believe that cattle production is as much a labor of love and cultural identity as it is a financial investment. Also, as cited in our manuscript and by other authors, it can serve as a sort of savings account for small holders. This latter benefit is at extreme risk due to climate change but is the subject for another time. Please see our extended discussion below on cattle where we take an even deeper dive into the literature on cattle.

The discussion/conclusion could be improved in two ways. Firstly, what are the implications of the results of this study for other countries in South America (and other continents). Perhaps the authors could spend a few sentences on which parameters will likely vary, including some examples. Particularly, in what way with the NPV of pastures change in export-driven sectors in countries like Brazil, and what does this mean for the economic competitiveness of financed restoration projects?

Again, thank you for this comment. We have added this to our manuscript in lines 775 to 849 (discussion):

“Yet low fertility soils common in the lowland tropics pose a challenge to cattle production leading to relatively low cattle density (~1 head per hectare as used herein), which is different from upland areas where high densities have been achieved (e.g., ⁵⁴⁻⁵⁶). This capacity is similar to the average cattle density found in the Brazilian Amazon³⁶, and could be one contributing factor to the low NPV. However, cattle ranching can involve more than rearing cows on a single pasture and may involve moving cattle between sites or even fattening operations that enhance profitability. Many areas of intensive cattle ranching are focused on sites with different site characteristics from our own and even have different types of cattle. For example, in one study in the Amazonas region of Peru, researchers found that typical pasture and

silvopasture systems generated NPVs of US\$318 and US\$321 per hectare (8% interest rate), respectively⁵⁷. Our sensitivity analysis (varying cattle density) shows an effective NPV of US\$502 on traditional pasture and \$24 per hectare (8% interest rate) on silvopasture at 0.85 head per hectare, with an increase in the silvopasture to US\$58 per hectare when cattle densities are 2 head per hectare (Table S3). While we erred on the conservative side (using 1 head per hectare cattle densities, see Methods), the results suggest that cattle density does have a strong effect on the NPV of the two systems. Improved financial outcomes can be achieved with support of an externally supported extension program³⁶.”

Additionally, we have a more thorough discussion in our extended response below.

Another improvement in the discussion/conclusion involves the inclusion of a statement of caution in relation to what the text now says about carbon payments and payments for ecosystem services. While the results underscore that carbon payments can render restoration projects competitive with pastures, it is important to nuance the extension to which restoration prompted by carbon payments will not unequivocally “restore biodiversity and other ecosystem services”. The monoculture plantations of native species, for instance, may have lower levels of biodiversity and other ecosystem services compared to naturally regenerated secondary forests or even enrichment planting despite higher NPVs reported here. Furthermore, the value losses as a result of extracting timber after the 30-year period used here has not been included in this study.

We really appreciate the suggestion about including a more nuanced discussion of how carbon payments will not unequivocally restore biodiversity or other ecosystem services. We have now included this in the discussion (lines 886-893) where we expound upon a more nuanced vision what carbon payments support (and what they may not). This will be particularly important for policy makers to understand as there can be tradeoffs between carbon and ecosystem services (but also sometimes they can be potentially bundled). We write:

“It shows that even on low fertility soils, when species are matched to site and subsidies applied, there are multiple options available to enhance carbon accumulation, improve livelihoods, and meet other landowner objectives. However, we do not focus on other benefits that forest restoration may provide (e.g. biodiversity or other ecosystem services). For example, a land use that has high carbon may not be the same land use that maximizes biodiversity or water regulation. Whether ecosystem benefits are positively correlated or trade-off is site specific and critically important to assess.”

We have also added a caveat to the discussion with Reviewer 1’s suggestion about how certain values may be lost from the extraction of timber at 30 years. This will be particularly important for policy makers to understand as there can be tradeoffs between carbon and ecosystem services (but also sometimes they can be potentially bundled). These values might be cultural, social and/or environmental (lines 938-941).

“Finally, we do not cover an important point about how certain environmental, social, or cultural values may be affected by the extraction of timber every 30 years, but the final decision to harvest is left to the landowner and constrained by society.”

In addition, we have now improved the discussion and conclusions based on Reviewer 1’s suggestion of expanding a discussion of the implications of the study’s results on the rest of South and Central America. We hypothesize that countries where governmental support, micro-banking, and strong forest extension programs are likely to accelerate the transition to more forest restoration buy-in. As a whole, carbon markets that are better developed, with clear standards, are also likely to better support forest restoration transitions. Further, small landowners may not have full access to resources that would allow them to participate in the forestry markets (e.g. selling timber). However, it might be better supported if a minimum number of hectares across multiple landowners is included in a forestry operation and where they can negotiate with buyers of timber.

Line comments and responses

L 96. South America as well, e.g. Brazil

Response: Added South America to L 128.

L 103. Third time this is mentioned [in reference to ‘significant upfront costs’].

Response: We have re-written this to read: “In contrast to naturally regenerating secondary forest, tree plantations can have establishment costs ranging from \$1,200-\$1,500 per hectare for non-native and native species monoculture plantations in Panama.” (L 134-136). We have also reframed the language throughout the introduction and abstract to make this point less repetitive.

L 104. Is this high or low relative to other countries? [In reference to the \$1,200-\$1,500 per hectare establishment costs].

Response: We have found that these values are comparable to other values in countries bordering Panama. We’ve now included that in the text, citing both Montagnini et al., (1995) and Pinnschmidt et al., (2022) . Additionally, Montagnini et al., (1997) found enrichment costs were \$551 per hectare for the first year (establishment and maintenance) and \$231 per hectare for subsequent years (L 137).

L 118. Remove ‘-’ between value and added.

Response: We have now removed the dash (L 152).

L 124. Do you here refer to naturally regenerated secondary forest? This is unclear.

Response: We meant to say natural regenerated secondary forest so have now added that to L 216.

L 129. Above you mention three cost-sharing scenarios. Maintain terminology to avoid confusion.

Response: Thank you for catching this. We have now updated the language; it now reads ‘cost-sharing scenarios’ rather than ‘financial support models’ (L 287).

L 148. Delete ‘ly’ from ‘fiscally’.

Response. Fixed (L 243).

L 162. Please provide values [in reference to the secondary forest NPV without carbon payments or cost-sharing].

Response: We have completely restructured our results and discussion around NPV to include the mean NPV for each land use at 7% as well as other interesting data points that are highlighted throughout and reads (L294-302):

“Without carbon payments or financial cost-sharing, secondary forests had a mean NPV of negative \$1,781 ha⁻¹ at 7% IR (which signifies it costs more to harvest the trees than revenue generated from the harvested trees).”

L 203 | Fig. 1 Caption. From the figure it is difficult to see how the range bars change across financial support scenarios. They shift upwards, but improving the image would make readability much easier.

Response.: Thank you for this suggestion. We have now included an open circle that represents the mean value of NPV. The circle makes it easier to see by how much the NPV changes across land uses and scenarios (L 468 | Fig 1 Caption).

L 225 | Fig. 4 Caption. Remove period and add ‘Native plantation’ [in reference to caption].

Response: We have now fixed this (L 493 | Fig 4. Caption).

L 235. Pretty strong claim. Perhaps nuance [in reference to using writing that socioeconomic research has ‘lagged’].

Response: Thank you. We have now rewritten this sentence to read: “Information on the biophysical aspects of forest restoration in low- and middle-income countries (LMICs) is more abundant than ever and socioeconomic research is becoming more nuanced and accessible. Combined, these data sources can shed light on the pathways to designing forest restoration programs that meet discrete objectives.” (L 502-505).

L 256-258. This is repetition from the introduction. More interesting would be to discuss how this paper contributes to answering this question and which issues remain open [in reference to the open question as to the extent of wood pricing/carbon credits effects on encouraging reforestation].

Response: Thank you for this suggestion. We have completely restructured the discussion to tackle the points raised here (and those raised by the other reviewers as well). We focus more on the comparison among land uses, the potential drawbacks of basing decisions on purely NPV data, and social influences on these findings.

L 268. Change word (e.g. recurring deforestation) [in reference to ‘re-deforestation’].

Response: Thank you for the suggestion, we have fixed this to include recurring deforestation (L 589).

L 281. Remove ‘from a’ [in reference to line starting with Species achieving a mean positive NPV ranged from a breakeven...].

Response: Thank you for catching this. We have rewritten this sentence (and the ones mentioned in the next two comments) to improve the clarity and create a more focused description of the results (starts at L572).

L 285. Change between to inbetween.

Response: We have now fixed these sentences (see above response).

L 310-311. Confusing [in reference to sentence starting with ‘Only with the inclusion of full upfront costs paid, do the NPV of native species or two native species...’]

Response: Yes, this was confusing and unclear. We have cleaned this section (starting on L572) to improve the clarity.

L 314. Naturally regenerated secondary forest.

Response: This is now fixed throughout the manuscript.

L 336. Change ‘a main one being’ to ‘particularly’.

Response: This is changed now (L 775).

L 346-348. Indeed. Yet I would also raise questions about a scenario where the economic incentives for moving towards restoration: what type of landscape would this be? It could be beneficial for carbon sequestration, but what about biodiversity? Doesn’t this also require studies on how biodiversity offsets could lead to different choices than carbon offsets? [In reference to sentence: ‘Additional work is needed to understand the psychology and socioeconomic drivers of decision making at the landowner scale, considering both the social and cultural aspects of cattle ranching.’]

Response: Yes, we completely agree. We have added a greater discussion around the potential effects of making decisions purely based on the economics (perhaps only planting monocultures, for example) and how certain financial supports and payments may open the door for a greater diversity of forest restoration (859-864). We write:

“Additional work is needed to understand the psychology and socioeconomic drivers of decision making at the landowner scale, considering both the social and cultural aspects of cattle ranching. Further, any forest restoration policy agreement can be strengthened with strong collaboration and input from farmers and landowners, particularly to reduce uncertainty in joining a restoration program (Shyamsundar et al., 2022).”

Additionally, we include more in the in the discussion about biodiversity (lines 889-893), but also see comments in general responses (above).

L 352. See above: since only carbon payments were studied here, it does not really say anything about biodiversity and ecosystem services. Carbon and biodiversity/ecosystem services could be correlated but this would be a secondary, unintentional side-effect. [In reference to sentence: ‘Supported by international agreements, forest restoration is being promoted as a key step to not only mitigate climate change, but also restore biodiversity and other ecosystem services (3).’]

Response: This is a good point. We have added a few sentences to the end of that paragraph to emphasize that a high carbon land use may not necessarily be a land use that maximizes biodiversity or other ecosystem services (L 889-893 and 904-908).

“However, we do not focus on other benefits that forest restoration may provide (e.g. biodiversity or other ecosystem services). For example, a land use that has high carbon may not be the same land use that maximizes multifunctionality, including biodiversity or water regulation (Messier et al., 2022). These pieces are critically important to study – to see when these benefits may be correlated or under what scenarios they may diverge.”

“Plantation carbon accrual for monocultures and mixtures including *Terminalia amazonia* are similar to those of our secondary forest but lower for other timber species and for the enrichment plantation (Table 3, Table S3). If we were to base our carbon payment on actual carbon rates, this might incentivize planting monospecific plantations of a single species, or reduce the diversity of species planted.”

L 372. \$ or US\$? Adjust accordingly throughout the manuscript.

Response: Thank you for this suggestion. We now have made it clear in the results and methods that we are working in US\$ so any reference to \$ is in that currency.

L 418. Change ‘were’ to ‘was’. [Refers to first line of Native Species Plantations section]

Response: This is now fixed (L1019).

L429-431. Unclear sentence. Above you mentioned monocultures of five species, while here you refer to 5 species mixture. The rest of the sentence is also unclear. [In reference to the last line of the Native Species Plantation section]

Response: We have rewritten the sentence to clarify the experimental design. It now reads: In addition to monoculture plots, the five target species were also planted in two-species mixture (where each of the target species is combined with the other) and five-species mixtures (where all five target species are planted together). Neither of these mixture plots are included in this paper but can be found in a previously published one (L1030-1034).

L 436. Change ‘see below’ to ‘see above’.

Response: This has now been fixed (L1039).

L 451. Include reference as manuscript in preparation. Cite manuscript year used. [In reference to García et al. (in prep)].

Response: Thank you for finding this. We have updated the references.

Reviewer #2

General Comments

Your paper addresses an important topic, forest restoration in low- and middle-income countries (LMICs) as a climate change mitigation measure. As you point out in the introduction, fundamental questions about the economics of forest restoration in LMICs remain unanswered because economists have focused so heavily on the opposite issue, deforestation. You have a valuable dataset for investigating and comparing forest restoration methods that are relevant in landscapes typical of much of the tropics, not just your study site in Panama. Your financial benefit-cost analysis is not fancy, but that's okay. The strength of your study lies in the importance of its topic, in terms of both research and application, and your data.

General Response

Thank you for this endorsement of our manuscript, reservations about our approach notwithstanding. There is a lot to unpack in the following general comments so we will address them each, paragraph at a time. Please see an extended response under Response regarding Carbon below (pages 22-26), where we dive into the carbon payment decisions and other reviewer comments related to carbon across the land uses.

I see a big problem with the study in its current form, however: the assumption of an identical carbon payment across the forest restoration methods it investigates. As a result of this assumption, the study ignores differences in the amounts of carbon that are sequestered and their timing, both of which affect carbon payments in most actual forest carbon offset programs and are critical for a science-based understanding of forest carbon sequestration. The study equates the carbon payment to the fixed annual payment the Panama Canal Authority offers to private landowners to protect mature forests. As I understand it, that payment was motivated by a concern about hydrological services, not carbon sequestration, and it refers to forest protection, not forest restoration.

These are interesting points, and we appreciate the opportunity for a deeper discussion here and in the sections that follow. The focus in this paper on flat payments is based on the work from Fenichel et al., (2019), a study that looked at economic incentives for forest-based ecosystem services in low- and middle-income countries (LMICs). A common obstacle the researchers found in LMICs is that landholders face many market imperfections, which include poor access to banking services (like savings accounts or bank borrowing). The main takeaway from their work is that a flat payment are preferable mechanisms of engaging low wealth landholders because it works around many of the existing market failures and most similarly mirrors the type of 'revenue' generation that a small landowner would receive from raising cattle, for example. In an analysis of the payment for ecosystem services (PES) program in Costa Rica (one that has been highly publicized throughout the world), Norden (2014) points out that they too use a flat payment system. It is true that the Panama Canal Authority's (ACP) program

did initially use a fixed payment to private landowners to protect mature forest to secure hydrological services and carbon loss. However, this program has now matured and led to new and different projects. In 2019, the ACP environmental division began a new program of enrichment planting in the Panama Canal Watershed in collaboration with colleagues from the Smithsonian Tropical Research Institute, where they pay the landowners \$130 per hectare each year for carbon accrual. The authors of the paper are also working on an enrichment project in Western Panama where landowners are also being paid the same amount for carbon (a flat rate of \$130 per hectare per year for enrichment planting). In eastern Panama, McGill University works with the Embera in Bayana to reforest with native species to offset carbon emissions at the university due to travel. The project initiated with the idea of re-evaluating budgets annually, but to reduce overheads and intense annual oversight by a third party, they are interested in transition to the use of a flat payment. Thus, the concept of a flat payment system *per se* is grounded in economic literature and practice.

There are two important points raised here, regarding carbon accrual: 1) the differences in the carbon sequestration across land uses and, 2) the timing (which we read to mean changes in growth/CO_{2e} over time). The reviewer rightfully (below) points out that we highlight our locally derived allometric equations and asks why we have not calculated the actual carbon between land uses. We took Reviewer 2's advice and have now calculated the Mg of CO_{2e} per hectare for each land uses. Calculations are presented here in Table RR1, and in the manuscript as Table 3 as well as further information in Table S3.

We have argued in our manuscript that we have very robust, landscape scale sampling (on infertile soils). Even with our very high number of sampling plots, we note very broad credible intervals (CI) around predicted carbon values. The implications of broad CI across the landscape are that to actually know how much carbon was accrued on a given plot of land, one must measure and calculate carbon on that piece of land, even within areas with very homogeneous, infertile soils. At the landscape scale we find, interestingly, that our monocultures of *Terminalia amazonia* had higher mean estimates of CO_{2e} per hectare at age 30 than the secondary forest (but overlapping CI). Even though all the plantation treatments that included *T. amazonia* had mean estimates and 95% credible intervals that overlapped with the secondary forests, not all species proved to be as productive. While carbon in enrichment trials fell below the low end of the CI for secondary forest, enrichment with several species nevertheless approached this lower limit.

Table RR1. CO_{2e} estimates by land use and treatment (included in Supplementary materials). Estimates (means) are in Mg CO_{2e} ha⁻¹ at age 30. Lower and upper estimates are the 95% predicted credible intervals from the Bayesian modeling.

Land use	Treatment	Estimate	Lower	Upper	
Secondary forest	Natural Regeneration	185.81	91.78	323.60	
	Ta mono	221.79	189.35	254.01	
	Dr mono	33.72	28.42	39.12	
	Ae mono	33.47	23.83	43.24	
	Pq mono	33.61	26.69	40.91	
	Tr mono	20.37	14.03	27.10	
	5 spp mixtures	132.09	108.78	155.49	
	Ae & Dr	34.14	24.56	44.76	
	Native species planting	Ae & Pq	37.90	24.72	52.23
		Ae & Ta	181.09	149.68	210.94
		Ae & Tr	24.12	17.54	31.10
		Dr & Pq	50.19	40.49	60.62
		Dr & Ta	182.41	154.31	210.05
		Dr & Tr	33.82	25.64	41.85
Pq & Ta		188.51	159.50	218.27	
Pq & Tr		21.09	15.83	26.57	
Enrichment planting	Ta & Tr	138.38	115.78	159.37	
	Bc enrichment	53.01	37.15	69.23	
	Cg enrichment	48.98	33.98	64.60	
	Dr enrichment	55.74	38.79	73.03	
	Dp enrichment	55.39	38.19	73.41	
	Ha enrichment	59.56	39.69	80.78	
	Pp enrichment	48.20	34.16	62.65	
	Ta enrichment	52.97	35.92	70.79	

The question about timing led us to create a graph that, although it showed a pattern we inherently knew, was still an eye-opener (see below). As we note in our manuscript, the choice to use a \$130 per year payment over 30 years for naturally regenerating secondary forest was based on a price per Mg CO₂e of \$18. Fig RR1 (also Fig S6) shows immense variability (CI) within a treatment for a given year across the landscape as well as a pattern of variable annual payment – which reflects the normal pattern of first accelerating and then deaccelerating stand growth – within and across treatments, were one to use the payment system for actual carbon accrued as suggested by the reviewer. For example, a naturally regenerating secondary forest would increase rapidly from time zero to a certain age and then eventually level off and/or decline (see Fig S6 in the supplementary files and also printed below Fig RR1). Very significant effort would be required to annually measure carbon accrual in order to set the value of the annual payments and, as we elaborate on elsewhere, would do nothing to smooth out the payments to the landowner. Indeed, they may incentivize a perverse outcome – early clearing of the land.

There is also the real possibility of less carbon accrual over time than the estimates we have for our study site. However, Hall et al., (2022) also used data from the study site to include coarse roots (from Sinacore et al., 2017), carbon in the top 20 cm of the mineral soils (from Neumann-Cosel et al., 2011), lianas (Lai et al., 2017) and from coarse woody debris (Gora et al., 2019). Hall et al. (2022) note that taken together, these other compartments add an additional 60% of carbon to the aboveground tree carbon in the region. This buffers against not only change in carbon accrual, but also can be considered toward the risk of leakage (typically estimated around 10%).

We have updated our paper to include the carbon sequestered on the secondary forests (Hall et al 2022), native species plantations, and enrichment plantations, the results of which are now included in the in the main body of the paper (Table 3) and supplementary materials (Table S3).

Fig. RR1 (S6) Annual carbon dioxide equivalent (CO₂e) payments (\$US) for focal land uses. Change in CO₂e payments based on the growth of CO₂e and the price per CO₂e over thirty years. Solid black lines represent the mean estimate of the payment, and the gray shading represents the 95% credible interval around the estimate. The dashed blue line represents the \$130 flat annual payment. SFD – secondary forest, Dr mono – *Dalbergia retusa* in monocultures, Ta mono – *Terminalia amazonia* in monocultures, Dr & Teak – *D. retusa* and teak enrichment plantings, Ta & teak – *T. amazonia* and teak enrichment plantings.

The study would be much more valuable if it investigated the financial incentives for forest restoration generated by payments based on actual amounts of carbon sequestered and carbon prices that are pertinent to current and prospective forest management. By the latter, I mean such prices as typical ones for actual afforestation/reforestation projects (\$5-10/ton CO₂e; see various issues of the World Bank’s State and Trends of Carbon Pricing reports and Ecosystem Marketplace’s State of the Voluntary Carbon Markets reports); prices needed to hit the Paris Accord targets (many studies estimate these); and the social cost of carbon (see Rennert et al., Comprehensive evidence implies a higher social cost of CO₂, Nature, 2022).

While we appreciate the reviewer’s perspective, in our study we chose to evaluate scenarios with a flat payment system. As we write above and below, this system is grounded in the economic literature and practice. It has advantages of smoothing payments to rural landowners lacking wealth and access to financial systems and is supported by a significant body of literature pertaining to equity and social justice (please see our extended discussion on this below and how we address this in the manuscript).

The level of our payment is linked to carbon accrual in secondary forest as we apply a flat payment for forest protection: US\$ 18 per Mg CO_{2e} in aboveground tree biomass. This is more than what the reviewer cites as what the World Bank and others suggest and apparently pay. However, it is significantly less than what is paid on the European Union's Emissions Trading Scheme (ETS), which has exceeded 100 euros per Mg CO_{2e}. It is also significantly less than the social cost of CO_{2e} of US \$185 per Mg CO_{2e}, which is mentioned in the article by Rennert et al. 2022 to which the reviewer pointed us. Finally, it is less than half the price used by Sandoval et al. (2022 – see below on discussion of cattle) for avoided methane emissions due to silvopasture system in Colombia (US\$ 45.25 per Mg CO_{2e}) and similar to the US\$ 15.1 per Mg CO_{2e} for soil organic carbon sequestered under improved pasture management in the Brazilian Amazon cited by de Oliveira Silva et al., (2017). We note the leap of faith one must have in extrapolating these last two measures to markets without very significant and costly (and surely impossible to measure at scale for Sandoval et al. 2022) monitoring. Thus, we feel our own price converted to Mg CO_{2e} is defensible.

You argue that a uniform, fixed payment is a practical means of scaling up carbon financing for forest restoration, but I am not convinced. There is great policy concern about the integrity of forest carbon offset projects. See, for example, news stories over the past 1-2 years about Lyme Timber CEO Jim Hourdequin; fraud allegations in Australia; and The Guardian's recent expose on REDD+ projects (<https://www.theguardian.com/environment/2023/jan/18/revealed-forest-carbon-offsets-biggest-provider-worthless-verra-aoe>). Governments, conservation organizations, and some companies are demanding better, not cruder, forest carbon payment programs. More accurate measurement of sequestered carbon is increasingly possible thanks to advances in remote sensing, including sub-meter satellite data and remarkable imagery from drone- and aircraft-mounted LiDAR. Companies on the cutting edge of the forest carbon sequestration business, such as Mombak in Brazil, are demonstrating the feasibility of bringing rigorously designed forest carbon projects to market and the willingness of carbon-emitting companies to pay a premium for them.

We agree that there have been very problematic cases of fraud concerning REDD+ projects and the media has been able to expose many of these cases (as Reviewer 2 mentions). Although our study focuses on restoration (rather than REDD+ forest conservation), we share concerns about the recent failures of wide spread planting efforts across the globe (e.g., Pearce 2022). With respect to Mr. Hourdequin and Lyme Timber, from what we have read, the issue does not seem to be an issue of fraud or lack of monitoring. Rather (based on the Bloomberg News article - <https://www.bnnbloomberg.ca/this-timber-company-sold-millions-of-dollars-of-useless-carbon-offsets-1.1738975>), Mr. Hourdequin comes forward to expose what he sees as a broken system that allows companies to pay for carbon credits from avoided deforestation or land/carbon degradation (e.g., through forestry or logging of mature forest) that is already protected. The system seems to allow for this, and this certainly seems to go against the spirit of avoiding emissions to fight climate change.

We appreciate and acknowledge improved monitoring with use of LiDAR. Here we wish to add that this is part of what are considered Transaction Costs (TAC). Reviewer 3 has pointed us to

literature regarding TAC which we now include as an extended discussion below and reference in the manuscript (L 942-956). We note that as LiDAR and other monitoring costs go down, this will benefit all methods of carbon payments by reducing this aspect of TAC.

You appear to have sufficient data to estimate annual amounts of carbon sequestered by the different forest restoration methods you investigate. You refer to annual tree inventories, locally derived allometric equations, etc. Why not take full advantage of these data, and link the resulting annual estimates of sequestered carbon to pertinent carbon prices? The paper currently highlights the effects of varying assumptions about the discount rate on carbon-based financial incentives for forest restoration. Although discount rates have a major impact on financial analyses, the effects of varying assumptions about carbon prices—especially their impacts on the relative attractiveness of the different restoration methods—would be more interesting and more valuable from a policy perspective.

As noted above, we see the value that the reviewer has mentioned in using the rich datasets we have to include the actual carbon accrued. We report on and discuss this above and below as well as in the manuscript itself (Table 3 and Table S3 Fig S6). We also understand the interest in linking the actual carbon accrued to the carbon payment to the landowner, but as explained and justified above and below, have chosen for the main body of the manuscript to focus on the flat payment as used by Fenichel et al. (2019), the authors from whom we derive our economic incentive models and justification thereof. As stated above, the annual flat payment system is based on real data and is already used widely across Panama and Costa Rica and is being adopted by other programs in LMICs (see above and below).

The flat payment is also based on the social, equity, and justice literature specifically studying forest restoration and payment for ecosystem services programs. As cited above, there is uncertainty in any new reforestation project on the amount of carbon that will be accumulating overtime. By focusing on payment based on measured carbon growth, we believe that the burden of uncertainty falls on the landowner (and away from the person paying for the carbon). Taking the risk off the landowner is particularly important in low- and middle- income countries and for small landowners who are often asked to take on the burden of restoration failure. Pascual et al., (2014) have made the case that ‘efficiency-led’ payment for services runs the risk of failing as the result of applying a single-objective tool to a complex socio-ecological situation. Additionally, choice-experiments and case studies have shown that flexible, long-term contracts with guaranteed payments are the best way to engage low-wealth individuals in PES programs (Lliso et al., 2021; Pascual et al., 2014). Because of these reasons, we consider the evaluation of flat carbon payments as a mechanism to incentivize different types of restoration as particular important and urgent. Please see our more detailed treatment of this below.

Line comments and responses

(please pay special attention to the one about p. 5)

p. 3, line 86: “were” should be “where”; “than those” should be “than on those.” These are rare typos in a very clean manuscript; thank you for preparing it so carefully.

Response: Thank you for catching this. We have now fixed this (L 93).

p. 4, line 94: You refer to the first restoration method as “secondary forests.” Secondary forests are an outcome, not a method. The restoration method that generates them is natural regeneration. I recommend replacing “secondary forests” with “natural regeneration” whenever possible in the paper, including in Fig. 1.

Response: This is a great point, and we appreciate the attention to detail. Throughout the text, we have now included “natural regeneration” or “natural regenerating” qualifiers to signify the method (when appropriate) and kept the outcome (secondary forests) when necessary.

p. 4, line 94: On the vulnerability of secondary forests in the Neotropics, you might also cite Schwartz et al., “Reversals of reforestation across Latin America limit climate mitigation potential of tropical forests” (Frontiers in Forests and Global Change, 2020) and Sloan, “Reforestation reversals and forest transitions” (Land Use Policy, 2022). Holl et al., “Redefining ‘abandoned’ agricultural land in the context of reforestation” (Frontiers in Forests and Global Change, 2022) might also be relevant.

Response: Thank you for the literature suggestions. We have now included the additional citations in L99-133).

p. 4, line 102: You refer to the second restoration method as “tree planting.” This is confusing, because your third method, enrichment planting, is also a form of tree planting. In a few places, you use “plantation” instead of “tree planting.” “Plantation” is more clear than “tree planting.” I recommend replacing “tree planting” with “plantation,” including in Fig. 1.

Response: Thank you for the suggestion. We have revised the paper to ensure there is greater clarity. We use native plantations to refer to the monocultures and mixtures of our native tree plantation system and enrichment plantings to refer to the area where native species are being planted into the teak plantations.

p. 5: Somewhere around this point in the paper, you need to explain the initial site conditions for the three restoration methods. Based on information in the Supplementary Materials, I infer that the initial conditions were pastures in all cases, with grazing ceasing and restoration starting in 2008-9. Is that correct? Relatedly, it would be helpful to explain that Year 0 for calculation of the NPVs for all the methods corresponds to 2008-9 and that the NPVs for the three restoration methods do not account for the opportunity cost of the land (i.e., the NPV of pasturing)—right?

Response: Thank you for the suggestion. In the introduction we have now included that the sites were originally abandoned pastures (L218). In the methods (L1103-1106) we have added more details about the restorations starting in 2008 (except for the enrichment that began in 2016). We also explain that year 0 is based in 2008 and the NPV calculations are based on the restoration initiation as year 0. The calculation does not include the opportunity cost of land here.

p. 6, line 162: I suspect that some readers might quibble with your choice of a 7% discount rate. I agree that it’s a suitable rate; I’ve used it in my own work, with the justification that it equals the mean of the real interest rate in LMICs. You might add a sentence explaining that you use a

private (financial) discount rate instead of a social (economic) because you are focusing on private land-management decisions.

Response: We really appreciate this suggestion and the nuance in wording so we have included your phrasing into the beginning of the results section (L295-302).

p. 7, Fig. 1: I like the layout of the figure, but the figure would be more informative if it displayed box plots instead of bars. Readers will be interested in the distribution of the NPVs, not just the range. Box plots would provide standard distributional information: medians, 25th and 75th percentiles, minima, maxima. The boxes would be small, but I believe they would be readable.

Response: Thank you for this suggestion. We originally tried a few different methods of showing the ‘big picture’ results for Fig 1 and found that the boxplot figures ended up being difficult to read based on the data we are showing. To ensure that the figures are as informative as possible, we added an open circle to Fig 1 to represent the mean of NPV at 7%. For Figs 2-4, we included the mean and credible intervals around the mean (90% and 95%) to give as much information as possible.

p. 7, Fig. 1: Why does the figure include Silvopasture, given that there are no NPVs for it? On the other hand, why not add Silvopasture as a fourth restoration method and include carbon payments for it?

Response: We realize that the submitted Fig 1 made it seem that silvopasture did not have an NPV. We have now fixed the figure so that the NPV for silvopasture is clear in Fig 1 (top left panel). We did not calculate the NPV for pasture or silvopasture in the half and full financial support levels, nor for the carbon payments, so have now added a gray “NA” to make it obvious that those are intentionally excluded from the figure. We hope this makes the data more readable.

pp. 8-9: Fig. 2-4 are not essential. Move them to the Supplementary Materials, and replace them with figures showing the implications of varying the carbon price.

Response: While we appreciate your point of view and suggestion, we also have received some positive feedback on the figures. In the end, we have decided to include them in the revision. We appreciate your idea about carbon estimates and price changes so have included a condensed version of the CO_{2e} for land uses in Table 3 of the main manuscript and a full version in the supplementary materials (Table S3), as well as a figure on CO_{2e} payment changes for the focal land uses (Fig. S6). We hope the estimates and information about pricing will easily enable those interested in adopting a variable payment program to be able to do so.

p. 9, line 237: You might cite Shyamsundar et al., “Scaling smallholder tree cover restoration across the tropics” (Global Environmental Change, 2022), for additional support on the importance of including smallholders in restoration initiatives.

Response: Thank you so much for the citation. It is an important paper and have included it in our discussion of smallholders and restoration initiatives, particularly the part about the importance of including landowners and markers in the development of programs, which can help reduce uncertainty and increase buy-in (starting on L 502)

p. 11, line 280: If restoration systems are financially viable without carbon payments, then the carbon sequestration they provide is not additional and thus would not qualify for carbon payments, would it? Adding some discussion on the implications of your findings for the circumstances under which the different restoration methods deliver incremental sequestration would enrich the paper.

Response: This is a good point. It is possible that in some programs, a restoration system that is financially viable without carbon payments would not qualify for participation in a carbon program. However, the fact is that precious little of the land held by small holders is naturally regenerating or being put into plantations and left that way. Elsewhere and in the manuscript, we discuss and reference how lack of clear land title can lead to re-deforestation to maintain one's claim on the land. With respect to plantations, small holders have capital access and income smoothing challenges that are the basis of a large part of the manuscript. Thus, whether or not something might be financially viable without carbon payments in theory confronts a harsh reality for many small holders. In addition, and in our system, *D. retusa* monocultures with full financial support is the only land use and treatment that has positive NPVs at a 7% interest rate. Based on the 'worst' case scenarios in tree growth, timber prices, and depending on the interest rate, there is a risk that *D. retusa* monocultures would not be financially viable (Fig 3). For that reason, we have modeled the NPV for *D. retusa* monocultures with the carbon payment scenarios. Further, we suspect there may be cases where forest restoration is of interest, but the land is not near a forested system, such that planting may be the only option. Of course, there could be options to plant a mix of timber and non-timber trees (some of which may never be cut for timber revenue).

p. 16, line 430: Including the results for mixed-species plantations would strengthen the paper. Why haven't you included them?

Response: We focused on the monocultures for the main body of this paper (although additional information is available in the supplementary materials, which includes mixtures of different native species). A more detailed discussion of the mixed-species plantations can be found in Sinacore et al. 2022 where specific combinations of mixtures are tested.

p. 16, lines 436-438: This sentence leaves me unclear as to how Year 0 is defined for enrichment planting. Is it defined as 2008 or 2016? For an apple-to-apples comparison to the other restoration methods, it should be defined as 2008, and the NPV for enrichment planting should be defined as the difference between the NPV for a teak plantation with enrichment planting and the NPV for one without it.

Response: Thank you for catching this. We have improved the text for clarity (L1103-1106) Year 0 for the enrichment planting is defined as 2008 (for direct comparison with the native species plantations).

p. 17, lines 483-4: I assume you meant to type "real rate of interest," not "nominal rate of interest," but if you used a nominal rate of interest, did you project the timber prices in Table 1 (and all other project revenues and expenses) forward in time with an inflation rate? If so, what rate did you use?

Response: Thank you for finding this error – we meant 'real rate of interest' and have now fixed it in the text (L1096).

Reviewer #3

General comments

This is a very interesting and informative study of the effect of carbon payments and cost share-arrangements on the financial performance of different forest restoration interventions in a particular location in the tropics (Panama), under various discount rates, and of how the financial performance of these restoration options compares to that of conventional and silvopasture alternatives.

The paper is timely and important given the urgency to deploy nature-based solutions in general and forest conservation and restoration in particular, and the methods seem sound. It also is well-written and concise, and contributes to a relatively sparse literature on an important issue: how to make forest restoration financially attractive for tropical and subtropical land operators.

The authors find that some interventions can have substantial NPV, even without carbon (C) payments and cost sharing of initial and maintenance costs, but this finding is sensitive to species choice and discount rate. In line with intuition, individually and especially in combination, C payments and cost sharing improve the financial attractiveness of restoration options, but relatively more so for some than for others.

I strongly support the publication of this manuscript. However, there are a few issues that need to be addressed prior to publication.

General Response

Thank you for taking the time to read our manuscript, your support of the publication, and for offering suggestions to improve and strengthen the paper. We have incorporated your suggestions into the revision and have a detailed response to the suggestions below.

Line comments and responses

1) 245-247: The authors state that “Our results demonstrate that in all cases financial support to cash strapped rural landowners will be necessary for them to assume the financial risk and ensure positive economic returns.”

I am not sure that your results alone support this very strong statement. The native species plantings have fairly substantial NPVs, so potentially, even cash-strapped farmers might be able to finance these activities in the absence of subsidies if credit access is available at sufficiently low interest rates. This cost-of-credit would need to be included in the NPV calculations to answer my question. It may well be that sufficiently low rates that would results in positive NPVs are not available; that credit access for the type of landowner relevant to tour analysis is

limited or that credit; or that landowners would be unwilling to assume the credits for the lower NPV that would result. But none of these factors are discussed in your analysis. I would suggest making your statement less categorical, or introducing the additional information that would support such a conclusion.

To my point above: line 44 states that “We show some, but not all options are economically viable”; and in 279-281, that “We found that the native species plantings showed the highest financial profitability (Fig. 1a) under most economic scenarios even without carbon payment or financial support structures (Fig. 4, Fig. S5).” Both of these are inconsistent with the statement in 245-247 that “in all cases financial support to cash strapped rural landowners will be necessary for them to assume the financial risk and ensure positive economic returns.”

Response: Thank you for catching this inconsistency. We have rewritten to the first part, which was intended to refer to evidence in the literature that financial support to rural landowners may be necessary to transition away from cattle revenue streams (Fenichel et al., 2019; Vincent et al., 2021) (L512). One of the difficulties for many landowners in LMICs is that they do not have access to banking and loans, even if the interest rates are low. We have reworded the discussion to emphasize the difference between the NPV results (independent from financial status of a landowner) and the evidence from the literature suggesting financial support mechanisms would be necessary in LMICs.

2) Discussion: I am glad to see the authors mention the non-financial drivers of land use decisions that affect land manager behavior.

Response: Yes, this is such an important piece to land-use decisions that cannot be reflected in the analyses of NPV that we have included in this paper.

3) 371-372: “There is evidence that carbon prices must be higher than \$100 ton CO_{2e} to incentivize large-scale forest restoration in the tropics”, citing Busch et al (2019). I do not think that this characterization of Busch et al.’s findings is correct. Busch et al. find that

“A carbon price of US\$20 tCO₂⁻¹ would incentivize tropical land users to increase reforestation by 31.8 Mha (8.2%) to 419.6 Mha ... from 2020 to 2050 relative to a BAU scenario (Supplementary Fig. 6). ... A carbon price of US\$50 tCO₂⁻¹ would increase reforestation by 84.1 Mha (21.7%) to 471.9 Mha...”

As their Fig. 1 shows, predicted sequestration from reforestation increases nearly linearly with the C payment level, so total additional reforestation extent due to C pricing at US\$ 100 tCO₂⁻¹ is nearly twice that at US\$ 50 tCO₂⁻¹, or around 160 M ha or 1.6 M km²—approximately the size of Mongolia or Iran. I doubt reforestation at this scale can be considered not large-scale. It is true that this level of reforestation is still far below the identified biophysical maximum potential, but it certainly nevertheless is large-scale. Please reword your statement accordingly.

Response: Thank you for correcting us here. We agree that reforesting the size of Mongolia or Iran would be considered large scale. We have completely reworked this section of the conclusion to focus on the potential outcomes of a flat payment vs a market-efficient C payment and included a brief mention of transaction costs associated with reforestation. The main part we reworked starts in the “Challenges and future directions” section (L895). We

focus on the idea of flat payments and market efficiency, as well as the social constraints around land use decisions.

4) Lines 501-503 state that “we apply a carbon payment of \$130 ha⁻¹ per year whereby the landowner cedes the rights to the sequestered carbon to the person or group making the payment.” Then, in lines 507-509, it states that landowners receive “an additional US\$ 130 ha⁻¹ for years 5-30” in the estimation of a second NPV. This is a bit confusing. It suggests to me that landowners holding secondary forests, or implementing native species plantations or enrichment plantings are paid US\$ 130 ha⁻¹ in years 1-4, and then US\$ 260 ha⁻¹ in years 5-30. If so, what is the rationale for this higher payment in years 5-30? On the other hand, in the remainder of the manuscript, I only see mention of US\$ 130 ha⁻¹ yr⁻¹. As a result, I am unsure what the actual payment levels are. Please clarify the description of payment levels.

Response: Thank you for bringing this to our attention; how the sentence was originally worded is a bit confusing, so we have updated it (L1103-1106). What we intended to say is that in addition to all, some, or none of the upfront costs being subsidized, the landowner would receive \$130 ha⁻¹ year for years 5-30. Regardless of land use, the landowner would only receive \$130 ha⁻¹ per year for years 5-30 (not \$260 if they had certain land uses).

5) I did not see any mention of C payments for silvopasture. If this is correct, could you explain why tree cover in silvopasture is not receiving any C payments? There certainly are PES programs in tropical countries that support silvopasture establishment, so it would be helpful to understand why no C payments for this land use were considered.

Response: This is a fair question. We have reviewed the literature for this response below. We explain that silvopasture means different things to different people. As we state, in our case it mostly means improved pasture and restricting access to streams. The economics of each system differs and is constrained by the soil fertility as this constrains the options for forage crops (see in extended response regarding cattle, pages 26-31). We focus on infertile soils, the most challenging for silvopasture systems and as near as we can tell almost entirely ignored in the silvopasture literature (but not in the improved pasture literature from Brazil). In the literature carbon payments for silvopasture and improved pasture and systems (fattening, etc.) are mostly given for reduced greenhouse gas emissions and increases in soil organic carbon. Monitoring of greenhouse gas reductions is complicated. Economic modeling of this uses very localized and short-term studies (imagine putting a cow in a polyethylene tube for 3 days and monitoring emissions) that can't be used at an operational scale and for which even the extrapolation is problematic. Soil carbon monitoring is more straight forward but still accrues slowly.

We have very solid data on our restoration treatments. We do not have, nor have we seen in our review, robust data (multiple sites across a landscape and over time that includes real variability) that can be extrapolated to model carbon benefits through silvopasture. As we state in our more extensive review below, we were very conservative and generous with our NPV calculations of traditional and silvopasture systems such that any carbon payments for these treatments would be marginal. We now explain our decision in the Methods section of our manuscript, including more detail on our own silvopasture system.

6) Fig 1: Very informative figure. However, I do not understand how traditional pasture can have a positive NPV range in the case with no financial support or C payments, but a zero NPV in the C payments scenario (as indicated by no visible marker in the No C payments panel -- the lower left panel in the figure)? Presumably, traditional pastures do not receive any C payments, so how can it be that C payments affect the NPV of traditional pasture?

Response: Thank you for catching this. We have updated Fig 1 in a few ways to make this clearer: (1) We added an open circle to mark the mean NPV for each land use; (2) We added NA's where the NPV was not calculated. We found that where we work silvopasture, in practice, does not look so different from pasture in terms of tree cover. For that reason, we have not included a carbon payment scenario for either of those land uses.

7) The authors approach of making C payments independent of actual biomass accumulation is an interesting one, and one that would reduce uncertainty for landowners. Presumably, this also might reduce the transaction costs (TAC) associated with C payments, as the monitoring of C accumulation may be less involved than in performance-based payments on the basis of "actual" C accumulation.

This issue of high TAC of tropical forest C offsets (e.g., Pearson et al. 2014; Rendón Thompson et al. 2013; van Kooten et al. 2002) is not discussed. These costs are a function of market access and market rules (search and contract costs, certification, mandatory inventory reports, mandatory C buffer pools etc.). If the C credits generated by the program studied in this paper do not get transacted on the global voluntary C markets but rather accrue to some local entity (say, the Panama Canal authority), then landowners presumably would face much lower TAC associated from this C scheme than they would on global markets. Still, these costs would not be zero. It would be good to see the topic of TAC at least briefly mentioned in the paper.

On a side note: If the local entity to which the C credits accrue were to seek to transact these credits on a C market (to recoup some of the program's costs), it would then of course need to comply with the requirements of that market, and so this entity would be the actor incurring these high TAC. I realize this would not affect the financial viability of the analyzed reforestation options for participating landowners – the topic of this paper-- but it would affect the overall cost of the program.

Response: Thank you for bringing up the topic of transaction costs and leading us to literature focused specifically on this topic. We do think that the flat payment would help reduce transaction costs and also have read literature on the social, equity, and justice sides of PES designs, finding that a flat payment is what would be most accepted by landowners in addition to reducing TAC; please see extended discussions below on carbon payments and transaction costs. Below we follow Pearson et al. (2014) to discuss how insurance costs, the largest transaction cost of a project, are accounted for by carbon stocks beyond those of aboveground tree biomass (AGB) and thus represent our insurance costs. We agree with the reviewer that monitoring costs (found by Pearson et al., (2014) as representing approximately 25% of the TAC in their example(s) of Latin American reforestation projects) should be reduced. Norden (2014) and Fenichel et al. (2019) assume TAC are included in the payments calculated for landowners

and by our estimate using Norden (2014) would be between US \$2.5 and US \$ 5 per Mg CO_{2e} (see below).

In reference to tying the carbon payment to the actual accrual, we have now included the CO_{2e} of each land use in a new table (Table 3) in the manuscript with the land uses presented in the main body of the manuscript. In the supplementary (Table S3) we present the CO_{2e} of each treatment within each land use. Interesting, any native species plantation combination with *Terminalia amazonia* has CO_{2e} at age 30 equivalent (if not more than the secondary forest). We discuss the fact that carbon markets that focuses on an average or mean carbon value ignore the variability in the data as evidenced in broad (and overlapping for many treatments) credible intervals (again, please see elsewhere in our response for more details on this).

Extended carbon payment discussion

All three reviewers raised concerns regarding our treatment of carbon sequestration and/or payments in our analyses. We address these concerns in greater detail here, as well as briefly in the response to each comment provided above. One reviewer had a particular concern about our decision to use a flat payment instead of linking the payment to the actual amount of carbon sequestered on each land use. While we acknowledge this concern, we would like to highlight that our choice of a flat payment structure was based on several considerations.

Flat payments

First, the concept of implementing a **flat payment structure in our study was inspired by the research conducted by Fenichel et al., (2019)**, which focused on incentive systems for forest-based services in the Panama Canal Watershed as an example but is meant to advance theory and practice more generally. In this particular region, a significant portion of farms (63%) are smaller than 1 hectare, and many farmers lack access to savings accounts or bank borrowing options (Adamowicz et al., 2019). Transitioning from cattle ranching to establishing planted forests involves substantial upfront costs and lacks the consumption-smoothing capabilities provided by cattle ranching. Therefore, a critical question emerged regarding the types of incentives required to support low-income farmers in transitioning to a forest-based landscape. **One of the key findings from Fenichel et al.'s paper was that a flat payment has the most significant impact in encouraging participation in forest restoration projects** (Fenichel et al., 2019, Fig 3). The results also indicated that subsidized loans and the availability of savings accounts with higher rates of return would be crucial in incentivizing land use change. These factors had an even greater effect on participation than a lump sum payment upfront, primarily due to the limited access to interest-bearing savings accounts among small landowners in the study area. Furthermore, the implementation of a flat payment system offers specific advantages, particularly when engaging low-wealth landholders and prioritizing the expansion of reforested areas. **Flat payments help address the imperfections present in financial service markets and the unequal access to such services among different individuals.**

Second, the use of a **flat payment reduces administrative complexities and transaction costs.** Implementing a payment system that accurately reflects the carbon sequestration on each

restoration type would require precise monitoring and verification mechanisms, which can be resource-intensive and time-consuming. In contrast, a flat payment approach provides a straightforward and streamlined method of compensation (see more in 'Transaction Cost' section of our response for more detail, below). **A flat payment system offers a level of predictability and stability for landowners or project participants.** By providing a fixed payment amount annually, it ensures a reliable income stream that incentivizes initial and continued participation in forest restoration and ecosystem service projects. This stability can be particularly crucial in low- and middle- income countries and long-term projects, where or market fluctuations may pose challenges to a variable payment structure. Arguably, a flat carbon payment does not distinguish between the carbon sequestering potential of different land uses. While there may be variations based on factors such as species composition or site conditions, some restoration options may not vary significantly (please see our analyses above in response to Reviewer 2). In such cases, a flat payment can provide an equitable and practical approach to compensation while also encouraging a greater diversity of forest restoration methods. For our **third reason, we use the data from our own study to show the carbon sequestering potential of different land uses and the CO₂e annual price based on growth.**

While we recognize the concerns raised by the reviewer, we believe that our decision to opt for a flat payment structure is justified. We also are mindful of the evolving discourse surrounding payment structures in carbon sequestration projects and acknowledge there are alternative approaches to designing PES initiatives. We further explain our justification for a flat carbon payment in the paragraphs that follow, focusing on the benefits to the landowner, the actual carbon sequestration across our study sites, framing this in the context of social equity and the latest research on socially oriented PES projects.

Social and equity aspects

Recognizing the reasons behind using a flat payment mechanism also highlights the importance of considering the human and social aspects of restoration. In the last few decades there has been growing criticism that restoration and PES systems have not sufficiently focused on these crucial aspects. To fully realize the potential of restoration in achieving both ecological and social goals, a necessary shift towards a people-centered focus is required (Lliso et al., 2021).

In support of this perspective, Elias et al., (2022) conducted a comprehensive review proposing actionable ways to prioritize the human and social dimensions of ecosystem restoration. Their recommendations aim to create fairer and more sustainable initiatives by addressing various stages of the restoration process. These stages encompass stakeholder recognition, community reengagement, socio-historical contexts, resource tenure for marginalized groups, equity promotion, generation of multiple benefits, **equitable distribution of costs and benefits**, utilization of diverse evidence and knowledge, questioning prevailing narratives, and practicing inclusive monitoring, evaluation, and learning (Elias et al., 2022).

Among these proposed recommendations, we specifically want to focus on the concept of distributing costs, benefits, and risks equitably. Restoration initiatives entail both environmental benefits, such as carbon sequestration and enhanced biodiversity, as well as social benefits, including employment opportunities and land right security. However,

restoration also involves costs, whether monetary, opportunity-related (such as foregone land use or income), or labor-related (such as the physical work involved with planting trees, fencing an area, maintaining the work). Additionally, risks associated with restoration can manifest as social tensions, power dynamics, and potential displacement of local land users as land use and values change. Different groups, particularly women and men, have different perspectives on and priorities regarding costs, risks, and benefits. **Unfortunately, equitable distribution of costs and benefits remains rare (Pham et al., 2016), often leading to overlooked and inadequately mitigated risks for marginalized groups (Covelli Metcalf et al., 2015).** Furthermore, the distribution of costs and benefits can vary over time, particularly as some restoration processes require long-term investments or entail opportunity costs with delayed returns. In this regard, **a flat payment structure can help alleviate the costs and risks borne by landowners participating in a restoration programs.** While it does place a greater burden of risk on the entity paying for carbon, the flat payment method better addresses the costs and risks generally assumed by the participants of the PES program.

It is worth noting that many PES approaches, including the initial model established for REDD+, are driven by economic efficiency principles and operate within idealized market-like conservation mechanisms. These approaches aim to incentivize voluntary, economically efficient conservation by selling/trading ecosystem services, such as carbon (Engel et al., 2008). However, the evaluation of conservation and restoration outcomes in terms of efficiency is often challenging due to the complexities involved in assessing ecological and economic factors. Consequently, cost effectiveness is often used as a proxy for efficiency, with managers seeking to maximize ecological outcomes within a limited budget (Pascual et al., 2014). This approach runs the risk of prioritizing large landowners as recipients of PES, potentially excluding smallholders (Kroeger, 2013). Yet including smallholders in low- and middle-income countries requires additional financial incentives outlined in Fenichel et al., (2019) and described above.

Nonetheless, this perspective neglects the multifaceted nature of PES programs, that are embedded within complex social and political systems (Hirsch et al., 2010). It is not surprising that, in response to public pressure and real-world considerations, PES initiatives have increasingly embraced equity dimensions in their design and implementation. For instance, the initial driving force behind REDD+ policies was low-cost carbon emissions mitigation, but these programs have evolved to incorporate social safeguards (Pascual et al., 2014). Notably, this shift toward equity-focused PES approaches has been observed in countries such as Costa Rica, Colombia, and Ecuador, neighbors of where we work in Panama.

In these countries, there has been a growing reconfiguration of the traditionally efficiency-focused PES design, which primarily focused on specific ecosystem benefits, such as carbon sequestration. Instead, a more equity-conscious approach is gaining traction, recognizing the broader range of benefits that can result from restoration and reforestation efforts, such as water regulation, improved livelihoods, etc. **A flat payment approach acknowledges the potential for a piece of land to have lower carbon sequestration performance but perhaps other unaccounted for benefits, such as water filtration or even social and cultural values.**

Addressing the socio-historical context of restoration in the tropics and the Global South is of paramount importance. Market-based approaches have often neglected historical inequities that encompass issues such as land access, land ownership disparities, and the overall productive capacity of the land. A comprehensive review incorporating perspectives from scholars, practitioners, and governments in both the Global North and Global South revealed significant concerns among respondents from the Global South: the need to prioritize co-benefits over trade-offs (Sovacool, 2023). Specifically, it highlighted the challenge of integrating society co-benefits into the business model of PES to prevent participants from planting whatever will get them the most credits. For instance, our study demonstrates that a monoculture plantation of *T. amazonia* would generate the most credits to a landowner, potentially incentivizing extensive establishment of *T. amazonia* monocultures across large areas. However, such a narrow focus on maximizing credits could ultimately result in reduced benefits that other types of land uses may provide.

Furthermore, one of the biggest concerns from Global South countries revolves around the vulnerability of single-commodity economies, fluctuating prices, and an overemphasis on the economy solely revolving around crediting projects. As mentioned earlier, injustices stemming from unfair distribution of risks or benefits and disproportionate accountability for actions and outcomes can unreasonably impact communities participating in PES programs (Davoudi and Brooks, 2014). Notably, **farmers expressed significant apprehension about the volatility of pricing**. For instance, a respondent highlighted the hardships faced by landowners with limited resources when they initially enter a program based on a carbon price of US\$10 (for instance), only to witness a drastic drop to US\$1 (as an illustration) as the program progresses (Sovacool, 2023, Table 3). The rationale for reducing the pricing is from areas not accumulating as much carbon as initially projected at the start of the project.

Carbon sequestration rates

In light of varying carbon sequestration rates across different land types, such as forests, tree plantations, and enrichment plantations, there is a compelling argument to consider using a carbon payment that correspond to the actual amount of carbon sequestered. Considering the suggestion of Reviewer 2, we calculated the CO_{2e} per hectare at age 30 for each of the treatments in the study (RR Table 1; please see above). Notably, and probably unsurprisingly, secondary forests had one of the highest CO_{2e} ha⁻¹ values (186 Mg ha⁻¹). Similarly, **the traditional plantation treatments involving *T. amazonia* yielded estimates similar to those of the secondary forests, ranging from 183-186 Mg ha⁻¹**. However, it is important to note that not all mean CO_{2e} estimates aligned with those of the secondary forest or *T. amazonia* plantations. Two crucial pieces of information are important to consider here. First, the credible intervals around the estimate show considerable overlap among many of the treatments. **Second, the credible interval range is quite wide, for example, 100-250 Mg CO_{2e} ha⁻¹ for secondary forest**. This is despite having some of the most robust data on secondary forests in the tropics. These finding underscore the significant variability of secondary forest growth across a landscape, as the data are based on annual measurements from 108 sites across a 30-year chronosequence. Although technology, particularly LiDAR, is advancing and improving carbon estimates, the development of the algorithm to predict carbon in Panama was based partly on the data obtained directly from the Agua Salud secondary forest network (Asner et al., 2013).

Implementing carbon accounting on the ground in a reforestation project would undoubtedly require significant effort, and it is conceivable that reducing the estimation error beyond what we have achieved with our own calculations, based on our experimental setup, would still prove challenging. However, it is important to acknowledge the value of sharing carbon estimates across different treatments. To facilitate this, we have included a comprehensive table in the supplementary materials, along with a condensed version within the methods section of the manuscript.

In summary, our decision to adopt a flat payment carbon system in this study was influenced by the research conducted by Fenichel et al., (2019), focusing on the Panama Canal Watershed. Additionally, our carbon estimates, coupled with the insights gained from other authors' work on social equity and justice, guided our approach.

Extended cattle discussion

All three reviewers had at least one comment pertaining to our analysis and treatment of traditional cattle and silvopasture treatments. Here we address these comments collectively, expanding on the explanations provided above under each specific concern. As the reviewers understand, our manuscript focusses on a financial analysis of landscape-scale treatments of different forest restoration techniques in Panama, with relevance for the neotropics and beyond. The strength and novelty of our analysis lie in several key aspects: 1) the fact that we use large and robust data sets, 2) we work on infertile soils, soils that are often overlooked in studies of plantation forestry in the tropics despite their widespread occurrence, 3) we analyze different restoration methods, 4) we have temporal data (remeasurements) over a reasonably long time period and are not based on chronosequence studies.

Cattle densities

We do not have the same quality of data for our two cattle treatments. Our traditional pasture and silvopasture treatments are long term but limited to a single farm of each treatment. In our initial submission, we decided to err on the side of caution and increased our traditional pasture and silvopasture treatments to a cattle density of 1 head per ha as that is what local practitioners claimed as their density, even though we have not observed such densities in our region. Based on the work of Chavarria et al., (2021), cited in our manuscript), the density of cattle in our traditional cattle and silvopasture treatment is 0.85 and 0.87 cattle per ha, respectively. However, the effective density of our traditional pasture is 0.435 head per ha. We call this effective density because the cattle are only present on the pasture for half of the year to avoid severe degradation. The cattle owner rotates his cattle every month such that the 0.87 cattle per ha value represents the situation as if the cows are on the single pasture for the entire year, such that the 0.87 ends up being twice the annual density. To be cautious we inflated our densities to 1 cow per hectare and 1 per silvopasture based landowner information in the area. We are confident of the expense side of our farm balance as we have vast experience in the cost and labor and supplies associated with fencing and cleaning and the cost of veterinary supplies comes from the market, etc.

Silvopasture systems

Our manuscript assesses silvopasture system *as practiced in the Panama Canal Watershed* (PCW) and promoted by the Panama Canal Authority (ACP) through one of their extension programs. We realize that not all silvopasture systems are alike. In the Panama Canal Watershed, as in many other areas of Panama, the two most important components of the silvopasture system are 1) improved pasture, and 2) stream fencing to encourage forest growth and to restrict – but not eliminate – access by cattle. This latter distinction is important because cattle rely on streams to drink water throughout the year and even congregate in the streams, seeking shade provided by the forest buffer in the dry season (Chavarria et al., 2021). In many silvopasture systems farmers/ranchers **restrict stream access by cattle throughout the year** and either create up slope storage ponds (problematic for lengthy dry seasons) and/or pump water up to drinking basins from streams. However, following local practice, we do not implement these measures, and consequently, **we do not include these associated expenses in our analysis**. Our experience and observations with the ACP program indicate that success and compliance of the program does not hinge on the survival of trees that were planted in the pasture as part of the program. **However, there is cost involved in planting and maintaining trees in the pastures, which we have not factored into our analysis**. It's worth noting that our own pasture already had a reasonable density of dispersed trees prior to initiating our treatment, providing shade benefits to our cattle. Additionally, we practice continual rotation between paddocks, which is an important component of silvopasture systems.

Silvopasture can often include line planting of leguminous shrubs or trees, with *Leucaena leucocephala* being the most commonly planted. *Leucaena*, grown in rows approximately 4 meters apart, provides cattle with an additional source of nutrition along with pasture grass. However, *Leucaena* **does not grow well or even survive on low nutrient acidic soils** like those found at our site and the subject of the widespread, nutrient poor soils of this manuscript. An alternative plant, *Tithonia diversifolia*, known in Spanish as “El botón de oro” is also grown in rows, used for living fences, and forage banks and is suitable for acidic soils (Murgueitio et al., 2011). Unfortunately, it failed at our site, which could have been a result of inadequate experience in maintaining it. To our knowledge, the use of *Tithonia diversifolia* has not been successfully implemented at scale in the PCW silvopasture systems. Therefore, we did not include the cost of planting and maintaining this in our system. We note that neither we nor the farmers working with us could get it to grow well – even after experts from a Colombian NGO, well known for its extension program with silvopasture systems, visited and guided us (CIPAV – Centro para la Investigacion en Sistemas Sostenibles de Produccion Agropecuaria). We also did not include the associated costs of an extension program to make this system work on poor soils. Certainly, it is worth considering including this cost, because technology transfers often play a crucial role, as noted by Skidmore et al., (2022) who found that improved production in the Brazilian Amazon can rarely be attained without a consultant. However, we again do not have good data on the cost of that within our study area. We have now included a comment on this in the manuscript (L976-978) where we discuss reasons for different cattle NPV values across different countries.

Another type of silvopasture system moves beyond incorporating shade and fodder shrubs and trees in line plantings within a pasture to convert the system into a virtual salad bowl of shrubs.

This Intensive Silvopastoral System (ISS) has been shown in **case studies** to be highly productive (Calle et al., 2013). However, to our knowledge, this system is not widely practiced in Panama and the ISS farm in the case study cited here is found on fertile soils. The fertile and non-acidic soils enable the successful growth of a variety of fodder plants, allowing for a dramatic increase in cattle density (up to 5 head per ha). Unfortunately, this crucial aspect is often not explicitly stated in cattle extension and promotion literature or even in the few peer reviewed papers we have come across on the subject. Soil fertility data is rarely made clear or presented in documents, necessitating us to cross-reference location with soil maps. This information gap emphasizes the need for further clarification and understanding.

NPV of cattle systems

Reviewer #1 wondered about the low NPV of the cattle systems and suggested we spend a few sentences elaborating on this, comparing, and contrasting to other countries in Latin America, particularly Brazil. Before we address the economics of cattle systems, we note that the average density of cattle in the Brazilian Amazon is 1 head per ha (Skidmore et al., 2022). We have included this reference in our manuscript and note this is the same number we used in our analysis for our traditional system analysis. One contributing factor to low NPV is the relatively low cattle density, which is representative of what can be expected on infertile, acidic soils. However, as we write below, cattle ranching is not simply about rearing cows on a single pasture. It may involve many movements (in Brazil often from illegal sites – see below) and fattening operations that can enhance profitability. Farm size is also critical, with small holders (emphasized in this manuscript for forestry) facing significant disadvantages (again, see below). We have wondered about the economics of cattle for years and have even conducted periodic searches of the literature for peer reviewed studies that would help us understand why it might be so widespread and so heavily promoted (silvopasture and other improved systems). Existing literature on the forest transition in Panama and beyond (e.g., Wright and Samaniego, 2008) has long highlighted the trend of individuals allowing their pastures go fallow as they migrate to cities for better economic opportunity (but they often recut the forest or clean the land to maintain land tenure rights – discussed in our manuscript). It appears that traditional cattle is not sufficiently profitable to keep people on the land in many areas.

To get a better understanding of the scientific basis for the promotion of cattle systems by research and outreach organizations, we have contacted local and regional experts, in addition to doing our own literature search. We sought more than one-off case studies from ecologically dissimilar regions and found a paper focused on areas in Peru. Chizmar et al., (2020) studied the economics of traditional cattle, silvopasture and other systems (including tourist stores on ranches) in the Amazonas region of Peru. However, the study site's characteristic, situated at over 4,000 m with Holdridge Life Zones of "dry forest lower montane tropical and moisture forest premontane tropical" (as per the authors), bear little resemblance to the biophysical conditions of low land tropical forests in areas of Panama. A further flag of concern is that the Holstein, Simmental, and Brown Swiss cattle found in the study site and region are not typically found in the lowland forest areas of Panama and beyond. With the breed and more hospitable climate at higher elevations we might expect these systems to be more financially viable; at the 8% discount rate assessed by the authors, the typical cattle pasture and typical silvopasture systems had NPVs of \$318.64 and \$321.91, respectively. Table RR 2 below (and included in

supplementary materials as Table S4) presents a sensitivity analysis conducted with our data showing the effective NPV of our traditional cattle at 8% interest rate/discount rate is \$502.42 per ha while our silvopasture is \$24.89 at 0.85 head per ha and \$58.57 at 2 head per ha. The results from Chizmar et al., (2020), a more hospitable site to raise cattle than sites targeted in our study, suggest that by erring on the conservative side, we over-estimate the NPV of cattle.

Table RR2. Sensitivity analysis of Net Present Value (NPV) of traditional and silvopasture systems in central Panama. Cattle density is varied as per explanation in the text above.

Interest rate	Pasture			Silvopasture		
	1 cow per hectare	0.87 cows per ha	0.435 cows per ha	1 cow per hectare	0.85 cows per hectare	2 cows per ha
	NPV (\$US per ha)					
1	2803.97	2439.45	1219.73	223.20	189.72	446.40
2	2407.05	2094.13	1047.07	167.41	142.29	334.81
3	2086.45	1815.21	907.60	126.40	107.44	252.81
4	1825.05	1587.79	793.90	95.85	81.47	191.69
5	1609.93	1400.64	700.32	72.69	61.79	145.38
6	1431.28	1245.22	622.61	54.80	46.58	109.60
7	1281.59	1114.98	557.49	40.68	34.58	81.37
8	1155.06	1004.90	502.45	29.28	24.89	58.57
9	1047.23	911.09	455.54	19.86	16.88	39.72
10	954.58	830.49	415.24	11.89	10.11	23.78
11	874.39	760.72	380.36	5.01	4.26	10.02
12	804.47	699.89	349.94	-1.05	-0.89	-2.10
13	743.09	646.49	323.24	-6.48	-5.50	-12.95
14	688.87	599.32	299.66	-11.40	-9.69	-22.79
15	640.69	557.40	278.70	-15.91	-13.52	-31.82

Silvopasture in the Americas

To learn from the experts about the economics of cattle in the Americas, at the regional level, we focused our search on work published by CIAT (International Center for Tropical Agriculture). Of the 15 documents we reviewed, only five were from peer reviewed sources, and none of them presented landscape scale replication and assessment of the economics of traditional systems or silvopasture systems. Charry et al., (2019) is a study on the willingness to accept higher payments for improved animal welfare; the study they cited for the economic benefits of silvopasture systems was from North American temperate zone. Enciso et al., (2019) regarded the economics of incorporating *Leucaena* and by the authors own data was on a fertile site at CIAT headquarters. Enciso et al., (2021) investigated grass species suitable for poorly drained, waterlogged soils in the Orinoquía region of Colombia and is thus not relevant to our study. Gaviria-Urbe et al., (2020) studies methane emissions and improved forage, again focusing on *Leucaena* and again conducted on fertile soils at CIAT headquarters. Morena Lerma et al., (2022) is a qualitative (as stated by the authors) public sector analysis of cattle in Costa Rica, Colombia, and Argentina that focusses on reduction of greenhouse gasses and presents no economic data.

The other 10 documents from CIAT included policy documents, fact sheets, and reports and included economic analyses of traditional and silvopasture systems conducted within the context of a payment for environmental service program. Again, most of the studies that were cited in these documents were conducted at relatively fertile sites using forage crops that will not grow well on infertile acid soils targeted in our manuscript. For example, Sandoval et al.,

(2022), conducted their analysis on fertile soils (which was not directly stated, it took tracking down citations within citations to discover this) with a stocking of 3 cattle (the exact area was not specified in the fact sheet but assumed to be per ha). Their systems nevertheless only achieved positive NPVs with the inclusion of carbon payments for avoided methane emissions. Notably, they used a price of US\$45.25 – more than twice the price we use in our manuscript – per Mg of CO₂e.

Methane emissions

The avoided methane emissions in Sandoval et al., (2022) and others trace back to a study by Gaviria-Uribe et al., (2020) conducted on relatively fertile soils using *Leucaena* as a forage crop. In this study, cattle were placed in polyethylene plastic tubes for 3 days to measure methane emissions after the appropriate dietary treatments. We are less than comfortable with the idea of using these sorts of limited, though no doubt difficult to obtain, data that are collected on sites very different from the infertile soils targeted in our manuscript and in our analysis to project carbon payments related on the transition from traditional to silvopasture systems.

Cattle in Brazil

Regarding the connection between our work and cattle systems in Brazil mentioned by Reviewer #1, it is worth noting that vast areas of the Brazilian Amazon Basin have been converted into cattle pasture, with 170 million head of cattle being produced there in 2015 (Elias et al., 2022). Recent efforts have been made to certify cattle production to ensure forests are not illegally cleared to produce cattle. Nevertheless, there are still instances of individuals associated with the industry raising cattle on illegally cleared lands, including land in protected and / or indigenous areas (West et al., 2022). These individuals rear the cattle for the first months or years and then sell them to certified ranches for finishing (Carvalho et al., 2021). Such complexities pose challenges to understanding the economics of the industry.

Skidmore et al., (2022) and Elias Arantes et al., (2018) report the average density in the Brazilian Amazon as 1 head per ha, on infertile soils. Cattle in the Amazon involves a mix of small holders and large ranches. In a financial analysis of cattle production in the Brazilian Amazon, de Oliveira Silva et al., (2017), concluded that economies of scale are necessary to make cattle profitable. Their analysis included farms averaging 600 ha, situated on sufficiently flat land to allow for mechanized production. They found the NPV of cattle ranged from approximately *negative* US\$13 to *positive* \$57 per ha, with the positive NPV including carbon payments. They found that increases in soil organic carbon (as opposed to reduced greenhouse gas emissions) in their system was the most important factor contributing to the increased carbon payment with carbon difference from the traditional system and the improved system being approximately 15 Mg per ha **over 20 years** (note they priced CO₂e at \$15.1 US per Mg). In addition to economies of scale, small holders in the Amazon face challenges in accessing credit (Skidmore et al., 2022 and references therein) an additional barrier to financing the improvements necessary to improve their financial viability.

One approach to enhancing cattle production in the Amazon is by moving cattle around and finishing or fattening them with grain during the last months of their lives. This practice can reduce the time it takes to grow a cow and make them bigger. In their analysis of

improvements through better feed, Skidmore et al., (2022) used data for grain production within 50 km of the fattening location: more distant areas incur increased transportation costs and cost of the grain could outweigh the benefit. This can be a further challenge to small holders or those living in a heterogeneous landscape with limited ability to produce grain for animal feed. The improved diet can also reduce greenhouse gas emissions, and this too can be marketed for carbon credits and impact the financial viability of an operation (Skidmore et al. 2022). Nevertheless, even with the inclusion of carbon payments and the benefits from fattening or finishing described above, the per ha NPV of improved operations remains lower than the NPV we found in our manuscript. Given the narrow profit margins, we can only assume it is the economies of scale and vast areas that are used to produce cattle that generate the profits and cash flows that drive the industry.

As with the case from the data in Colombia, we are uncomfortable incorporating a carbon payment based on limited controlled studies that extrapolate reduced methane or other greenhouse gas emissions across farms, regions, and even countries. Though more straightforward to measure, we feel the same about carbon sequestration in soils. Our manuscript is to test the profitability of different restoration techniques on infertile soils without and with subsidies and carbon payments. Considering the points we have raised earlier, along with our very conservative and profitable (as compared to other analysis) evaluation of cattle, and taking into account Reviewer 2's concerns about exaggerated and even fraudulent claims related to avoided deforestation, we feel justified in excluding carbon payments for avoided emissions with cattle in our analysis. We briefly address this in the Methods section of our manuscript.

Extended transaction costs discussion

Two reviewers (R2 and R3) either directly or indirectly brought up the issue of transaction costs (TAC). Reviewer 3 mentioned it directly while Reviewer 2 indirectly referred to a specific aspect of transaction costs in their comment about monitoring. Reviewer 2 argued that recent instances of fraudulent or inappropriate and, arguably immoral, practices with the Reduced Emissions due to land Degradation and Deforestation (REDD and REDD+) have highlighted the need for better monitoring, to which governments and markets have called for. Reviewer 3 partially addresses this comment in their review by pointing out that making payments independent of biomass would reduce uncertainty for landowners and presumably reduce the transaction costs associated with monitoring. We address questions Reviewer 2 raised regarding our economic incentives and subsidy approach extensively elsewhere in this response. In bringing up the issue of TAC directly, Reviewer 3 raised an important issue that, while they recognize that it is not the topic per se of this manuscript, it should (and we agree needs to) be discussed. We now briefly mention this in our manuscript (see lines 926 to 938) and further address this in our specific response to reviewers for both Reviewer 3 and Reviewer 2. Below we address reviewer's comments in more detail.

Reviewer 3 lists things that are included in Transaction Costs (TAC) and pointed us to three specific articles. Van Kooten et al., (2002) lists these in their article as do the others. Van

Kooten, (2020) looks at planting trees in Canada (western) as an offset; Rendón Thompson et al., (2013) look at REDD+ projects in Peru; Pearson et al., (2014) look at both REDD+ and reforestation and afforestation in East Africa and Latin America (REDD in SE Asia). Costs (TAC) vary widely. Reviewer 3 helpfully summarized TAC as “search and contract costs, certification, mandatory inventory reports, mandatory C buffer pools etc.”

Insurance costs

Pearson et al., (2014) point out that in most projects globally, total project costs are underestimated by 30% as TAC are not included. Comparing transaction costs across projects and continents, Pearson et al., (2014) found that insurance costs can be as much or greater than 80% of the transactions costs and found them to be the single largest transaction cost and approximately 40% of these costs for the reforestation project evaluated in South America. Insurance costs are handled by creating a buffer pool for carbon, some quantity of carbon is set aside in case of unforeseen circumstances. In a review of protocols dominating forest carbon certification, (Haya et al., 2023) report buffer pools ranging from 7% to 27% of the total carbon in a project (their Table 4). Sinacore et al., (2017) found coarse roots of trees excavated within plantations near our study sites to represent approximately 27% of total tree dry biomass (carbon). In their review of eight dominant IFM (improved forest management) protocols, Haya et al., (2023) found all but one (VCS VM001 – which in contrast to the other protocols only includes aboveground biomass) include belowground biomass, a pool that includes both coarse roots (≥ 2 mm) and fine roots < 2 mm, a distinction we make as Sinacore et al., (2017) only measured coarse roots.

We have justified elsewhere in this response our use of flat payments (see carbon response above). As referenced in our manuscript, we have pilot programs testing the ability to bring flat payments systems to scale. The first is with the Panama Canal Authority where we are paying US \$130 per ha per year to landowners who allow their cattle pastures to go fallow and regenerate in secondary forest. The carbon payments from individuals in this project is based only on aboveground carbon. In this program the Panama Canal Authority finances the planting of native timber species in an enrichment planting program (following our experimental design). Hall et al., (2022) (cited in the manuscript) published several deforestation and carbon accrual scenarios for central Panama (23% of the country) and also published carbon accumulation for aboveground biomass based on the 1.2 million measurements of secondary forest growth used in this manuscript. Hall et al., (2022) also used data from the study site to include coarse roots (from Sinacore et al., 2017), carbon in the top 20 cm of the mineral soils (from Neumann-Cosel et al., 2011), lianas (Lai et al., 2017) and from coarse woody debris (Gora et al., 2019). Hall et al. note that taken together, these other compartments add an additional 60% of carbon to the aboveground tree carbon in the region. As noted, coarse roots by themselves account for what would be required as a buffer or for insurance in carbon trading programs.

The second site where we are piloting flat carbon payments is in the Ngäbe-Buglé Comarca (semi-autonomous indigenous area in Panama...and larger than any individual Province in Panama). Here we are restoring forest on degraded land. If reviewers are interested they can read about it here: <https://stri.si.edu/story/indigenous-reforestation>. We use the flat \$130 per

ha per year (after year 4) carbon payments with payments guaranteed for the first 20 years by a funder (funds already received). Finally, McGill University is in the process of offsetting their carbon emissions associated with university travel. They are quite interested in transitioning their program to a flat payment system.

Monitoring costs

Pearson et al., (2014) found monitoring costs to be the second highest component of TAC, accounting for approximately 25% (as read from their Table 2) of the project cost. We follow Reviewer 3 in our belief that monitoring costs when flat payments are made need not be high; Reviewer 2 also points out that these costs are coming down with technology. Pearson et al., (2014) found TAC to be US \$7.71 Mg CO₂e. Assuming insurance costs (41%) are already covered in belowground and other pools, the TAC would be approximately US \$4.55. In their discussion of the flat payment system used on Costa Rica, Norden (2014) notes that the TAC are US \$ 12 per ha per year, or about 18% of the total carbon payment (costs that do not appear to include insurance costs). The total payment in our projects on a per Mg basis and as reported in our manuscript is US \$18 per Mg CO₂e at \$130 ha⁻¹. Using Norden (2014) TAC in our projects would be approximately US\$ 3.24. Given that our monitoring costs would be low with flat carbon payments, it seems reasonable from these data to assume that TAC may range from as little as US \$2.50 to as much as US \$ 5.00 per Mg CO₂e. Fenichel et al., (2019) and Norden, (2014) include these costs as part of their flat payments to landowners. Given that our insurance costs are already covered in our model, that monitoring costs should be relatively low, and the uncertainty in TAC but not prohibitive, for simplicity, we follow these authors and assume they are covered in the payments to landowner.

We now briefly discuss in the manuscript the issue of TAC, citing the relevant literature, explaining that we do not specifically look at TAC, assuming they are included in the payment to the landowner.

Literature cited

- Adamowicz, W., Calderon-Etter, L., Entem, A., Fenichel, E.P., Hall, J.S., Lloyd-Smith, P., Ogden, F.L., Regina, J.A., Rouhi Rad, M., Stallard, R.F., 2019. Assessing ecological infrastructure investments. *Proceedings of the National Academy of Sciences* 1–8. <https://doi.org/10.1073/pnas.1802883116>
- Asner, G.P., Mascaro, J., Anderson, C., Knapp, D.E., Martin, R.E., Kennedy-Bowdoin, T., van Breugel, M., Davies, S., Hall, J.S., Muller-Landau, H.C., Potvin, C., Sousa, W., Wright, J., Birmingham, E., 2013. High-fidelity national carbon mapping for resource management and REDD+. *Carbon Balance and Management* 8, 7. <https://doi.org/10.1186/1750-0680-8-7>
- Calle, Z., Murgueitio, E., Chará, J., Molina, C.H., Zuluaga, A.F., Calle, A., 2013. A strategy for scaling-up intensive silvopastoral systems in Colombia. *Journal of sustainable forestry* 32, 677–693.
- Carvalho, R., Rausch, L., Munger, J., Gibbs, H.K., 2021. The role of high-volume ranches as cattle

- suppliers: Supply chain connections and cattle production in Mato Grosso. *Land* 10, 1098.
- Charry, A., Narjes, M., Enciso, K., Peters, M., Burkart, S., 2019. Sustainable intensification of beef production in Colombia—Chances for product differentiation and price premiums. *Agric Econ* 7, 22. <https://doi.org/10.1186/s40100-019-0143-7>
- Chavarria, K.A., Saltonstall, K., Vinda, J., Batista, J., Lindmark, M., Stallard, R.F., Hall, J.S., 2021. Land use influences stream bacterial communities in lowland tropical watersheds. *Sci Rep* 11, 21752. <https://doi.org/10.1038/s41598-021-01193-7>
- Chizmar, S., Castillo, M., Pizarro, D., Vasquez, H., Bernal, W., Rivera, R., Sills, E., Abt, R., Parajuli, R., Cubbage, F., 2020. A discounted cash flow and capital budgeting analysis of silvopastoral systems in the Amazonas region of Peru. *Land* 9, 1–15. <https://doi.org/10.3390/land9100353>
- Covelli Metcalf, E., Mohr, J.J., Yung, L., Metcalf, P., Craig, D., 2015. The role of trust in restoration success: public engagement and temporal and spatial scale in a complex social-ecological system: Trust in restoration success. *Restor Ecol* 23, 315–324. <https://doi.org/10.1111/rec.12188>
- Davoudi, S., Brooks, E., 2014. When Does Unequal become Unfair? Judging Claims of Environmental Injustice. *Environ Plan A* 46, 2686–2702. <https://doi.org/10.1068/a130346p>
- de Oliveira Silva, R., Barioni, L.G., Hall, J.A.J., Moretti, A.C., Fonseca Veloso, R., Alexander, P., Crespolini, M., Moran, D., 2017. Sustainable intensification of Brazilian livestock production through optimized pasture restoration. *Agricultural Systems* 153, 201–211. <https://doi.org/10.1016/j.agsy.2017.02.001>
- Elias Arantes, A., Couto, V.R. de M., Sano, E.E., Ferreira, L.G., 2018. Livestock intensification potential in Brazil based on agricultural census and satellite data analysis. *Pesquisa Agropecuária Brasileira* 53, 1053–1060.
- Elias, M., Kandel, M., Mansourian, S., Meinzen-Dick, R., Crossland, M., Joshi, D., Kariuki, J., Lee, L.C., McElwee, P., Sen, A., Sigman, E., Singh, R., Adamczyk, E.M., Addoah, T., Agaba, G., Alare, R.S., Anderson, W., Arulingam, I., Bellis, S., Kung V., Birner, R., De Silva, S., Dubois, M., Duraisami, M., Featherstone, M., Gallant, B., Hakhu, A., Irvine, R., Kiura, E., Magaju, C., McDougall, C., McNeill, G.D., Nagendra, H., Nghi, T.H., Okamoto, D.K., Paez Valencia, A.M., Pagella, T., Pontier, O., Post, M., Saunders, G.W., Schreckenber, K., Shelar, K., Sinclair, F., Gautam, R.S., Spindel, N.B., Unnikrishnan, H., Wilson, G. taa’a gaagii ng. aang N., Winowiecki, L., 2022. Ten people-centered rules for socially sustainable ecosystem restoration. *Restoration Ecology* 30. <https://doi.org/10.1111/rec.13574>
- Enciso, K., Charry, A., Castillo, Á.R., Burkart, S., 2021. Ex-ante evaluation of economic impacts of adopting improved forages in the Colombian Orinoquía. *Frontiers in Environmental Science* 9, 673481.
- Enciso, K., Sotelo, M., Peters, M., Burkart, S., 2019. The inclusion of *Leucaena diversifolia* in a Colombian beef cattle production system: An economic perspective. *Tropical Grasslands-Forrajes Tropicales* 7, 359–369.
- Engel, S., Pagiola, S., Wunder, S., 2008. Designing payments for environmental services in theory and practice: An overview of the issues. *Ecological Economics* 65, 663–674. <https://doi.org/10.1016/j.ecolecon.2008.03.011>
- Fenichel, E.P., Adamowicz, W., Ashton, M.S., Hall, J.S., 2019. Incentive systems for forest-based

- ecosystem services with missing financial service markets. *Journal of the Association of Environmental and Resource Economists* 6, 319–347. <https://doi.org/10.1086/701698>
- Gaviria-Urbe, X., Bolivar, D.M., Rosenstock, T.S., Molina-Botero, I.C., Chirinda, N., Barahona, R., Arango, J., 2020. Nutritional quality, voluntary intake and enteric methane emissions of diets based on novel Cayman grass and its associations with two *Leucaena* shrub legumes. *Frontiers in Veterinary Science* 764.
- Gora, E.M., Kneale, R.C., Larjavaara, M., Muller-Landau, H.C., 2019. Dead wood necromass in a moist tropical forest: stocks, fluxes, and spatiotemporal variability. *Ecosystems* 22, 1189–1205.
- Hall, J.S., Plisinski, J.S., Mladinich, S.K., van Breugel, M., Lai, H.R., Asner, G.P., Walker, K., Thompson, J.R., 2022. Deforestation scenarios show the importance of secondary forest for meeting Panama’s carbon goals. *Landscape Ecology* 37, 673–694. <https://doi.org/10.1007/s10980-021-01379-4>
- Haya, B.K., Evans, S., Brown, L., Bukoski, J., Butsic, V., Cabiyo, B., Jacobson, R., Kerr, A., Potts, M., Sanchez, D.L., 2023. Comprehensive review of carbon quantification by improved forest management offset protocols. *Frontiers in Forests and Global Change* 6, 12.
- Hirsch, P.D., Adams, W.M., Brosius, J.P., Zia, A., Bariola, N., Dammert, J.L., 2010. Acknowledging Conservation Trade-Offs and Embracing Complexity: Conservation Trade-Offs and Complexity. *Conservation Biology* no-no. <https://doi.org/10.1111/j.1523-1739.2010.01608.x>
- Kroeger, T., 2013. The quest for the “optimal” payment for environmental services program: Ambition meets reality, with useful lessons. *Forest Policy and Economics* 37, 65–74. <https://doi.org/10.1016/j.forpol.2012.06.007>
- Lai, H.R., Hall, J.S., Turner, B.L., van Breugel, M., 2017. Liana effects on biomass dynamics strengthen during secondary forest succession. *Ecology* 98, 1062–1070. <https://doi.org/10.1002/ecy.1734>
- Lliso, B., Pascual, U., Engel, S., 2021. On the role of social equity in payments for ecosystem services in Latin America: A practitioner perspective. *Ecological Economics* 182, 106928. <https://doi.org/10.1016/j.ecolecon.2020.106928>
- Messier, C., Bauhus, J., Sousa-Silva, R., Auge, H., Baeten, L., Barsoum, N., Bruelheide, H., Caldwell, B., Cavender-Bares, J., Dhiedt, E., Eisenhauer, N., Ganade, G., Gravel, D., Guillemot, J., Hall, J.S., Hector, A., Hérault, B., Jactel, H., Koricheva, J., Kreft, H., Mereu, S., Muys, B., Nock, C.A., Paquette, A., Parker, J.D., Perring, M.P., Ponette, Q., Potvin, C., Reich, P.B., Scherer-Lorenzen, M., Schnabel, F., Verheyen, K., Weih, M., Wollni, M., Zemp, D.C., 2022. For the sake of resilience and multifunctionality, let’s diversify planted forests! *CONSERVATION LETTERS* 15. <https://doi.org/10.1111/conl.12829>
- Montagnini, F., Eibl, B., Grance, L., Maiocco, D., Nozzi, D., 1997. Enrichment planting in overexploited subtropical forests of the Paranaense region of Misiones, Argentina. *Forest Ecology and Management* 99, 237–246. [https://doi.org/10.1016/S0378-1127\(97\)00209-0](https://doi.org/10.1016/S0378-1127(97)00209-0)
- Montagnini, F., González, E., Porras, C., Montagnini, F., González, E., Porras, C., Rheingans, R., Rica, C., 1995. Mixed and pure forest plantations in the humid neotropics: a comparison of early growth, pest damage and establishment costs. *Commonwealth Forestry Association* 1 74, 306–314. <https://doi.org/10.2307/42608324>
- Morena Lerma, L., Díaz Baca, M.F., Burkart, S., 2022. Public policies for the development of a

- sustainable cattle sector in Colombia, Argentina, and Costa Rica: A comparative analysis (2010–2020). *Frontiers in Sustainable Food Systems* 6, 72.
- Murgueitio, E., Calle, Z., Uribe, F., Calle, A., Solorio, B., 2011. Native trees and shrubs for the productive rehabilitation of tropical cattle ranching lands. *Forest Ecology and Management* 261, 1654–1663.
- Neumann-Cosel, L., Zimmerman, B., Hall, J.S., van Breugel, M., Elsenbeer, H., 2011. Soil carbon dynamics under young tropical secondary forests on former pastures - a case study from Panam. *Forest Ecology and Management* 261, 1625–1633.
- Norden, A., 2014. Payment Types and Participation in Payment for Ecosystem Services Programs: Environment for Development.
- Pascual, U., Phelps, J., Garmendia, E., Brown, K., Corbera, E., Martin, A., Gomez-Baggethun, E., Muradian, R., 2014. Social Equity Matters in Payments for Ecosystem Services. *BioScience* 64, 1027–1036. <https://doi.org/10.1093/biosci/biu146>
- Pearson, T.R.H., Brown, S., Sohngen, B., Henman, J., Ohrel, S., 2014. Transaction costs for carbon sequestration projects in the tropical forest sector. *Mitig Adapt Strateg Glob Change* 19, 1209–1222. <https://doi.org/10.1007/s11027-013-9469-8>
- Pham, T.T., Mai, Y.H., Moeliono, M., Brockhaus, M., 2016. Women’s participation in REDD+ national decision-making in Vietnam. *Int. Forest. Rev.* 18, 334–344. <https://doi.org/10.1505/146554816819501691>
- Pinnschmidt, A., Yousefpour, R., Nölte, A., Murillo, O., Hanewinkel, M., 2022. Economic potential and management of tropical mixed-species plantations in Central America. *New Forests*. <https://doi.org/10.1007/s11056-022-09937-7>
- Rendón Thompson, O.R., Paavola, J., Healey, J.R., Jones, J.P.G., Baker, T.R., Torres, J., 2013. Reducing Emissions from Deforestation and Forest Degradation (REDD+): Transaction Costs of Six Peruvian Projects. *E&S* 18, art17. <https://doi.org/10.5751/ES-05239-180117>
- Sandoval, D., Florez, J.F., Enciso, K., Sotelo, M., Burkart, S., 2022. Economic and environmental evaluation of a silvo-pastoral system in Colombia: An ecosystem service perspective.
- Shyamsundar, P., Cohen, F., Boucher, T.M., Kroeger, T., Erbaugh, J.T., Waterfield, G., Clarke, C., Cook-Patton, S.C., Garcia, E., Juma, K., Kaur, S., Leisher, C., Miller, D.C., Oester, K., Saigal, S., Siikamaki, J., Sills, E.O., Thaug, T., Trihadmojo, B., Veiga, F., Vincent, J.R., Yi, Y., Zhang, X.X., 2022. Scaling smallholder tree cover restoration across the tropics. *Global Environmental Change* 76, 102591. <https://doi.org/10.1016/j.gloenvcha.2022.102591>
- Sinacore, K., Hall, J.S., Potvin, C., Royo, A.A., Ducey, M.J., Ashton, M.S., 2017. Unearthing the hidden world of roots: Root biomass and architecture differ among species within the same guild. *PLOS ONE* 12, 1–22. <https://doi.org/10.1371/journal.pone.0185934>
- Skidmore, M.E., Sims, K.M., Rausch, L.L., Gibbs, H.K., 2022. Sustainable intensification in the Brazilian cattle industry: the role for reduced slaughter age. *Environ. Res. Lett.* 17, 064026. <https://doi.org/10.1088/1748-9326/ac6f70>
- Sovacool, B.K., 2023. Expanding carbon removal to the Global South: Thematic concerns on systems, justice, and climate governance. *Energy and Climate Change* 4, 100103. <https://doi.org/10.1016/j.egycc.2023.100103>
- van Kooten, G.C., 2020. How effective are forests in mitigating climate change? *Forest Policy and Economics* 120, 102295.
- van Kooten, G.C., Shaikh, S.L., Suchánek, P., 2002. Mitigating Climate Change by Planting Trees: The Transaction Costs Trap. *Land Economics* 78, 559–572.

<https://doi.org/10.2307/3146853>

Vincent, J.R., Curran, S.R., Ashton, M.S., 2021. Forest Restoration in Low- and Middle-Income Countries. *Annu. Rev. Environ. Resour.* 46, 289–317. <https://doi.org/10.1146/annurev-environ-012220-020159>

West, T.A., Rausch, L., Munger, J., Gibbs, H.K., 2022. Protected areas still used to produce Brazil's cattle. *Conservation Letters* e12916.

Wright, S.J., Samaniego, M.J., 2008. Historical, demographic, and economic correlates of land-use change in the Republic of Panama. *Ecology and Society* 13.

Reviewers' Comments:

Reviewer #1:

Remarks to the Author:

I thank the authors again for the opportunity to review this paper. I think the authors have made the necessary revisions to the text and I have no further comments to contribute. I believe that the results are very interesting and will be a good contribution to the literature.

Reviewer #2:

Remarks to the Author:

Thank you for responding so thoroughly to my comments on the original manuscript. Indeed, in nearly 40 years of reviewing manuscripts for a wide range of journals, I have never before seen such a neatly organized and detailed response to reviewers' comments as the booklet you prepared. I wish more authors used that format.

I'm fully satisfied with the revisions you've made in response to my comments. I don't have any remaining comments.

Reviewer #3:

Remarks to the Author:

Thank you for addressing my comments, and for providing detailed explanations to all reviewers' comments. I think the MS is clearly improved.

Note: I am reviewing the Track Changes Microsoft Word version of the revised manuscript, and none of the line numbers in your response to reviewers match this Word version. I am using the line numbers in this Word version of the revised MS in my comments below. However, I have accepted all your tracked change sin the text below to be able to use Track Changes to show what changes I consider should be made to your revised version.

1) L527: "we use a flat carbon payment fee..." Delete "fee". You are talking about carbon payments; there is no need for "fee".

2) I saw in your response to another reviewer that you changed all "\$" to "US\$". But in the Track Changes Word file I am reviewing, I still see many instances of just "\$." Please change these to "US\$".

3) LL528-530: "..., we know that secondary forest in central Panama accrues carbon at an annual smoothed rate of 6.9 Mg CO₂e in aboveground tree biomass⁶² (AGB) or approximately \$18 per Mg CO₂e at \$130 ha⁻¹."

This should be changed to: "..., we know that secondary forest in central Panama accrues carbon at an annual smoothed rate of 6.9 Mg CO₂e in aboveground tree biomass⁶² (AGB), or approximately \$130 ha⁻¹ at \$18 per Mg CO₂e at \$130 ha⁻¹."

4) LL719-721: "While [ADD: carbon stocks in"] [delete: of] secondary forests in Agua Salud are high for moist tropical forests, at approximately 185 Mg CO₂e at 30 years³¹ (Table 3) they are about..."

5) Reading your response to my comment number 5 on your original, I believe your characterization of the "silvopasture system" is misleading. My comment was that "5) I did not see any mention of C payments for silvopasture. If this is correct, could you explain why tree cover in silvopasture is not receiving any C payments? There certainly are PES programs in tropical countries that support silvopasture establishment, so it would be helpful to understand why no C payments for this land use were considered." Your response was: "This is a fair question. We have reviewed the literature for this response below. We explain that silvopasture means different things to different people. As we state, in our case it mostly means improved pasture and restricting access to streams."

All definitions of silvopasture that I am familiar with in the published literature describe it as the

purposive integration of trees into a grazing system (hence the term "silvo"). Even a quick Google search of the term brings up only results that meet this description. Improved pasture and restricted stream access does not meet the silvopasture definition. If your "silvopasture" system does not include the planting or retention of trees as a feature of the grazing lands, then I suggest you use a different descriptor for this pasture system. "Improved pasture" seems like a perfectly reasonable choice, and you can explain that this also includes some restriction on stream access. But a riparian tree buffer that is fenced off from grazing lands does not integrate trees into grazing lands (the defining feature of a silvopasture system). Using silvopasture to describe a system that does not include trees in pasture lands in some purposive manner risks conjuring up an incorrect image of what this system looks like, for anyone who is familiar with silvopasture systems. Note that I do appreciate the discussion on silvopasture that you provide in your response to reviewers, in which you explain clearly what your "silvopasture" system looks like. However, I did not see any definition of silvopasture, or any description of the system that you describe as silvopasture, in the revised manuscript. Thus, my concern about using the term silvopasture for this system remains unmitigated.

Response to Reviewers

Reviewer #1 (Remarks to the Author):

Comment: I thank the authors again for the opportunity to review this paper. I think the authors have made the necessary revisions to the text and I have no further comments to contribute. I believe that the results are very interesting and will be a good contribution to the literature.

Response: Thank you for reviewing our paper again and for your helpful comments in the initial review. We look forward to the paper being published.

Reviewer #2 (Remarks to the Author):

Comment: Thank you for responding so thoroughly to my comments on the original manuscript. Indeed, in nearly 40 years of reviewing manuscripts for a wide range of journals, I have never before seen such a neatly organized and detailed response to reviewers' comments as the booklet you prepared. I wish more authors used that format.

I'm fully satisfied with the revisions you've made in response to my comments. I don't have any remaining comments.

Response: Thank you so much for the compliment. We really did enjoy being given the opportunity to dive deeper and take a bit more space to give thorough responses.

Reviewer #3 (Remarks to the Author):

Comment: Thank you for addressing my comments, and for providing detailed explanations to all reviewers' comments. I think the MS is clearly improved.

Note: I am reviewing the Track Changes Microsoft Word version of the revised manuscript, and none of the line numbers in your response to reviewers match this Word version. I am using the line numbers in this Word version of the revised MS in my comments below. However, I have accepted all your tracked changes in the text below to be able to use Track Changes to show what changes I consider should be made to your revised version.

Response: Thank you so much for reviewing our paper again. We really appreciate you catching additional edits to clean up the paper.

Line Comments:

1) L527: “we use a flat carbon payment fee...” Delete “fee”. You are talking about carbon payments; there is no need for “fee”.

Response: Fixed (L419)

2) I saw in your response to another reviewer that you changed all “\$” to “US\$”. But in the Track Changes Word file I am reviewing, I still see many instances of just “\$.” Please change these to “US\$”.

Response: Thank you for catching this. Now fixed throughout the text.

3) LL528-530: “..., we know that secondary forest in central Panama accrues carbon at an annual smoothed rate of 6.9 Mg CO₂e in aboveground tree biomass⁶² (AGB) or approximately \$18 per Mg CO₂e at \$130 ha⁻¹.”

This should be changed to: “..., we know that secondary forest in central Panama accrues carbon at an annual smoothed rate of 6.9 Mg CO₂e in aboveground tree biomass⁶² (AGB), or approximately \$130 ha⁻¹ at \$18 per Mg CO₂e at \$130 ha⁻¹.”

Response: Thank you for catching this, we fixed this sentence (L424).

4) LL719-721: “While [ADD: carbon stocks in"] [delete: of] secondary forests in Agua Salud are high for moist tropical forests, at approximately 185 Mg CO₂e at 30 years³¹ (Table 3) they are about...”

Response: We have now fixed this (L 606-607).

5) Reading your response to my comment number 5 on your original, I believe your characterization of the “silvopasture system” is misleading. My comment was that “5) I did not see any mention of C payments for silvopasture. If this is correct, could you explain why tree cover in silvopasture is not receiving any C payments? There certainly are PES programs in tropical countries that support silvopasture establishment, so it would be helpful to understand why no C payments for this land use were considered.” Your response was: “This is a fair question. We have reviewed the literature for this response below. We explain that silvopasture means different things to different people. As we state, in our case it mostly means improved pasture and restricting access to streams.”

All definitions of silvopasture that I am familiar with in the published literature describe it as the purposive integration of trees into a grazing system (hence the term “silvo”). Even a quick Google search of the term brings up only results that meet this description. Improved pasture and restricted stream access does not meet the silvopasture definition. If your “silvopasture” system does not include the planting or retention of trees as a feature of the grazing lands, then I suggest you use a different descriptor for this pasture system. “Improved pasture” seems like a perfectly reasonable choice, and you can explain that this also includes some restriction on stream access. But a riparian tree buffer that is fenced off from grazing lands does not integrate trees into grazing lands (the defining

feature of a silvopasture system). Using silvopasture to describe a system that does not include trees in pasture lands in some purposive manner risks conjuring up an incorrect image of what this system looks like, for anyone who is familiar with silvopasture systems. Note that I do appreciate the discussion on silvopasture that you provide in your response to reviewers, in which you explain clearly what your “silvopasture” system looks like. However, I did not see any definition of silvopasture, or any description of the system that you describe as silvopasture, in the revised manuscript. Thus, my concern about using the term silvopasture for this system remains unmitigated.

Response: Thank you for bringing up this point. We did not explain well our reasoning in the original response to your comment and it seems we misunderstood your comment. Our system does indeed include tree retention and living fences, in excess of the streamside buffers. We also have a fenced padlock systems where cattle are rotated to different areas throughout the year, a feature of silvopasture systems but not of traditional systems. Thus, we do consider it a silvopasture system.

Below we have an aerial photo of part of our research area taken in 2008. You can see that the area marked “Native Species” is devoid of trees save the stream side buffers. This is part of one of our plantation blocks from which we used data in this ms. You can also see a small catchment marked “Naturally Regenerating Secondary Forest”. This is our secondary forest watershed (note: has 2 of 54 sites from our landscape scale sampling of secondary forest). Note also the Silvopasture System. You can see that we have a high density of trees that were retained. As they were retained, and are already quite large, the incremental carbon gain is very small.

Our silvopasture system was initially established by contractors hired by the Panama Canal Authority (ACP). They intended to increase the density of trees on our wooded pasture, but the trees died. Our financial analysis of silvopasture does not include this failure but rather the costs incurred by fencing, reseeding pasture, and management. We were not thinking of the tree retention in our response to you as we both took them for granted and took different (broader) direction to discuss silvopasture systems. A better response would have been to explain that we do indeed retain trees (as evidenced by the photo below) and that these trees provide shade benefits to the cattle. Our experience in our plantations and from the literature is that, given their size, they accrue relatively small amounts of carbon. They are also low value trees such that given their density they will generate negligible income, income that would likely break even with harvesting costs.

In our longer response about cattle, we did take the opportunity to mention that silvopasture does often mean different things to different people. While we agree with the definition of silvopasture provided and agree that many cases around the world practice silvopasture by that definition, in our experience in Panama, we have noticed that silvopasture does not necessarily follow all of the principles outlined in that definition. Additionally, in the papers we read on the topic, it seems that silvopasture in the tropics is sometimes practiced (often in Panama) differently. For that reason, we felt it was important to mention in our response that there seems to be some flexibility in the way silvopasture is defined.